# CAUSAL-EPIG: A PREDICTION-ORIENTED ACTIVE LEARNING FRAMEWORK FOR CATE ESTIMATION

## ABSTRACT

Estimating the Conditional Average Treatment Effect (CATE) is often constrained by the high cost of obtaining outcome measurements, making active learning essential. However, conventional active learning strategies suffer from a fundamental objective mismatch. They are designed to reduce uncertainty in model parameters or in observable factual outcomes, failing to directly target the unobservable causal quantities that are the true objects of interest. To address this misalignment, we introduce the principle of causal objective alignment, which posits that acquisition functions should target unobservable causal quantities, such as the potential outcomes and the CATE, rather than indirect proxies. We operationalize this principle through the Causal-EPIG framework, which adapts the information-theoretic criterion of Expected Predictive Information Gain (EPIG) to explicitly quantify the value of a query in terms of reducing uncertainty about unobservable causal quantities. From this unified framework, we derive two distinct strategies that embody a fundamental trade-off: a comprehensive approach that robustly models the full causal mechanisms via the joint potential outcomes, and a focused approach that directly targets the CATE estimand for maximum sample efficiency. We provide theoretical justification for our framework, establishing a formal link between our information-theoretic objective and the minimization of CATE estimation error. Extensive experiments demonstrate that our strategies consistently outperform standard baselines, and crucially, reveal that the optimal strategy is context-dependent, contingent on the base estimator and data complexity. Our framework thus provides a principled guide for sample-efficient CATE estimation in practice.

## 1 INTRODUCTION

Understanding the causal effects of interventions is central to reliable decision-making in complex domains. Causal inference provides a principled framework for this purpose by modeling the underlying dependencies in real-world data (Pearl, 2009; Hernan & Robins, 2023; Wager, 2024). Its importance is evident across domains such as healthcare (Foster et al., 2011), economics (Heckman, 2000), and recommendation systems (Gao et al., 2024a), where accurately assessing the impact of actions is critical for designing effective policies and delivering personalized interventions. Estimating the Conditional Average Treatment Effect (CATE) is a key problem in this context, as it captures how treatment effects vary across individuals (Künzel et al., 2019). While randomized controlled trials remain the gold standard for causal inference, they are often impractical due to prohibitive costs and ethical barriers (Benson & Hartz, 2017). Consequently, researchers increasingly rely on observational data, which scale more readily but introduce the additional challenge of controlling for confounding to ensure valid causal conclusions (Imbens & Rubin, 2015; Chernozhukov et al., 2024).

Beyond the challenge of controlling for confounding, a critical practical constraint in observational studies is the acquisition of ground-truth outcome data. This typically requires a costly process, such as expert annotation or long-term patient follow-up, to obtain a reliable outcome for each subject (Nwankwo et al., 2025). Consequently, in many real-world scenarios, this process is expensive, logistically demanding, and subject to privacy or ethical restrictions (Gao et al., 2024b; Kallus & Mao, 2025; Tipton & Mamakos, 2025). In healthcare, for example, measuring outcomes may require costly diagnostic tests or invasive procedures such as biopsies and large-scale tumor imaging, where the resulting label scarcity can severely impact the accuracy of CATE estimation (Bi et al., 2019; Wen et al., 2025). In economics and the social sciences, outcomes such as long-term income trajectories

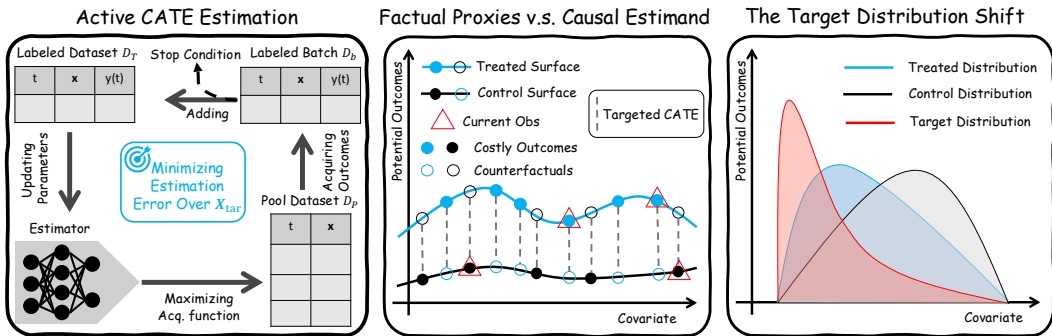

Figure 1: **(Left)** Illustrates the pool-based active learning pipeline for CATE estimation. **(Middle)** Highlights the fundamental proxy-target disconnect: the goal is to learn the CATE, the data consist only of single factual outcomes as indirect proxies. **(Right)** Shows the challenge of target distribution shift, where the sampling pool differs from the target population.

or behavioral changes often require extensive, costly follow-up (McKenzie, 2012). These resource constraints are further compounded when the study population differs systematically from the target population of interest (Kern et al., 2024). For instance, a health maintenance organization in California might need to rely on evidence from a study conducted years prior in Switzerland, whose participants fail to reflect the heterogeneity of the local population (Kallus et al., 2018). This challenge of generalizing findings across populations, formally known as ensuring external validity (Rothwell, 2005) or, more specifically, transportability (Bareinboim & Pearl, 2013; Pearl & Bareinboim, 2022), critically undermines the real-world utility of causal estimates. To address the dual challenges of resource scarcity and population shift, effective methods must be both sample-efficient and robustly target-aware to ensure CATE estimates generalize beyond the study cohort.

**Challenges.** Active learning (AL) offers a principled framework for maximizing estimation accuracy under a fixed budget, yet its application to CATE estimation is hindered by a fundamental challenge. Standard AL methods are built for a world of factual observations, designed to reduce uncertainty about observable outcomes or model parameters. The objective of CATE estimation, however, is to precisely quantify an unobservable counterfactual difference. This misalignment between a fact-based acquisition process and a counterfactual-based goal is the primary obstacle, leading to inefficient data selection that fails to reduce uncertainty where it matters most: in the treatment effect itself.

Existing literature has made valuable progress in adapting conventional AL paradigms for CATE estimation. Seminal works (Jesson et al., 2021; Wen et al., 2025) have explored criteria like factual outcome uncertainty or information gain about model parameters. Detailed related work are provided in App. B. While an important step, these approaches largely inherit the foundational misalignment. Optimizing for such proxies, rather than the CATE itself, limits their effectiveness. This is compounded by a vulnerability to distribution shift, as their acquisition criteria typically evaluate utility over the sampling pool, which may not represent the target population. Consequently, a critical gap persists: *the need for a causally-aligned acquisition strategy designed to directly target treatment effect uncertainty while remaining robust to the distributional shifts common in causal inference.*

**Contributions.** To address the critical gap in the literature, this paper makes the following contributions. *A New Principle.* We introduce the principle of causal objective alignment, arguing that the structural disconnect between observable data and the causal estimand mandates acquisition functions that are explicitly designed for the final causal goal (Sec. 3). *A Novel Information-Theoretic Framework.* We develop Causal-EPIG, a novel information-theoretic framework that operationalizes our principle (Sec. 4.1). From this unified framework, we derive two distinct, principled acquisition strategies: one that models the foundational potential outcomes, and a second that directly targets the final CATE estimand. *Broad Model Compatibility.* We demonstrate that Causal-EPIG is a flexible framework that naturally accommodates a range of popular Bayesian CATE estimators (Sec. 4.2), including Gaussian Process (GP)-based models like Causal Multi-task GP (Alaa & Van Der Schaar, 2017) and Non-Stationary GP (Alaa & Schaar, 2018), as well as the tree-based Bayesian Causal Forests (Hahn et al., 2020). *Theoretical Justification.* We provide a formal theoretical justification for our framework (Sec. 4.4), establishing a rigorous link between our acquisition objective and

the minimization of CATE estimation error (Prop. 1), proving the theoretical superiority of our prediction-oriented utilities over parameter-based baselines (Prop. 2), and providing a novel convergence analysis that bounds the posterior uncertainty under our greedy acquisition strategy (Thm. 1). *Extensive Empirical Validation.* We conduct comprehensive experiments showing that both strategies derived from our framework significantly outperform a wide array of baselines (Sec. 5). Crucially, our results validate our central hypothesis that the choice between the comprehensive and focused strategies embodies a context-dependent trade-off, providing nuanced evidence that the optimal form of causal alignment depends on the interplay between the base model and the problem's nature.

## 2 PRELIMINARIES AND PROBLEM SETUP

**Potential Outcomes and CATE Estimation.** Our analytical framework is grounded in the Neyman-Rubin potential outcomes model (Rubin, 2005). We define the random variables $\mathrm{x}$, $\mathrm{t}$, and $\mathrm{y}$ to represent the covariates, treatment, and outcome, respectively, with domains $\mathcal{X}$, $\{0, 1\}$, and $\mathcal{Y}$. We denote realizations by $\boldsymbol{x}$, $t$, and $y$. The two potential outcomes are $\mathrm{y}(0)$ and $\mathrm{y}(1)$, corresponding to the outcome under control and treatment. The propensity score is $\pi(\boldsymbol{x}) = p(\mathrm{t} = 1 | \mathrm{x} = \boldsymbol{x})$. Our primary goal is to estimate the CATE, defined as $\tau(\boldsymbol{x}) := \mathbb{E}[\mathrm{y}(1) - \mathrm{y}(0) \mid \mathrm{x} = \boldsymbol{x}]$. For a detailed summary of our notation, see App. C. To ensure identifiability, we impose the following assumptions.

**Assumption 1** *Unconfoundedness: Given the covariates $\mathrm{x}$, treatment assignment $\mathrm{t}$ is independent of the potential outcomes, i.e., $(\mathrm{y}(1), \mathrm{y}(0)) \perp\!\!\!\perp \mathrm{t} | \mathrm{x}$. This implies that $\mathrm{x}$ captures all common causes of treatment and outcome. Positivity (Common Support): For any covariates $\boldsymbol{x}$, the probability of receiving any given treatment is non-zero: $0 < \pi(\boldsymbol{x}) < 1$. SUTVA (Stable Unit Treatment Value): An individual's potential outcomes are unaffected by the treatment assignments of others (No Interference), and the observed outcome is the potential outcome corresponding to the treatment received, i.e., $\mathrm{y} = \mathrm{t}\mathrm{y}(1) + (1 - \mathrm{t})\mathrm{y}(0)$ (Consistency).*

Under Ass. 1, the CATE becomes identifiable as the difference in the conditional expectations of the observed outcome, which we denote $f(\boldsymbol{x}, t) := \mathbb{E}[\mathrm{y} | \mathrm{x} = \boldsymbol{x}, \mathrm{t} = t]$. This is expressed as:

$$\tau(\boldsymbol{x}) = f(\boldsymbol{x}, 1) - f(\boldsymbol{x}, 0) = \mathbb{E}[\mathrm{y} | \mathrm{x} = \boldsymbol{x}, \mathrm{t} = 1] - \mathbb{E}[\mathrm{y} | \mathrm{x} = \boldsymbol{x}, \mathrm{t} = 0]. \tag{1}$$

### 2.1 POOL-BASED ACTIVE ESTIMATION OF CATE

In this setting, we begin with a large unlabeled pool of instances $D_P = \{(\boldsymbol{x}_i, t_i)\}_{i=1}^{n_P}$ and a small, often initially empty, labeled training set $D_T = \{(\boldsymbol{x}_i, t_i, y_i)\}_{i=1}^{n_T}$. The active learning loop proceeds iteratively: a model trained on the current $D_T$ informs an acquisition function, which selects a batch of $n_b$ instances from $D_P$ to be labeled. These are added to $D_T$, and the process repeats until a budget of $n_B$ labels is exhausted (Jesson et al., 2021; Qin et al., 2021). Our objective is to learn a CATE model, $\hat{\tau}(\boldsymbol{x})$, that is accurate over a specific target distribution of interest, $p_{\text{tar}}(\boldsymbol{x})$, which may differ from the distribution of the sampling pool $p_{\text{pool}}(\boldsymbol{x})$. To formalize this, we evaluate performance using the square root of the Precision in Estimating Heterogeneous Effects ($\sqrt{\epsilon_{\text{PEHE}}}$) (Hill, 2011). This metric is defined as the root mean squared error over the target distribution and is empirically estimated using a finite target set $\boldsymbol{X}_{\text{tar}}$ drawn from $p_{\text{tar}}(\boldsymbol{x})$:

$$\sqrt{\epsilon_{\text{PEHE}}}[\hat{\tau}] := \sqrt{\mathbb{E}_{p_{\text{tar}}(\boldsymbol{x})}\left[(\hat{\tau}(\boldsymbol{x}) - \tau(\boldsymbol{x}))^2\right]} \approx \sqrt{\frac{1}{|\boldsymbol{X}_{\text{tar}}|} \sum_{\boldsymbol{x} \in \boldsymbol{X}_{\text{tar}}} (\hat{\tau}(\boldsymbol{x}) - \tau(\boldsymbol{x}))^2}. \tag{2}$$

**Remark 1 (Observational Constraint vs. Experimental Design)** *A key constraint in our setup is that we operate on observational data, even during acquisition. For any instance $(\boldsymbol{x}_i, t_i)$, we can only query its pre-existing outcome $\mathrm{y}_i(t_i)$ and cannot intervene to assign a new treatment and observe the counterfactual. This limitation distinguishes our problem from adaptive experimental design, which requires the freedom to assign treatments (Toth et al., 2022; Kato et al., 2024; Cha & Lee, 2025; Klein et al., 2025; Zhang et al., 2025). This constraint is common in sensitive domains like healthcare and social sciences, where treatment assignment is governed by external factors. A more detailed discussion of the related literature on adaptive experimental design is provided in App. B.3.*

> 📌 **Key Objective.** In this problem setup, the central challenge is to design a principled utility function, $U(\cdot)$, that quantifies the informativeness of any single candidate data point $(\boldsymbol{x}, t)$ from the pool. The acquisition strategy is then to select the candidate, denoted $(\boldsymbol{x}_s, t_s)$, that is deemed most valuable by maximizing this function:
>
> $$(\boldsymbol{x}_s, t_s) = \underset{(\boldsymbol{x}, t) \in D_P}{\arg \max} U(\boldsymbol{x}, t \mid D_T, \boldsymbol{X}_{\text{tar}}). \tag{3}$$
>
> While this defines the selection of a single instance, this process is typically extended to the batch setting by greedily selecting the $n_b$ candidates that yield the highest utility scores.

## 3 ALIGNING ACTIVE LEARNING WITH CAUSAL OBJECTIVES

This section analyzes the unique structure of active outcome acquisition for CATE estimation, revealing a fundamental misalignment with standard AL paradigms. We show that this misalignment points toward a core principle, Causal Objective Alignment (COA), that should guide the design of principled and sample-efficient acquisition strategy within this domain.

**From Indirect Proxies to the Causal Estimand.** In standard AL, the path from query to knowledge is direct. The learning objective is aligned with the data-generating process: one queries a point $\boldsymbol{x}_i$ to observe a label $y_i$, which is a direct (though noisy) signal for the target function $f(\boldsymbol{x}_i)$. Naïve applications of this paradigm to CATE estimation simply adopt these standard targets: they might focus on the uncertainty of the observable response surface, $f(\boldsymbol{x}, t)$, or on the uncertainty of the model's internal parameters, $\theta$. However, this creates a fundamental mismatch, as illustrated in Fig. 1. Both the data we can acquire (factual outcomes) and the model's parameters are only indirect proxies for our true inferential goal. This goal is to understand the complete unobservable causal mechanism, which is characterized by the two potential outcome surfaces, $y(0)$ and $y(1)$, and the CATE function, $\tau(\boldsymbol{x})$, derived from them, as shown in Eq. 1. This profound disconnect between indirect proxies and the unobservable causal quantities that truly matter motivates our core design principle:

**Principle 1 (Causal Objective Alignment)** *An effective acquisition strategy for active CATE estimation should be causally aligned. Its utility function should quantify the value of a query by targeting unobservable causal quantities, such as the potential outcomes or the CATE itself, to ensure alignment with the final inferential goal, rather than indirect proxies.*

The COA principle's requirement that utility be quantified relative to a fixed target population, $\boldsymbol{X}_{\text{tar}}$, naturally frames active CATE estimation as a transductive learning problem (a connection detailed in App. B.2.1). This shift in perspective from a general inductive model to one tailored for a specific set of individuals illuminates a conceptual spectrum of acquisition strategies. This spectrum ranges from naïve approaches targeting indirect proxies (e.g., factual uncertainty) to sophisticated, causally-aligned strategies. Within these aligned approaches, the principle reveals a powerful dichotomy: strategies that target the foundational components of the causal mechanism (the potential outcome surfaces), versus those that directly target the final causal effect itself. The importance of this alignment is amplified under distribution shift ($p_{\text{tar}}(\boldsymbol{x}) \neq p_{\text{pool}}(\boldsymbol{x})$), where misaligned objectives may fail entirely to reduce uncertainty for the target population. This unified perspective, grounding the problem in both causal alignment and a transductive objective, provides the robust conceptual foundation for the Causal-EPIG framework we now introduce.

## 4 ACTIVE CATE ESTIMATION VIA CAUSAL-EPIG

This section operationalizes the COA principle by introducing the Causal-EPIG framework: a unified, information-theoretic approach to designing acquisition functions. Instead of proposing a single "best" criterion, we demonstrate that this framework naturally gives rise to two distinct and principled strategies, embodying a fundamental trade-off between modeling robustness and directness, the optimal balance of which may depend on both the underlying CATE estimator and the data-generating process. We first present the formal definitions of these strategies and discuss their conceptual differences. We then demonstrate the framework's compatibility with advanced Bayesian CATE estimators. Further implementation details are provided in App. E.

Table 1: Comparison of information-theoretic acquisition functions for active CATE estimation. For brevity, $D'_T$ denotes the training set augmented with a candidate point: $D'_T = D_T \cup \{(\boldsymbol{x}, t)\}$.

| | **Non-Causal-Aware** | **Causal-Aware** |
|---|---|---|
| **EIG** | $\mathrm{I}(y; \theta \mid D'_T)$ $\quad$ ($\mu$-BALD) | $\mathrm{I}(y; \theta_\tau \mid D'_T)$ $\quad$ (Causal-EIG) |
| **EPIG** | $\mathbb{E}_{p_{\mathrm{pool}}(\boldsymbol{x}^*, t^*)}\Big[\mathrm{I}(y; y^* \mid (\boldsymbol{x}^*, t^*), D'_T)\Big]$ | $\mathbb{E}_{p_{\mathrm{tar}}(\boldsymbol{x}^*)}\big[\mathrm{I}(y; (y^*(0), y^*(1)) \mid \boldsymbol{x}^*, D'_T)\big]$ $\quad$ (PO-based) $\newline$ $\mathbb{E}_{p_{\mathrm{tar}}(\boldsymbol{x}^*)}\Big[\mathrm{I}(y; \tau(\boldsymbol{x}^*) \mid \boldsymbol{x}^*, D'_T)\Big]$ $\quad$ (CATE-based) |

### 4.1 CAUSAL-EPIG: AN INFORMATION-THEORETIC ACQUISITION FUNCTION

Our framework is grounded in the information-theoretic concept of mutual information (MI). Formally denoted as $\mathrm{I}(a; b) = \mathrm{H}(a) - \mathrm{H}(a \mid b)$, where $\mathrm{H}(\cdot)$ represents entropy, MI quantifies the information that a random variable a provides about another variable b. Equivalently, it measures the expected reduction in uncertainty about b gained from observing a. The design of Causal-EPIG is best motivated by a direct contrast with these standard AL criteria, as illustrated in Tab. 1. As defined above, methods like EIG/BALD are *parameter-focused*, aiming to reduce uncertainty over the model parameters ($\theta$). This objective is indirect; reducing global parameter uncertainty does not guarantee a targeted reduction in CATE uncertainty (Houlsby et al., 2011; Jesson et al., 2021). This limitation persists even for causal adaptations. For instance, in models like BCF that adopt a separable structure, $f(\boldsymbol{x}, t) = \mu(\boldsymbol{x}) + t \cdot \tau(\boldsymbol{x})$, with parameters $\theta = (\theta_\mu, \theta_\tau)$. In such models, one could target the CATE parameters $\theta_\tau$ specifically. However, this still focuses on the model's internal representation rather than its final predictive output (Fawkes et al., 2025). Standard EPIG elevates the objective by targeting a future prediction ($y^*$), but it remains tethered to a single *factual* outcome. This is insufficient because CATE is inherently a comparative quantity, $\tau(\boldsymbol{x}) = \mathbb{E}[y(1) - y(0)|\boldsymbol{x}]$. A data point that is highly informative for one potential outcome might offer little information about the other, and thus may not efficiently reduce uncertainty about their difference (Smith et al., 2023).

☛ **A Comprehensive Strategy: Targeting the Causal Mechanism (Causal-EPIG-$\mu$).** A direct application of our COA principle is to target the complete causal mechanism for a target individual, which is fully described by the joint distribution of their potential outcomes, $(y^*(0), y^*(1))$. This comprehensive approach correctly accounts for the inherent dependence between the two outcomes. This leads to our Potential Outcome-based (PO-based) strategy, Causal-EPIG-$\mu$:

$$\text{Causal-EPIG-}\mu(\boldsymbol{x}, t) \coloneqq \mathbb{E}_{p_{\mathrm{tar}}(\boldsymbol{x}^*)}\left[\mathrm{I}(y; (y^*(0), y^*(1)) \mid \boldsymbol{x}^*, D'_T)\right]. \tag{4}$$

By seeking data that maximally reduces uncertainty over this joint distribution, Causal-EPIG-$\mu$ aims to build a holistic and robust statistical model of the foundational surfaces from which the CATE is derived. The objective of this strategy is to obtain a more complete and nuanced picture of the underlying individual-level mechanism. A potential consequence is that some acquisition budget may inevitably be dedicated to resolving uncertainty in the prognostic baseline (i.e., the average outcome) rather than exclusively clarifying the contrast between the potential outcomes. For completeness, we also discuss a simpler, additive variant in App. F.2, which approximates this broader objective.

☛ **A Focused Strategy: Directly Targeting the Causal Estimand (Causal-EPIG-$\tau$).** In contrast to the comprehensive strategy, an alternative approach is to focus the entire acquisition budget on the final inferential goal itself: the CATE function $\tau(\boldsymbol{x}^*)$. This focused strategy is designed to yield maximum sample efficiency for CATE estimation when the causal effect is a sufficiently learnable signal, by prioritizing data points that most directly resolve uncertainty in this causal estimand. Formally, we define the Causal-EPIG-$\tau$ utility as the expected information gain about the CATE:

$$\text{Causal-EPIG-}\tau(\boldsymbol{x}, t) \coloneqq \mathbb{E}_{p_{\mathrm{tar}}(\boldsymbol{x}^*)}\Big[\mathrm{I}(y; \tau(\boldsymbol{x}^*) \mid \boldsymbol{x}^*, D'_T)\Big]. \tag{5}$$

The mutual information term represents the expected reduction in CATE posterior entropy. An equivalent and computationally useful formulation uses the KL divergence to frame this utility as the expected belief update about the CATE, $\tau(\boldsymbol{x}^*)$, after a potential observation $y$:

$$\text{Causal-EPIG-}\tau(\boldsymbol{x}, t) = \mathbb{E}_{p_{\mathrm{tar}}(\boldsymbol{x}^*)}\left[\mathrm{KL}\Big(p(y, \tau(\boldsymbol{x}^*) \mid \boldsymbol{x}^*, D'_T) \,\|\, p(y \mid D'_T)p(\tau(\boldsymbol{x}^*) \mid \boldsymbol{x}^*, D'_T)\Big)\right]. \tag{6}$$

Intuitively, a high utility score signifies that an observation at $(\boldsymbol{x}, t)$ is expected to significantly change our beliefs about the CATE in the target population, marking it as a highly informative candidate.

**Positioning the Causal-EPIG Framework.** Our Causal-EPIG framework is distinguished from related methods, particularly the Causal-BALD family (Jesson et al., 2021), by its fundamentally prediction-focused objective. This distinction is crucial: while a method like $\tau$-BALD calculates the information a CATE prediction provides about the model's internal parameters, our Causal-EPIG-$\tau$ calculates the information a future factual *observation* provides about a target CATE prediction. Our approach thus bypasses the parameters to directly target the final quantity of interest. A second key design axis lies within our framework, concerning how information gain across the target population is aggregated. The mean-marginal formulation, which we adopt in this work, approximates the total gain by averaging the information for each target point independently. In practice, this expectation is estimated via a simple sum over a finite target set. In contrast, a more theoretically complete global formulation would compute the mutual information with the entire vector of target predictions jointly, $\mathrm{I}(y; \boldsymbol{\tau})$, thereby directly leveraging all inter-target dependencies (Hübotter et al., 2024). Our choice represents a pragmatic trade-off between computational scalability and theoretical completeness. We provide a detailed taxonomy of these formulations in App. F.2.

**The Comprehensiveness-Focus Trade-off.** The choice between the comprehensive and focused strategies is not absolute; rather, it ultimately depends on the problem context. The optimal approach is determined by the inductive biases of the base estimator: models that directly parameterize the CATE function, such as BCF, may benefit from the focused Causal-EPIG-$\tau$, while models that instead characterize the outcome surfaces, such as Gaussian Processes, may gain more from the robustness of Causal-EPIG-$\mu$. The complexity of the data distribution also matters: a simple, low-noise CATE function is well aligned with the CATE-based strategy, whereas a more complex causal signal may be more reliably captured as a natural byproduct of the robust surface modeling encouraged by the PO-based strategy. Ultimately, our framework does not claim a universally superior solution but instead provides principled tools whose effectiveness remains inherently context-dependent.

## 4.2 REALIZATION WITH BAYESIAN CATE ESTIMATORS

While model-agnostic, the Causal-EPIG framework's practical implementation varies by CATE estimator. We outline realization strategies for two major classes of Bayesian models.

**Exact Realization with GP Models.** For CATE estimators based on GPs, such as CMGP (Alaa & Van Der Schaar, 2017) and NSGP (Alaa & Schaar, 2018), the joint posterior predictive distribution over any set of points is, by construction, a multivariate Gaussian. Consequently, the required predictive variances and covariances can be extracted directly from the GP's analytical posterior covariance matrix. In this ideal setting, the mutual information has an exact closed-form solution. For example, for two jointly Gaussian variables, this is given by:

$$\mathrm{I}(\mathrm{a}; \mathrm{b}) = \frac{1}{2} \log \frac{\mathrm{Var}[\mathrm{a}]\mathrm{Var}[\mathrm{b}]}{\mathrm{Var}[\mathrm{a}]\mathrm{Var}[\mathrm{b}] - \mathrm{Cov}[\mathrm{a}, \mathrm{b}]^2}. \tag{7}$$

This allows for a highly efficient and exact implementation of Causal-EPIG with GP-based models.

**Approximate Realization for General Bayesian Models.** For more complex models where the posterior is analytically intractable and represented by samples, such as with the MCMC output of Bayesian regression tree (Hill, 2011) and BCF (Hahn et al., 2020), a direct computation of the mutual information is infeasible. To make Causal-EPIG tractable for this broad class of models, we employ a computationally efficient Gaussian approximation, following prior work (Kirsch, 2023; Jesson et al., 2021). This strategy involves fitting a multivariate Gaussian to the posterior draws, with the mean vector and covariance matrix estimated empirically from the set of $n_M$ posterior samples. For instance, when applying this to BCF, the crucial covariance term is computed from its MCMC draws as $\mathrm{Cov}[y, \tau(\boldsymbol{x}^*)] = \widehat{\mathrm{Cov}}(\{f(\boldsymbol{x}, t|\theta_j)\}_{j=1}^{n_M}, \{\tau(\boldsymbol{x}^*|\theta_j)\}_{j=1}^{n_M})$. Once this approximation is made, we can reuse the convenient closed-form solution for mutual information (Eq. 7). This approach provides a versatile recipe for pairing Causal-EPIG with a wide range of sample-based Bayesian models, bypassing the need for expensive nested Monte Carlo simulations.

### 4.3 BUDGETED ACQUISITION ALGORITHM

The Causal-EPIG utility serves as our acquisition function for selecting the most informative data points. We employ this utility in an iterative active learning strategy designed to estimate the CATE under a fixed budget (Qin et al., 2021; Jesson et al., 2021). This process, shown in Fig. 1 and detailed in Alg. 1 (App. E.1), begins with a warm-start phase where a small, random batch of data is labeled. Subsequently, in each round, the algorithm computes the Causal-EPIG utility for all candidates in the unlabeled pool. A batch of points with the highest utility scores is then selected and their outcomes are queried. The CATE model is subsequently retrained on the newly expanded labeled set. This cycle of scoring, acquiring, and retraining continues until the budget is exhausted.

### 4.4 THEORETICAL ANALYSIS

We now provide a theoretical justification for our Causal-EPIG framework. Since conducting a fully general analysis for arbitrary Bayesian CATE estimators is challenging, we focus on joint GP-based models, which our acquisition strategies are designed to accommodate. We first show that the optimal Bayesian AL objective for CATE estimation reduces to minimizing the posterior CATE variance. We then analyze the convergence behavior of our Causal-EPIG strategy under this objective.

#### 4.4.1 OBJECTIVE: CATE ERROR VS. JOINT VARIANCE

The following result connects the CATE estimation error to a tractable model-based criterion and shows that the relevant acquisition objective is the posterior joint variance of the CATE.

**Proposition 1** *Assume $f(\boldsymbol{x}, 0)$ and $f(\boldsymbol{x}, 1)$ are modeled by a joint GP, and the CATE estimator is the posterior mean $\hat{\tau}_s(\boldsymbol{x}) = \mathbb{E}_s[\tau(\boldsymbol{x})]$. For pool-based active CATE estimation, the optimal choice to minimize the expected model-based estimation error, $\mathbb{E}_{s+1}\big[\epsilon_{\mathrm{PEHE}}^{\mathcal{M}}(\hat{\tau}_{s+1})\big]$, simplifies under the GP assumption to minimizing the integrated posterior CATE variance:*

$$\arg\min_{(\boldsymbol{x},t)\in D_P} \mathbb{E}_{s+1}\big[\epsilon_{\mathrm{PEHE}}^{\mathcal{M}}(\hat{\tau}_{s+1})\big] = \arg\min_{(\boldsymbol{x},t)\in D_P} \mathbb{E}_{p_{\mathrm{tar}}(\boldsymbol{x})}\big[\mathbb{V}\mathrm{ar}_{s+1}[\tau(\boldsymbol{x})]\big], \tag{8}$$

*where $\mathbb{V}\mathrm{ar}_{s+1}[\tau(\boldsymbol{x})]$ is the posterior variance of the Bayesian random variable $\tau(\boldsymbol{x})$, which explicitly retains the joint posterior structure:*

$$\mathbb{V}\mathrm{ar}_{s+1}[\tau(\boldsymbol{x})] = \mathbb{V}\mathrm{ar}_{s+1}[f(\boldsymbol{x}, 1)] + \mathbb{V}\mathrm{ar}_{s+1}[f(\boldsymbol{x}, 0)] - 2\,\mathrm{Cov}_{s+1}\big(f(\boldsymbol{x}, 1), f(\boldsymbol{x}, 0)\big). \tag{9}$$

A detailed proof is provided in App. G.6.

#### 4.4.2 CONVERGENCE OF POSTERIOR UNCERTAINTY

Prop. 1 shows that our acquisition strategy should aim to reduce the joint CATE variance $\mathbb{V}\mathrm{ar}[\tau(\boldsymbol{x})]$. To analyze its convergence, we examine the behavior of the underlying potential outcome components. We map this component-level problem to the transductive active learning (TAL) framework (Hübotter et al., 2024) by modeling the potential outcomes $f_t(\boldsymbol{x})$ as a single GP $f(\tilde{\boldsymbol{x}})$ over an augmented input space $\tilde{\mathcal{X}} = \mathcal{X} \times \{0, 1\}$ with a multitask kernel. As detailed in App. G.1, this construction induces an augmented target space $\tilde{\boldsymbol{X}}_{\mathrm{tar}}$ (potential outcomes) and an augmented pool space $\tilde{\mathcal{D}}_P$ (factual observations), making our objective equivalent to reducing posterior uncertainty over $\tilde{\boldsymbol{X}}_{\mathrm{tar}}$ by querying from $\tilde{\mathcal{D}}_P$. Our convergence analysis focuses on the global, joint PO-based strategy, denoted Causal-EPIG-$\mu$-G (discussed in App. F.2), as it is a direct instantiation of the Global Information Theoretic Learning (ITL) strategy from the TAL framework (Hübotter et al., 2024, Eq. 2). To analyze its convergence, we adapt the Global ITL analysis (Hübotter et al., 2024, Thm. 3.3) from the TAL framework and first define two key quantities.

**Definition 1** *Let $\tilde{\boldsymbol{X}}_{\mathrm{tar}}$ and $\tilde{\mathcal{D}}_P$ be defined as above. The global information capacity $\gamma_{n_B}$ from $n_B$ observations, and the irreducible uncertainty $\eta_{\mathcal{D}_P}^2(\tilde{\boldsymbol{x}}^*)$ for a target $f(\tilde{\boldsymbol{x}}^*) = f(\boldsymbol{x}^*, t)$, are:*

$$\gamma_{n_B} \stackrel{\mathrm{def}}{=} \max_{\tilde{\boldsymbol{X}} \subseteq \tilde{\mathcal{D}}_P, |\tilde{\boldsymbol{X}}| \le n_B} \mathrm{I}\big(f_{\tilde{\boldsymbol{X}}_{\mathrm{tar}}}; \boldsymbol{y}_{\tilde{\boldsymbol{X}}}\big), \qquad \eta_{\mathcal{D}_P}^2(\tilde{\boldsymbol{x}}^*) \stackrel{\mathrm{def}}{=} \mathbb{V}\mathrm{ar}[f(\tilde{\boldsymbol{x}}^*) \mid \mathcal{D}_P]. \tag{10}$$

To prove convergence, the TAL framework requires the utility function to be submodular. We formalize this in our context:

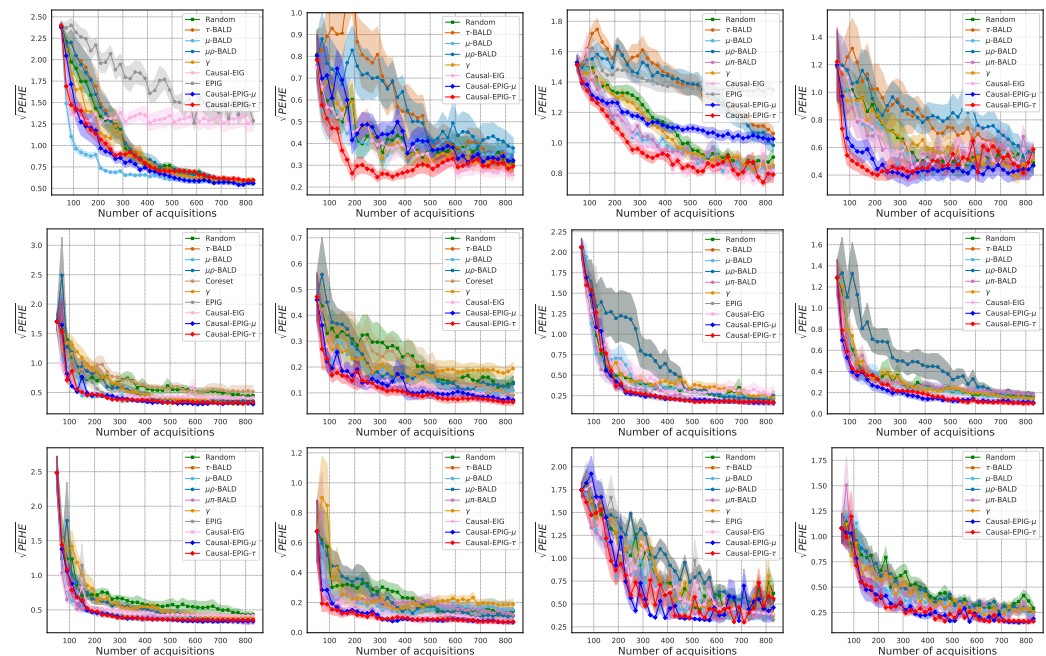

Figure 2: Comparison of $\sqrt{\text{PEHE}}$ on two simulation datasets of three CATE estimators (BCF, CMGP and NSGP, arranged by row) on the CausalBALD and Hahn (linear) simulation datasets. The columns represent the experimental setup for each dataset: regular and a distributional shift setting.

**Assumption 2** *The utility function for the Causal-EPIG-$\mu$-G strategy, defined as the joint information gain $\psi_{\tilde{\boldsymbol{X}}_{\text{tar}}}(\tilde{\boldsymbol{X}}) \overset{def}{=} \mathrm{I}(f_{\tilde{\boldsymbol{X}}_{\text{tar}}}; \mathrm{y}_{\tilde{\boldsymbol{X}}})$, is a submodular set function.*

This assumption is essential for the greedy Causal-EPIG-$\mu$-G strategy to provide a constant-factor approximation of the optimal information gain, which is a key component of the TAL convergence proof. A detailed discussion of this assumption and its validity is provided in App. G.3. We now bound the marginal variance under this assumption.

**Theorem 1** *Suppose the data acquisition follows the greedy Causal-EPIG-$\mu$-G strategy and let $n_B$ denote the total number of acquired outcomes from $\mathcal{D}_P$. Under standard GP assumptions and Ass. 2, there exists a constant $C > 0$ such that for any $n_B \geq 1$ and for each target potential outcome $\tilde{\boldsymbol{x}}^* \in \tilde{\boldsymbol{X}}_{\text{tar}}$, the marginal variance satisfies:*

$$\mathbb{V}\text{ar}[f(\tilde{\boldsymbol{x}}^*) \mid \mathcal{D}_T] \leq \eta^2_{\mathcal{D}_P}(\tilde{\boldsymbol{x}}^*) + C\big(\gamma_{n_B}/\sqrt{n_B}\big). \tag{11}$$

The convergence analysis proof for this acquisition strategy is presented in App. G.3.

## 5 EXPERIMENTAL RESULTS

To assess the sample efficiency of our Causal-EPIG framework, we conduct extensive experiments on several benchmarks. These include synthetic datasets based on the data-generating processes (DGPs) from Causal-BALD (Jesson et al., 2021) and Hahn et al. (2020), as well as two well-established semi-synthetic benchmarks: the Infant Health and Development Program (IHDP) (Hill, 2011) and AIDS Clinical Trials Group Study 175 (ACTG-175) (Hammer et al., 1996). Full details regarding the DGPs, dataset characteristics, and partitioning for each benchmark are available in App. D.

**Base Bayesian CATE Estimators, Baselines, and Metrics.** To demonstrate the flexibility of our framework, we implement Causal-EPIG with three distinct and well-established Bayesian CATE estimators: BCF, CMGP, and NSGP. These models are natural partners for our information-theoretic acquisition functions, as they provide the necessary posterior uncertainty over the CATE. For brevity, the main text focuses on these primary models; comprehensive results for all setups, including an

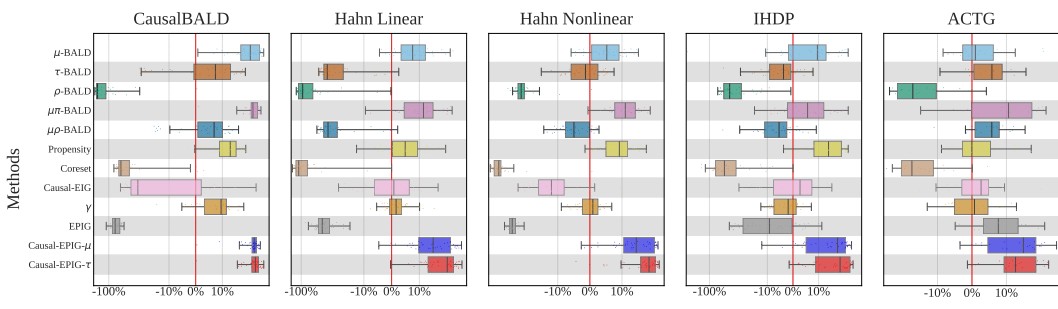

Figure 3: Average relative improvement of acquisition functions over *Random acquisition* on five datasets: CausalBALD, Hahn (linear), Hahn (nonlinear), IHDP, and ACTG-175.

additional estimator from the Causal-BALD study, are provided in App. H. For our baselines, we compare against a range of acquisition functions, including Random, $\gamma$-acquisition (S-type error rate control) (Sundin et al., 2019), coreset selection (Qin et al., 2021), Causal-EIG (Fawkes et al., 2025), and the suite of methods from Causal-BALD (Jesson et al., 2021). Detailed implementations for all methods are available in App. E. Our primary evaluation metric is the Root PEHE ($\sqrt{\hat{\epsilon}_{\text{PEHE}}}$ or $\sqrt{\text{PEHE}}$ for short; Eq. 2), computed on the target set $X_{\text{tar}}$. All results are reported as the mean and standard deviation across 10 independent runs. In addition to performance curves, we report the relative Root PEHE improvement over the Random baseline for a holistic summary of sample efficiency. This metric is calculated at each acquisition step $k$ as $(\sqrt{\text{PEHE}}_{\text{Random}}(k) - \sqrt{\text{PEHE}}_{\text{Method}}(k))/\sqrt{\text{PEHE}}_{\text{Random}}(k)$. Finally, we aggregate these point-wise improvements across all steps to visualize the distribution of performance gains for each method, offering insight into its consistency throughout the active learning process.

## 5.1 SYNTHETIC DATA

**Results.** Fig. 2 presents our main findings on the synthetic datasets, demonstrating the strong performance of the strategies derived from our Causal-EPIG framework. On these benchmarks, the focused strategy, **Causal-EPIG-$\tau$** (red curve), proves particularly effective, consistently establishing a new state of the art in sample efficiency. Across all three base estimators (BCF, CMGP, and NSGP) and in settings both with and without distribution shift, it is either the top-performing method or among the very best, rapidly converging to a lower error than all baselines. The comprehensive strategy, **Causal-EPIG-$\mu$** (blue curve), also proves to be highly effective, significantly outperforming most baseline methods. We note one insightful interaction with the base model: its performance is slightly attenuated when paired with BCF. We hypothesize this is because BCF models the prognostic effect ($\mu$) and the treatment effect ($\tau$) separately; therefore, predicting the potential outcomes required by Causal-EPIG-$\mu$ may accumulate estimation errors from both components of the BCF model. These trends are summarized in Fig. 3, which aggregates the performance gains and confirms that Causal-EPIG-$\tau$ achieves the highest average improvement. Overall, these results provide strong empirical validation for our COA principle, demonstrating that in these synthetic settings where the CATE function is well-specified, the directness of the focused Causal-EPIG-$\tau$ strategy yields superior performance. Comprehensive results, detailed analyses, and ablation studies on stability (varying initializations, pool sizes, batch sizes, and the Deep-GP estimator) are provided in App. H.1, H.2, H.5.

**Computational Considerations.** The superior sample efficiency of our Causal-EPIG framework comes at the cost of a more computationally intensive acquisition function compared to simpler baselines. This represents a deliberate trade-off. The effectiveness of our approach is therefore most pronounced in settings where the cost of labeling is the dominant factor in the data acquisition pipeline, such as in clinical trials or industrial experiments where acquiring each new label can be time-consuming and expensive. In these common real-world scenarios, the marginal computational overhead is typically negligible compared to the cost of labeling, making the trade-off highly favorable. A detailed breakdown of the per-sample runtimes is provided in App. F.3.

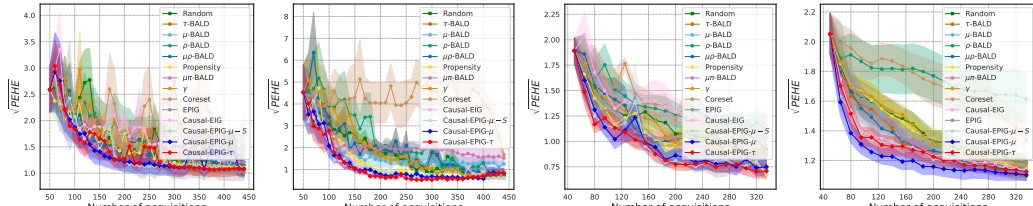

Figure 4: Comparison of $\sqrt{\text{PEHE}}$ on IHDP and ACTG-175 datasets. (1) CMGP on IHDP, (2) CMGP on IHDP under target shift, (3) CMGP on ACTG-175, and (4) BCF on ACTG-175.

## 5.2 SEMI-SYNTHETIC DATA

**Benchmarks.** We evaluate our framework on two well-established semi-synthetic benchmarks designed to mimic challenges of real-world observational studies. The first, the IHDP (Hill, 2011), simulates selection bias by removing a non-random subset of the treated group from a randomized trial. The second, ACTG-175 (Hammer et al., 1996), constructs an observational cohort by excluding participants based on their enrollment symptoms.

**Results and Analysis.** The results on these more realistic semi-synthetic benchmarks (Fig. 4 and Fig. 3) highlight the practical effectiveness of our Causal-EPIG framework and showcase the nuances of the trade-off between its two strategies. Across both the IHDP and ACTG datasets, both Causal-EPIG-$\mu$ and Causal-EPIG-$\tau$ consistently deliver top-tier performance, demonstrating the overall strength of our causally-aligned, prediction-focused approach. The strong performance of the comprehensive Causal-EPIG-$\mu$ strategy is particularly noteworthy, suggesting that its robust approach of modeling the entire causal mechanism is highly effective in these complex, lower signal-to-noise settings. Furthermore, these results underscore our thesis that the optimal acquisition strategy is context-dependent. On the IHDP benchmark, which is defined by significant selection bias, we observe that propensity-based baselines also perform competitively. This finding is expected and reinforces our core argument: specialized methods excel when the problem conditions match their design assumptions. The key advantage of the Causal-EPIG framework, therefore, is not that one of its strategies is universally dominant, but that it provides practitioners with *two distinct, powerful, and generally reliable strategies* that achieve better sample efficiency across different and challenging conditions. Full results are provided in App. H.3, H.4.

## 6 DISCUSSIONS

**Conclusion.** This work addressed the fundamental misalignment between standard active learning and CATE estimation by introducing the principle of causal objective alignment. We operationalized this principle with the Causal-EPIG framework, a flexible information-theoretic approach that yields two distinct acquisition strategies: a comprehensive strategy targeting the full potential outcome mechanism and a focused strategy targeting the CATE itself. Our extensive experiments confirmed that both of our causally-aligned strategies significantly outperform strong baselines, and more importantly, validated our central hypothesis that the choice between them embodies a context-dependent trade-off. This key finding provides strong empirical evidence for our principle: while aligning the acquisition objective with the causal goal is crucial, the optimal strategy is itself context-dependent. By providing a framework that navigates this trade-off, our work enables more cost-effective and reliable CATE estimation in critical domains.

**Limitations and Future Work.** Causal-EPIG currently assumes the absence of unobserved confounding and relies on well-calibrated posterior uncertainty from the base CATE model, which may be unreliable in low-data regimes (Zhang et al., 2025). While our framework is model-agnostic, its overall effectiveness is fundamentally bounded by the performance of the underlying CATE estimator. A promising direction is to integrate more powerful and well-calibrated models, such as CausalPFN (Balazadeh et al., 2025), to further enhance sample efficiency. Key future directions include extending the method to account for hidden confounding (Li et al., 2023), and adapting our framework for adaptive experimental design, shifting the focus from selecting which existing data to label to deciding which new interventions to perform.

## ETHICS STATEMENT

This research does not include human or animal subjects, nor does it rely on sensitive or proprietary personal data. All datasets employed are publicly available and commonly used in the community. The work poses no risks related to privacy, security, fairness, discrimination, or potential harmful applications. We have adhered to the ICLR Code of Ethics, and all aspects of this study were conducted with a commitment to integrity, transparency, and responsible research practices.

## REPRODUCIBILITY STATEMENT

We have made extensive efforts to ensure the reproducibility of this work. App. D provides detailed information on the datasets and preprocessing procedures. App. E reports the full implementation details, including model architectures, training protocols, baselines, and the corresponding hyper-parameter configurations. Further experimental results and analyses are presented in App. H. To promote transparency and reproducibility, we will make our source code publicly available upon acceptance of the paper.

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

# Appendix

## Table of Contents

## A  ACKNOWLEDGMENT OF LLM USAGE

Large language models were employed exclusively as writing assistants. Their role was limited to surface-level editing tasks, such as correcting typographical errors, improving grammar, and refining phrasing. They were not used for research design, scientific analysis, generation of results. All scientific ideas, methodologies, analyses, and conclusions are solely the responsibility of the authors.

## B  ADDITIONAL RELATED WORKS AND DISCUSSIONS

### B.1  ACTIVE CATE ESTIMATION

Our work addresses *active outcome acquisition for CATE estimation*: a setting where treatment assignments are observational, but outcome measurements are costly (Nwankwo et al., 2025). The goal is to intelligently select which outcomes to query from an existing cohort to best improve a CATE model. Existing literature in this area has primarily adapted standard active learning heuristics. One line of work focuses on diversity-based sampling, such as coreset selection, which seeks a representative subset of the covariate space (Qin et al., 2021; Wen et al., 2025). Another focuses on controlling specific causal error types rather than the overall estimation error (Sundin et al., 2019). While valuable, these methods rely on indirect proxies, such as geometric diversity or specific error metrics, that are not explicitly aligned with the primary goal of reducing CATE uncertainty. More closely related are information-theoretic approaches from Bayesian active learning. These methods are parameter-focused, but differ in their precise objective. Causal-EIG (Fawkes et al., 2025), for instance, directly targets the information gain about the CATE-specific parameters ($\theta_\tau$). Causal-BALD (Jesson et al., 2021) takes a different approach, targeting the information a specific causal prediction (e.g., $\tau(\boldsymbol{x})$) provides about the full set of model parameters ($\theta$). While both are advanced causal-aware criteria, they remain focused on model-internal proxies rather than the final predictive estimand itself. Our work bridges this final gap by introducing the Causal-EPIG framework, a prediction-focused approach based on EPIG (Smith et al., 2023). It directly targets the expected information gain about the causal estimand, ensuring maximal alignment between the acquisition process and the end goal. We provide a more comprehensive review of related literature, including the distinct lines of work on active experimental design and transductive active learning, in App. B.

### B.2  INDUCTIVE AND TRANSDUCTIVE GOALS IN ACTIVE LEARNING

Active Learning (AL) is typically framed by two distinct objectives: inductive and transductive learning. The classic inductive goal, mirroring standard supervised learning, is to train a model that generalizes to unseen data. Most prior AL research has followed this inductive tradition, which fundamentally relies on the assumption that data is independent and identically distributed (IID) (Settles, 2009). In contrast, the transductive goal is to optimize performance on a specific, known set of unlabeled target instances. Pool-based AL exhibits a fascinating duality here. Its mechanism is inherently transductive, as acquisition functions leverage the entire unlabeled pool to make decisions. However, its ultimate goal is usually inductive: to use the pool as a resource to build a generalizable model. However, a critical challenge arises when the distribution of the sampling pool ($p_{\text{pool}}$) differs from the target population's distribution ($p_{\text{tar}}$), a problem known as distribution shift. In this more challenging setting, the transductive selection mechanism must be explicitly directed to serve an inductive goal on the out-of-distribution target set. Recent work has begun to develop such target-aware strategies (MacKay, 1992; Hübotter et al., 2024; Smith et al., 2023), providing a foundation upon which our causally-aligned framework is built.

### B.2.1  WHAT IS THE CONNECTION BETWEEN ACTIVE CATE ESTIMATION AND TAL?

Active CATE estimation can be understood as a unique and compelling instance of transductive learning, which we term **structural transduction**. This perspective clarifies why acquisition functions should be defined with respect to a specific target population, even in the absence of covariate distribution shift, saying $p_{\text{pool}}(\boldsymbol{x}) = p_{\text{tar}}(\boldsymbol{x})$. In TAL (Hübotter et al., 2024), the objective is to infer labels for a pre-defined, fixed set of unlabeled points, $\mathcal{A}$. The learner actively selects queries from a sampling pool, $\mathcal{S}$ (where $\mathcal{S}$ is not necessarily a subset of $\mathcal{A}$), to maximize accuracy specifically on the set $\mathcal{A}$. The key idea is that knowledge of the full set $\mathcal{A}$ from the outset can guide a more efficient

querying strategy than a purely inductive approach, which aims to learn a model that generalizes to the entire data distribution. At first glance, the connection to active CATE estimation is straightforward: the target population, $X_{\text{tar}}$, for which we want to estimate the CATE, is analogous to the unlabeled set $\mathcal{A}$. However, a subtle distinction arises that complicates this analogy. One might argue that if the covariate distributions of the sampling pool and the target set are identical ($p_{\text{pool}}(\boldsymbol{x}) = p_{\text{tar}}(\boldsymbol{x})$, and $X_{\text{pool}} = X_{\text{tar}}$), the task is simply to learn the function $\tau(\boldsymbol{x})$ inductively. The resolution lies in recognizing that the transductive nature of active CATE estimation is not primarily distributional, but **structural**. This stems from a fundamental gap between the data we can observe and the quantity we aim to estimate:

- **The Observation Space.** Through experiments, we can only ever observe individual *factual* outcomes. A single query at $(\boldsymbol{x}_i, t_i)$ yields a noisy observation of one point on the response surface, $f(\boldsymbol{x}_i, t_i)$.
- **The Target Inferential Space.** Our ultimate goal is to infer the CATE, $\tau(\boldsymbol{x}_i) = f(\boldsymbol{x}_i, 1) - f(\boldsymbol{x}_i, 0)$, for every individual $\boldsymbol{x}_i \in X_{\text{tar}}$. This requires knowledge of a *pair* of potential outcomes, $(f(\boldsymbol{x}_i, 0), f(\boldsymbol{x}_i, 1))$, for each individual. This paired set is our true, albeit unobservable, target.

While the positivity assumption guarantees that information about both $f(\boldsymbol{x}, 0)$ and $f(\boldsymbol{x}, 1)$ exists within the sampling pool for any $\boldsymbol{x}$ in the population, it does not resolve the core challenge: *any single observation only reveals one of the two quantities required for an individual's CATE. The essence of **structural transduction**, therefore, is the process of inferring the complete, paired set of potential outcomes for the entire target population, $\{(f(\boldsymbol{x}_i, 0), f(\boldsymbol{x}_i, 1))\}_{\boldsymbol{x}_i \in X_{\text{tar}}}$, from a sequence of sparse, unpaired factual observations.*

Then, let us discuss the more challenging and realistic setting where the sampling pool and target populations differ ($p_{\text{pool}}(\boldsymbol{x}) \neq p_{\text{tar}}(\boldsymbol{x})$). Our central argument for **structural transduction** remains fully intact, as the fundamental mismatch between observing single factual outcomes and inferring paired potential outcomes is a structural property of the CATE estimand, independent of the data distribution. However, this distribution shift introduces a second, more conventional reason for the problem's transductive nature. Even if one were to focus solely on learning the function $\tau(\boldsymbol{x})$, the task is no longer simply inductive. The goal becomes optimizing the estimate of $\tau(\boldsymbol{x})$ specifically for the *known, fixed target set* $X_{\text{tar}}$, using data from a different distribution $p_{\text{pool}}(\boldsymbol{x})$. To bridge this gap efficiently, the acquisition strategy must leverage knowledge of the target set's features, for instance, to up-weight the importance of acquiring samples in regions of high target density. This act of tailoring the learning process to a specific target set is the very definition of transduction. Thus, under distribution shift, active CATE estimation is transductive for a twofold reason: it is **structurally** transductive due to the nature of the causal estimand, and **distributionally** transductive due to the target-aware objective.

Therefore, even in the absence of covariate shift, active CATE estimation task remains transductive. Knowledge of the full target set $X_{\text{tar}}$ is essential because the utility of any candidate query must be evaluated based on how it facilitates this complex inferential leap from the observable to the unobservable causal estimand for the specific population of interest. This perspective provides the foundational justification for our Causal Objective Alignment perspective in Sec. 3 and the Causal-EPIG framework in Sec. 4, which explicitly operationalizes this transductive objective.

### B.3 ADAPTIVE EXPERIMENTAL DESIGN

A significant body of work in active causal learning/inference focuses on active/adaptive experimental design, where the primary goal is to optimize the treatment assignment policy itself and also target at minimizing the predictive performance. (1) One major research line involves adaptive sampling/randomization, where treatment probabilities are updated based on accumulating data to minimize the variance of an estimator like the ATE. This area is built on firm theoretical foundations (van der Laan, 2008; Hahn et al., 2011), with recent works proposing refined designs that use online estimates of nuisance components and exploit martingale structures for valid inference (Kato et al., 2021; Tabord-Meehan, 2023), alongside specialized estimators like A2IPW (Kato et al., 2020) tailored for such adaptive data (Cook et al., 2024). (2) A complementary line of work considers design choices for a fixed, finite pool of individuals. This research ranges from foundational analyses of the tradeoff between covariate balance and robustness (Efron, 1971) to modern active sampling frameworks with finite-sample guarantees, such as those based on leverage score sampling (Addanki et al., 2022;

Ghadiri et al., 2023) or the Gram-Schmidt Walk (Harshaw et al., 2024). (3)A related line of research approaches CATE estimation from the perspective of Bayesian experimental design. Within this domain, recent advances have focused on incorporating real-world complexities. For instance, some methods integrate regulatory constraints (Klein et al., 2025) or structural uncertainty from causal discovery (Toth et al., 2022) into the design process. Others have developed GP-based acquisition functions to minimize the posterior variance of the CATE estimator (Cha & Lee, 2025), or provided finite-sample theoretical guarantees for their estimators in settings like social networks (Zhang et al., 2025).

Our work addresses a fundamentally different scenario. While active experimental design asks, *Who should we treat?*, our setting of active outcome acquisition for observational data asks, *Whose outcome should we measure?* This is critical in domains like healthcare where treatments are already assigned due to ethical or practical constraints, but the resources for acquiring costly outcomes (e.g., biopsies, genetic sequencing) are scarce. The challenge shifts from designing interventions to efficiently allocating measurement resources. Although the action spaces differ, both fields share the goal of allocating a limited resource to reduce causal uncertainty. This suggests that our core principle of a target-aware strategy could inform future work in adaptive experimental design, pointing to a promising direction for bridging these two research areas.

## C   FURTHER PRELIMINARIES

This section provides supplementary material to support the main text. We begin by presenting a comprehensive table of notations used throughout the paper for easy reference. Following this, we review fundamental concepts from information theory that form the theoretical basis for our proposed acquisition function, Causal-EPIG.

### C.1   NOTATIONS

Tab. 2 provides a consolidated summary of the key mathematical notations used in this work, organized by their conceptual domain.

Table 2: Table of Notations

| Symbol | Description |
| --- | --- |
| **General Mathematical Notations** | |
| $a, \mathrm{a}$ | A scalar value and its corresponding random variable. |
| $\boldsymbol{a}, \mathbf{a}$ | A vector and its corresponding random vector. |
| **Core Causal Inference Variables** | |
| $\mathbf{x}, \mathbf{t}, \mathbf{y}$ | Random variables for covariates, treatment, and outcome. |
| $\boldsymbol{x}, t, y$ | Specific realizations of the covariates, treatment, and outcome. |
| $\mathcal{X}, \{0, 1\}, \mathcal{Y}$ | The domains (support) for covariates, treatment, and outcomes, respectively. |
| $\mathrm{y}(0), \mathrm{y}(1)$ | Potential outcomes under the control ($t = 0$) and treatment ($t = 1$) conditions. |
| $\pi(\boldsymbol{x})$ | The propensity score: the probability of receiving treatment given covariates, $p(\mathbf{t} = 1 \mid \mathbf{x} = \boldsymbol{x})$. |
| **CATE and Evaluation Metrics** | |
| $\tau(\boldsymbol{x})$ | The Conditional Average Treatment Effect (CATE), the primary quantity of interest, defined as $\mathbb{E}[\mathrm{y}(1) - \mathrm{y}(0) \mid \mathbf{x} = \boldsymbol{x}]$. |
| $\hat{\tau}(\boldsymbol{x})$ | The estimated CATE function produced by a model. |
| $\sqrt{\epsilon_{\mathrm{PEHE}}}$ | Root PEHE at the population level, i.e., the square root of the mean integrated squared error between the true and estimated CATE. |
| $\sqrt{\hat{\epsilon}_{\mathrm{PEHE}}}$ | Empirical root PEHE, i.e., the square root of the mean squared error over a finite evaluation set $\boldsymbol{X}_{\mathrm{tar}}$. Sometimes, we use $\sqrt{\mathrm{PEHE}}$ for short. |
| $p_{\mathrm{pool}}(\boldsymbol{x})$ | The probability distribution of covariates for the target population of interest. |
| $p_{\mathrm{tar}}(\boldsymbol{x})$ | The probability distribution of covariates for the target population of interest. |

| Symbol | Description |
|---|---|
| **Active Learning Setting** | |
| $D_P$ | The unlabeled pool of instances available for querying. |
| $D_T$ | The labeled training set, which is iteratively augmented with new data. |
| $n_P, n_T$ | The number of instances in the pool ($D_P$) and training set ($D_T$), respectively. |
| $\boldsymbol{X}_P, \boldsymbol{X}_T$ | The sets of covariates (features) in the pool and training datasets, respectively. |
| $\boldsymbol{X}_{\text{tar}}$ | A representative set of samples from the target distribution, used for evaluating the PEHE. |
| $n_b$ | The batch size: the number of instances selected from the pool in each acquisition step. |
| $n_B$ | The total budget for labeling, representing the maximum size of $D_T$. |
| **Acquisition Function and Optimization** | |
| $U(\cdot)$ | The utility function (or acquisition function) that scores candidate data points for labeling. |
| $(\boldsymbol{X}_b, \boldsymbol{t}_b)$ | The optimal batch of instances chosen by maximizing the utility function $U(\cdot)$. |
| $\theta$ | A general representation of model parameters. |
| $\theta_\tau$ | The specific subset of model parameters that define the CATE function, $\tau(\boldsymbol{x})$. |
| $\boldsymbol{x}^*$ | A random covariate vector drawn from the target distribution $p_{\text{tar}}(\boldsymbol{x})$, representing a target location for CATE estimation. |

## C.2 INFORMATION THEORY PRELIMINARIES

We then briefly reviews the information-theoretic concepts used in our acquisition functions.

**Entropy and Mutual Information.** The differential entropy of a continuous random variable $\mathbf{a}$ with probability density function (PDF) $p_{\mathbf{a}}(\boldsymbol{a})$ measures its uncertainty:

$$\mathrm{H}(\mathbf{a}) = -\int_{\mathcal{A}} p_{\mathbf{a}}(\boldsymbol{a}) \log p_{\mathbf{a}}(\boldsymbol{a}) \, d\boldsymbol{a}. \tag{12}$$

The mutual information, $\mathrm{I}(\mathbf{a}; \mathbf{b})$, quantifies the reduction in uncertainty about $\mathbf{a}$ that results from observing another random variable $\mathbf{b}$. It is defined as the difference between the marginal and conditional entropies:

$$\mathrm{I}(\mathbf{a}; \mathbf{b}) = \mathrm{H}(\mathbf{a}) - \mathrm{H}(\mathbf{a} \mid \mathbf{b}). \tag{13}$$

In active learning, this quantity provides a principled measure of the expected information gain from a new observation.

**The Multivariate Gaussian Case.** These concepts admit closed-form expressions for the multivariate Gaussian distribution, which is central to many Bayesian models. For a random vector $\mathbf{a} \in \mathbb{R}^d$ following a multivariate normal distribution $\mathcal{N}(\boldsymbol{\mu}, \boldsymbol{\Sigma})$, the differential entropy is determined by the determinant of its covariance matrix, $|\boldsymbol{\Sigma}|$:

$$\mathrm{H}(\mathbf{a}) = \tfrac{1}{2} \log \left( (2\pi e)^d |\boldsymbol{\Sigma}| \right). \tag{14}$$

Furthermore, for two jointly Gaussian random vectors $(\mathbf{a}, \mathbf{b})$ with a joint distribution, the mutual information has the analytical form:

$$\mathrm{I}(\mathbf{a}; \mathbf{b}) = \tfrac{1}{2} \log \left( \frac{|\boldsymbol{\Sigma}_{aa}| |\boldsymbol{\Sigma}_{bb}|}{|\boldsymbol{\Sigma}|} \right), \tag{15}$$

where $|\boldsymbol{\Sigma}_{aa}|$, $|\boldsymbol{\Sigma}_{bb}|$, and $|\boldsymbol{\Sigma}|$ are the determinants of the marginal and joint covariance matrices, respectively. This closed-form solution is crucial for the efficient computation of information gain in models like Gaussian Processes.

# D  DATASETS

Our evaluation of Causal-EPIG is conducted on four datasets: two fully synthetic benchmarks and two semi-synthetic benchmarks derived from the real-world covariates of the IHDP and ACTG-175 studies. While the fully synthetic settings provide controlled environments, the semi-synthetic datasets introduce the complex covariate distributions characteristic of real-world applications. It is important to note that for all datasets, the data-generating process for outcomes and treatments is known. Consequently, the ground-truth CATE can be precisely calculated, allowing for accurate performance assessment across all settings. To further test for generalization and robustness, we also evaluate on the IHDP, Hahn, and CausalBALD datasets under a covariate shift scenario.

**Experimental Protocol**    Before detailing the specific data-generating processes, we outline the standardized experimental protocol applied to all synthetic benchmarks. For each, we generate a pool set ($D_P$) of 2000 instances, a validation set of 200 instances for model tuning, and a separate test set of 2000 instances. To rigorously evaluate the acquisition functions, we conduct experiments under two distinct scenarios designed to probe different learning properties:

- **Standard (IID) Setting:** This scenario assesses the classic *inductive* learning objective, where the goal is to learn a general model of the underlying data distribution. During active learning, the acquisition function's target set is the pool itself ($\boldsymbol{X}_{\text{tar}} = \boldsymbol{X}_P$). We evaluate the final model's performance on both the pool set (to measure in-distribution learning) and the held-out test set. The performance on the test set is critical as it validates the generalization capability of the strategy.
- **Distribution Shift Setting:** This scenario is designed to assess the *transductive* property of an acquisition function, its ability to strategically select data from a source distribution to optimize performance on a specific, known target distribution. Here, the target set for the acquisition function is explicitly set to the test set ($\boldsymbol{X}_{\text{tar}} = \boldsymbol{X}_{\text{test}}$). While we report performance on both the pool and test sets, the primary metric is the performance on the test set, as it directly measures how effectively the acquisition function handles the distribution shift.

## D.1  CAUSALBALD SYNTHETIC DATASET

We first use a fully synthetic dataset adapted from the simulation in the CausalBALD paper (Jesson et al., 2021), which is adapted from Kallus et al. (2019), which allows for precise evaluation against a known ground truth.

**Standard Setting.**    In the standard (no-shift) scenario, the data-generating process is defined as follows. The one-dimensional covariate $\boldsymbol{x}$ is drawn from a standard normal distribution, $\boldsymbol{x} \sim \mathcal{N}(0, 1)$. The treatment assignment t is a random variable drawn from a Bernoulli distribution, where the probability of receiving treatment (t = 1) is given by the propensity score $\pi(\boldsymbol{x})$:

$$\text{t} \mid \boldsymbol{x} \sim \text{Bern}(\pi(\boldsymbol{x})), \quad \text{where} \quad \pi(\boldsymbol{x}) = \text{sigmoid}(2\boldsymbol{x} + 0.5). \tag{16}$$

The observed outcome y is then generated based on $\boldsymbol{x}$ and t with additive standard normal noise, $\epsilon \sim \mathcal{N}(0, 1)$. This process implicitly defines the mean potential outcome functions:

$$\begin{aligned} \mu_0(\boldsymbol{x}) &= 1 + 2\sin(2\boldsymbol{x}), \\ \mu_1(\boldsymbol{x}) &= 2\boldsymbol{x} + 3 - 2\sin(2\boldsymbol{x}). \end{aligned} \tag{17}$$

This results in the true CATE function: $\tau(\boldsymbol{x}) = \mu_1(\boldsymbol{x}) - \mu_0(\boldsymbol{x}) = 2\boldsymbol{x} + 2 - 4\sin(2\boldsymbol{x})$.

**Covariate Shift Setting.**    To evaluate model robustness, we introduce a covariate shift scenario. In this setting, the training and pool data are generated exactly as described above, with covariates drawn from $\mathbf{x} \sim \mathcal{N}(0, 1)$. However, the testing set or the target set, used for evaluation, is drawn from a different distribution where the covariate follows a uniform distribution, $\mathbf{x}_{\text{test}} \sim \mathcal{U}(0.2, 0.5)$. The underlying potential outcome functions and the CATE function remain unchanged across both settings, isolating the effect of the covariate shift.

## D.2  HAHN SYNTHETIC DATASET

Our second synthetic dataset is based on the simulation design from (Hahn et al., 2020), featuring a five-dimensional covariate vector $\mathbf{x} \in \mathbb{R}^5$.

**Standard Setting.** The covariates are generated as follows: three continuous variables from a standard normal distribution ($\mathbf{x}_1, \mathbf{x}_2, \mathbf{x}_3 \sim \mathcal{N}(0, 1)$), one binary variable from a Bernoulli distribution ($\mathbf{x}_4 \sim \text{Bernoulli}(0.5)$), and one categorical variable from a uniform distribution over three levels ($\mathbf{x}_5 \sim \mathcal{U}\{1, 2, 3\}$). Following the original paper, we use the "nonlinear" prognostic function and the "heterogeneous" treatment effect function. The prognostic score $\mu(\boldsymbol{x})$ is defined as:

$$\mu(\boldsymbol{x}) = -6 + g(\boldsymbol{x}_5) + 6|\boldsymbol{x}_3 - 1|, \tag{18}$$

where $g(\cdot)$ is a helper function mapping the categorical covariate to a scalar offset: $g(1) = 2$, $g(2) = -1$, and $g(3) = -4$. The true CATE function $\tau(\boldsymbol{x})$ is defined by an interaction term:

$$\tau(\boldsymbol{x}) = 1 + 2\boldsymbol{x}_2\boldsymbol{x}_4. \tag{19}$$

We construct the propensity score $\pi(\boldsymbol{x})$ with an intentional deviation from the original design in (Hahn et al., 2020) to create a more challenging evaluation scenario. Our formulation utilizes the non-monotonic Gaussian PDF instead of the original's CDF, and models the influence of the covariate $\boldsymbol{x}_1$ as an external additive term. This modification induces a more complex relationship between covariates and treatment assignment, providing a more rigorous test of the active learning strategies under evaluation. To define the score, the prognostic score is first scaled as $\tilde{\mu}(\boldsymbol{x}) = 3\mu(\boldsymbol{x})/\sigma_\mu$, where $\sigma_\mu$ is the standard deviation of $\mu(\boldsymbol{x})$ across the population. The propensity score is then defined as:

$$\pi(\boldsymbol{x}) = 0.8 \cdot \phi(\tilde{\mu}(\boldsymbol{x})) - 0.5\boldsymbol{x}_1 + \xi, \tag{20}$$

where $\phi(\cdot)$ denotes the standard normal probability density function and $\xi \sim \mathcal{U}(0.05, 0.15)$ is a random noise term. Treatment is assigned via $t \sim \text{Bernoulli}(\pi(\boldsymbol{x}))$. The final observed outcome $y$ is generated by adding Gaussian noise to the expected outcome, $y = \mu(\boldsymbol{x}) + t \cdot \tau(\boldsymbol{x}) + \epsilon$, where the noise is scaled to achieve a signal-to-noise ratio of 3.

**Covariate Shift Setting.** For the corresponding covariate shift scenario, the training and pool data are generated as above. For the test set, however, the three continuous covariates are drawn from a uniform distribution, $\mathbf{x}_1, \mathbf{x}_2, \mathbf{x}_3 \sim \mathcal{U}(0.2, 0.5)$, instead of a standard normal. The distributions of the discrete covariates ($\mathbf{x}_4, \mathbf{x}_5$) and the underlying functional forms for $\mu(\boldsymbol{x})$ and $\tau(\boldsymbol{x})$ remain the same.

### D.3 IHDP SEMI-SYNTHETIC DATASET

We use the well-known Infant Health and Development Program (IHDP) dataset within the semi-synthetic framework of (Hill, 2011). This setup uses real-world covariates from 747 subjects (139 treated, 608 control), comprising 6 continuous and 19 binary variables, but simulates the outcomes to provide a known ground truth. The 747 subjects are split into a training/pool set of 523 and a test set of 224. All continuous covariates are standardized.

**Standard Setting.** In the standard scenario, a sparse coefficient vector $\beta_B$ is generated by sampling each element from the set $\{0.0, 0.1, 0.2, 0.3, 0.4\}$ with probabilities $\{0.6, 0.1, 0.1, 0.1, 0.1\}$, respectively. The mean potential outcomes are then generated as:

$$\begin{aligned} \mu_0(\boldsymbol{x}) &= \exp((\boldsymbol{x} + 0.5)\beta_B), \\ \mu_1(\boldsymbol{x}) &= (\boldsymbol{x} + 0.5)\beta_B - \omega_B, \end{aligned} \tag{21}$$

where the offset $\omega_B$ is calculated to fix the true Average Treatment Effect on the Treated (ATT) to 4. Potential outcomes are formed by adding standard normal noise, $y_0(\boldsymbol{x}) = \mu_0(\boldsymbol{x}) + \epsilon$ and $y_1(\boldsymbol{x}) = \mu_1(\boldsymbol{x}) + \epsilon$, with $\epsilon \sim \mathcal{N}(0, 1)$. The final observed outcome is $y = (1 - t) \cdot y_0(\boldsymbol{x}) + t \cdot y_1(\boldsymbol{x})$.

**Covariate Shift Setting.** For the covariate shift scenario, the training data is generated as described above. On the test set, however, the first two continuous covariates (birth weight and head circumference) are resampled from a uniform distribution, $\mathcal{U}(0, 0.5)$. Furthermore, the outcome-generating mechanism is altered. The coefficient vector $\beta_B$ is sampled as before, but the first two coefficients (corresponding to the shifted covariates) are set to zero. The mean potential outcomes are then redefined as:

$$\begin{aligned} \mu_0(\boldsymbol{x}) &= \exp((\boldsymbol{x} + 0.5)\beta_B), \\ \mu_1(\boldsymbol{x}) &= \exp((\boldsymbol{x} + 0.5)\beta_B) + 3 \cdot \boldsymbol{x}_{\text{bw}} \cdot \boldsymbol{x}_{\text{b.head}}. \end{aligned} \tag{22}$$

This induces a new ground-truth CATE, $\tau(\boldsymbol{x}) = 3 \cdot \boldsymbol{x}_{\text{bw}} \cdot \boldsymbol{x}_{\text{b.head}}$, creating a challenging scenario where the model must generalize to both a different covariate distribution and a new functional form for the treatment effect.

### D.4 ACTG-175 Semi-Synthetic Dataset

Our final semi-synthetic exercise uses the AIDS Clinical Trials Group Study 175 (ACTG-175) dataset (Hammer et al., 1996). The original data comes from a randomized trial, from which an observational study is recreated by removing a non-random subset of patients, specifically, those not showing symptomatic HIV infection. The resulting dataset consists of 813 subjects and 12 covariates (3 continuous and 9 binary), as described in Tab. 3. The design is slightly unbalanced, with 281 individuals in the treated group and 532 in the control. The dataset is partitioned into a training/pool set (70%, 569 subjects) and a test set (30%, 244 subjects). The continuous covariates are standardized, and outcomes are simulated using a process with non-linearities and interactions.

Table 3: Description of Covariates from the ACTG-175 Dataset.

| Variable | Description |
| --- | --- |
| age | Numeric: age in years |
| wtkg | Numeric: weight in kilograms |
| hemo | Binary: history of haemophilia (1 = yes) |
| homo | Binary: homosexual activity (1 = yes) |
| drugs | Binary: history of intravenous drug use (1 = yes) |
| oprior | Binary: non-zidovudine antiretroviral therapy prior to study (1 = yes) |
| z30 | Binary: zidovudine use in the 30 days prior to study (1 = yes) |
| preanti | Numeric: number of days of prior antiretroviral therapy |
| race | Binary: race (0 = White, 1 = non-white) |
| gender | Binary: gender (0 = female, 1 = male) |
| str2 | Binary: antiretroviral history (0 = naive, 1 = experienced) |
| karnof_hi | Binary: Karnofsky score (0 = score < 100, 1 = score = 100) |

The prognostic score $\mu(\boldsymbol{x})$ and the CATE function $\tau(\boldsymbol{x})$ are defined as:

$$\mu(\boldsymbol{x}) = 6 + 0.3\boldsymbol{x}_{\text{wtkg}}^2 - \sin(\boldsymbol{x}_{\text{age}}) \cdot (\boldsymbol{x}_{\text{gender}} + 1) + 0.6\boldsymbol{x}_{\text{hemo}} \cdot \boldsymbol{x}_{\text{race}} - 0.2\boldsymbol{x}_{z30},$$
$$\tau(\boldsymbol{x}) = 1 + 1.5\sin(\boldsymbol{x}_{\text{wtkg}}) \cdot (\boldsymbol{x}_{\text{karnof\_hi}} + 1) + 2\boldsymbol{x}_{\text{age}}. \tag{23}$$

The mean potential outcomes are constructed as $\mu_0(\boldsymbol{x}) = \mu(\boldsymbol{x})$ and $\mu_1(\boldsymbol{x}) = \mu(\boldsymbol{x}) + \tau(\boldsymbol{x})$. The potential outcomes are then formed by adding Gaussian noise, $y_0(\boldsymbol{x}) = \mu_0(\boldsymbol{x}) + \epsilon$ and $y_1(\boldsymbol{x}) = \mu_1(\boldsymbol{x}) + \epsilon$. The observed outcome is $y = (1 - t) \cdot y_0(\boldsymbol{x}) + t \cdot y_1(\boldsymbol{x})$, where the noise $\epsilon$ is drawn from $\mathcal{N}(0, \sigma_y^2)$ with the standard deviation $\sigma_y$ set to one-eighth of the prognostic score's range, i.e., $\sigma_y = (\max(\mu) - \min(\mu))/8$.

### D.5 AL Process Datasets Setup

Across all datasets, we follow a consistent experimental protocol to ensure fair comparisons. To account for randomness in data splits and model initialization, all results are averaged over 10 independent trials. The active learning process for each trial begins with a *warm-start phase*, where an initial labeled training set $D_T$ is created by randomly selecting 50 instances from the unlabeled pool $D_P$. Following this, the iterative acquisition process begins. In each step, the acquisition function selects a new batch of instances from the remaining pool to be labeled and added to $D_T$. The parameters for this process vary by dataset. For the synthetic datasets (Hahn and CausalBALD), we perform 40 acquisition steps with a batch size of 20 (800 total acquisitions). For the IHDP dataset, we perform 40 steps with a batch size of 10 (400 total acquisitions). Finally, for the ACTG-175 dataset, we perform 20 steps with a batch size of 15 (300 total acquisitions). This process results in final training sets of size 850 (Hahn, CausalBALD), 450 (IHDP), and 350 (ACTG-175), respectively.

## E Model Details

In this section, we provide implementation details for the models and methods used in our study. We begin by presenting the overarching algorithm for the active CATE estimation loop in Alg. 1. The subsequent subsections delve into the components of this algorithm, first describing our three primary Bayesian CATE estimators: BCF (Hahn et al., 2020), CMGP (Alaa & Van Der Schaar,

2017), and NSGP (Alaa & Schaar, 2018). We also briefly discuss other models used for ablation studies. Finally, we detail the acquisition functions evaluated within this framework, including Random, Causal-BALD (Jesson et al., 2021), Coreset (Qin et al., 2021), EPIG (Smith et al., 2023), Causal-EIG (Fawkes et al., 2025), and our proposed Causal-EPIG-$\mu$ and Causal-EPIG-$\tau$.

### E.1  ACTIVE CATE ESTIMATION LOOP

Here, we formalize the pipeline for active CATE estimation used throughout our experiments. The procedure, detailed in Alg. 1, outlines a general batch acquisition strategy for improving a CATE estimator, $\hat{\tau}(\cdot)$. The pipeline begins with a random warm-start, followed by an iterative loop: the acquisition function scores candidates from the pool based on their expected utility for CATE estimation, a batch of the most informative points is acquired, and the CATE model is retrained on the newly augmented dataset.

---

**Algorithm 1** Budgeted Batch Active Learning for CATE Estimation

---

**Require:** Unlabeled pool $D_P$, Target set $\boldsymbol{X}_{\text{tar}}$, Utility function $U$, Batch size $n_b$, Max budget $n_B$.
**Ensure:** Final labeled set $D_T$ and final CATE estimator $\hat{\tau}(\cdot)$.

 1: Initialize labeled set $D_T \leftarrow \varnothing$.
      // – Warm-start Phase –
 2: Select an initial random batch $D_{\text{init}} \subset D_P$ of size $n_b$.
 3: Query factual outcomes for all $(\boldsymbol{x}, t) \in D_{\text{init}}$.
 4: Update $D_T \leftarrow D_T \cup D_{\text{init}}$ and $D_P \leftarrow D_P \smallsetminus D_{\text{init}}$.
 5: Train initial CATE estimator $\hat{\tau}(\cdot)$ on $D_T$.
      // – Main Active Learning Loop –
 6: **while** $|D_T| < n_B$ **and** $D_P \neq \varnothing$ **do**
 7:      Compute utility scores for all candidates in the pool:
 8:      $S \leftarrow \{U(\boldsymbol{x}_i, t_i \mid D_T, \boldsymbol{X}_{\text{tar}})$ for each $(\boldsymbol{x}_i, t_i) \in D_P\}$.
 9:      Select batch $D_b$ corresponding to the $n_b$ highest scores in $S$.
10:      Query factual outcomes for all $(\boldsymbol{x}, t) \in D_b$.
11:      Update $D_T \leftarrow D_T \cup D_b$ and $D_P \leftarrow D_P \smallsetminus D_b$.
12:      Retrain or update estimator $\hat{\tau}(\cdot)$ on the new $D_T$.
13: **end while**
14: **return** $D_T, \hat{\tau}(\cdot)$

---

**Batch Acquisition Strategy**   The procedure outlined in Alg. 1 involves acquiring a batch of $n_b$ new outcomes in each round of active learning. The simplest method for this is to score all candidates in the pool, rank them by their utility, and select the top-$n_b$ points. However, this approach can lead to selecting a batch with redundant information. More sophisticated methods, such as the greedy selection strategy proposed in BatchBALD (Kirsch et al., 2019), aim to select a diverse batch by accounting for information overlap, but this comes at a significant computational cost.

To balance performance and efficiency, a practical approximation was introduced in prior work (Kirsch et al., 2023) and subsequently used by CausalBALD (Jesson et al., 2021). This strategy, sometimes referred to as *softmax-BALD*, re-normalizes the utility scores of all candidates using a softmax function before selecting the top-$n_b$ points. This was shown to approximate the performance of the more expensive greedy methods while remaining computationally fast.

For the baselines adapted from CausalBALD, we adhere to the established practice of using a softmax-based stochastic acquisition. However, for our proposed Causal-EPIG methods, we empirically found that this strategy did not yield a discernible performance advantage over a simpler top-$n_b$ approach. This is demonstrated in Fig. 5, where the zero-temperature setting ($T = 0$), which is equivalent to a deterministic top-$n_b$ selection, performs on par with tempered stochastic selections. Therefore, to maximize computational efficiency without compromising performance, we adopt the direct top-$n_b$ selection strategy for all Causal-EPIG variants.

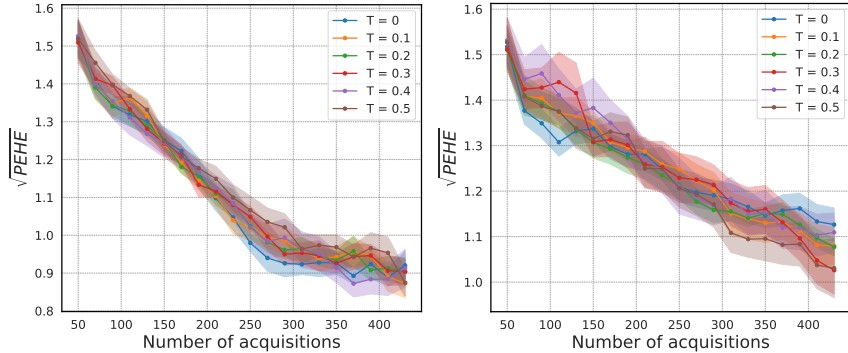

Figure 5: Ablation study of the temperature parameter for Causal-EPIG-$\mu$, with performance measured by $\sqrt{PEHE}$. The left panel (Causal-EPIG-$\tau$) serves as a reference, while the right panel (Causal-EPIG-$\mu$) illustrates the effect of varying the temperature.

### E.2 IMPLEMENTATIONS OF DIFFERENT ACQUISITION FUNCTIONS

The core component of the active CATE estimation loop is the acquisition function, which quantifies the utility of each candidate $(\boldsymbol{x}, t)$ in the unlabeled pool $D_P$. This utility score guides the selection of the most informative instances for labeling. In our experiments, we compare our proposed Causal-EPIG strategies against several well-established baseline methods. This subsection provides the implementation details for each of these acquisition functions. For all methods, batch acquisition is performed by selecting the $n_b$ candidates with the highest utility scores.

**Random Acquisition.** This is the simplest baseline, involving no active selection strategy. At each acquisition step, a batch of $n_b$ candidates is selected uniformly at random from the remaining unlabeled pool $D_P$. The utility score for every candidate can be considered a random variable drawn from a uniform distribution, $U(\boldsymbol{x}, t) \sim \mathcal{U}(0, 1)$. This method serves as a lower bound on performance, representing data collection without model guidance.

**Causal-BALD Variants.** We include the full suite of acquisition functions from the Causal-BALD framework (Jesson et al., 2021) as information-theoretic baselines. This framework adapts the standard BALD objective to the causal setting by calculating the expected information gain about the model parameters $\theta$. We benchmark against all variants proposed in the original work:

- $\tau$**-BALD**, which is defined as the mutual information between the $\tau(\boldsymbol{x})$ and the mode parameters $\theta$, saying $I(y(1) - y(0), \theta | D'_T)$.

- $\mu$**-BALD**, which is defined as the mutual information between the corresponding potential outcome and the model parameters, saying $I(y(t), \theta | D_T \cup (\boldsymbol{x}, t))$.

- **Propensity** (Propensity-based), which targets the propensity score function ($\pi$).

- Combined variants, such as $\mu\pi$**-BALD** and $\mu\rho$**-BALD**, that target a weighted sum of the information gain from multiple components.

A fundamental distinction separates our Causal-EPIG framework from the Causal-BALD family. Causal-BALD variants are **parameter-focused**, aiming to reduce uncertainty over the model's internal representation ($\theta$). In contrast, our framework is **prediction-focused**, directly targeting uncertainty about the causal quantities themselves. For instance, while $\tau$-BALD maximizes information gain about the CATE *parameters* ($\theta_\tau$), our Causal-EPIG-$\tau$ maximizes the information a factual observation provides about the CATE *function* ($\tau(\boldsymbol{x}^*)$). Similarly, $\mu$-BALD reduces uncertainty over the potential outcome *parameters*, whereas our Causal-EPIG-$\mu$ reduces predictive uncertainty about the potential outcome *values* themselves ($y^*(t^*)$). For all experiments, we utilize the official implementation provided by the authors[1].

---

[1]https://github.com/OATML/causal-bald

**Sign Ambiguity BALD (Adapted from Sundin et al.).** This baseline is an information-theoretic strategy, inspired by the work of Sundin et al. (Sundin et al., 2019) and the Causal-BALD framework (Jesson et al., 2021), that focuses acquisition on points where the **sign of the CATE** is most ambiguous. The utility is the BALD objective (mutual information) applied to a conceptual Bernoulli variable representing the sign of the effect. For our Bayesian CATE estimators, which yield $K$ posterior samples for the CATE, $\{\tau_k(\boldsymbol{x})\}_{k=1}^K$, we approximate the mutual information via Monte Carlo. First, for each posterior sample $\tau_k(\boldsymbol{x})$, we compute a sign-related probability, using the overall posterior standard deviation $\sigma_\tau(\boldsymbol{x}) = \mathrm{std}(\{\tau_k(\boldsymbol{x})\})$ as a measure of uncertainty:

$$\gamma_k(\boldsymbol{x}) := \Phi\left(-\frac{|\tau_k(\boldsymbol{x})|}{\sigma_\tau(\boldsymbol{x})}\right), \tag{24}$$

where $\Phi(\cdot)$ is the standard normal CDF. The final utility is then the estimated mutual information:

$$\mathrm{Sundin}(\boldsymbol{x}) := \mathrm{H}(\mathrm{Bernoulli}(\bar{\gamma}(\boldsymbol{x}))) - \frac{1}{K}\sum_{k=1}^K \mathrm{H}(\mathrm{Bernoulli}(\gamma_k(\boldsymbol{x}))), \tag{25}$$

where $\bar{\gamma}(\boldsymbol{x})$ is the mean of the $\gamma_k(\boldsymbol{x})$ samples. This score is maximized for candidates where the ensemble of posterior samples is most conflicted about the sign of the CATE.

**Coreset Selection (QHTE).** We implement the coreset-based acquisition strategy from QHTE (Qin et al., 2021)[2]. The core idea of this method is to select a representative subset of data points that "cover" the input space for both the treated and control groups independently. The strategy operates in two stages. First, it partitions the unlabeled pool $D_P$ into a treated pool $D_P^1 = \{(\boldsymbol{x}, t = 1)\}$ and a control pool $D_P^0 = \{(\boldsymbol{x}, t = 0)\}$. Then, it applies the coreset selection algorithm separately within each of these two pools. For each candidate $\boldsymbol{x}$ in a given pool (e.g., $D_P^1$), its utility is defined as its minimum distance to any point already in the corresponding labeled set (e.g., $\boldsymbol{X}_T^1$):

$$\mathrm{QHTE}(\boldsymbol{x}, t) := \min_{\boldsymbol{x}' \in \boldsymbol{X}_T^t} d(\boldsymbol{x}, \boldsymbol{x}'), \quad \text{for } t \in \{0, 1\}. \tag{26}$$

The distance metric $d(\boldsymbol{x}_i, \boldsymbol{x}_j)$ is derived from the posterior covariance of the model's predictions, as available in both GP and BCF models. After calculating these utility scores for all candidates in both pools, the scores are combined, and the top $n_b$ candidates overall are selected for labeling. This two-pronged approach ensures that the selected batch contains representative samples from both treatment arms.

**Causal-EIG.** Causal-EIG is a method originally proposed for the task of prospective causal effect estimation (Fawkes et al., 2025), which aims to evaluate the utility of an entire dataset before it is acquired. We adapt this method for our pool-based active learning setting. The original approach calculates the EIG that a new dataset provides about the causal model's parameters. To apply it to our task, we treat each candidate data point $(\boldsymbol{x}, t)$ as a potential dataset of size one. The resulting utility function is trying to maximize the information gain about the parameters of the CATE function, $\theta_\tau$:

$$\mathrm{Causal\text{-}EIG}(\boldsymbol{x}, t) := \mathrm{I}(y; \theta_\tau \mid \boldsymbol{x}, t, D_T). \tag{27}$$

Following the original paper, we implement this acquisition function using both BCF and CMGP as the base CATE estimators and utilize the official code provided by the authors[3].

**EPIG (Expected Predictive Information Gain).** EPIG (Smith et al., 2023) is an information-theoretic acquisition function that addresses a key limitation of BALD. Instead of focusing on the indirect objective of reducing uncertainty over model parameters ($\theta$), EPIG directly quantifies the expected reduction in predictive uncertainty on other unseen data points. The utility of a candidate point $(\boldsymbol{x}, t)$ is defined as the expected mutual information between its unknown label $y$ and the label $y^*$ of a randomly chosen point $(\boldsymbol{x}^*, t^*)$ from the data distribution:

$$\mathrm{EPIG}(\boldsymbol{x}, t) := \mathbb{E}_{p_{\mathrm{tar}}(\boldsymbol{x}^*, t^*)}\Big[\mathrm{I}(y; y^*(t^*) \mid (\boldsymbol{x}^*, t^*), D_T')\Big]. \tag{28}$$

Intuitively, EPIG prioritizes points that are expected to be most informative about the labels of other points in the dataset.

---

[2]https://github.com/Qcer17/QHTE

[3]https://github.com/LucileTerminassian/causal_prospective_merge

### E.3 BAYESIAN CAUSAL FORESTS

Our first estimator, BCF, leverages tree ensembles with a careful reparameterization and orthogonalization strategy to provide robust CATE estimates (Hahn et al., 2020). To improve computational efficiency, we utilize its accelerated extension, XBCF (Krantsevich et al., 2023). As BCF is built upon Bayesian Additive Regression Trees (BART) (Hill, 2011), we begin with an overview of this foundational method.

#### E.3.1 THE BART FOUNDATION

BART models an unknown function $f(\boldsymbol{x})$ as a sum-of-trees ensemble:

$$f(\boldsymbol{x}) = \sum_{l=1}^{L} g_l(\boldsymbol{x}; T_l, M_l), \tag{29}$$

where each $g_l$ is a regression tree defined by its structure $T_l$ and leaf parameters $M_l = \{\mu_{l1}, \dots, \mu_{lb_l}\}$. To prevent overfitting, BART imposes regularizing priors on the tree structure (favoring shallow trees) and the leaf parameters (shrinking predictions towards zero). Posterior inference is performed via MCMC backfitting, which iteratively samples each tree conditional on the others.

#### E.3.2 BCF FOR CAUSAL INFERENCE

BCF adapts BART to causal inference by modeling the conditional outcome as $\mathbb{E}[y|\boldsymbol{x}, t] = \mu(\boldsymbol{x}) + \tau(\boldsymbol{x})t$, where $\mu(\boldsymbol{x})$ (prognostic function) and $\tau(\boldsymbol{x})$ (CATE function) are independent BART ensembles. We use the accelerated reparameterization from Krantsevich et al. (2023):

$$f_\theta(\boldsymbol{x}, t) = a\, \tilde{\mu}_{\mathrm{bcf}}(\boldsymbol{x}) + b_t\, \tilde{\tau}_{\mathrm{bcf}}(\boldsymbol{x}), \tag{30}$$

where $a, b_t$ are scaling factors and the CATE is given by $(b_1 - b_0)\tilde{\tau}_{\mathrm{bcf}}(\boldsymbol{x})$. A key feature is orthogonalization, where $\tilde{\mu}_{\mathrm{bcf}}$ is fit on the treatment-residualized outcome $y - b_t\tilde{\tau}_{\mathrm{bcf}}(\boldsymbol{x})$, forcing it to capture variation independent of the treatment effect and leading to more robust CATE estimates. The posterior distribution of the CATE is constructed from MCMC samples. For each posterior draw $s$, a sample of the CATE is $\tau^{(s)}(\boldsymbol{x}^*) = (b_1^{(s)} - b_0^{(s)}) \cdot \tilde{\tau}_{\mathrm{bcf}}^{(s)}(\boldsymbol{x}^*)$. While collecting these samples provides the marginal posterior $p(\tau(\boldsymbol{x}^*) \mid D_T)$, information-based acquisition requires the joint predictive posterior $p(y, \tau(\boldsymbol{x}^*) \mid (\boldsymbol{x}, t), \boldsymbol{x}^*, D_T)$. We approximate this as a multivariate Gaussian (Kirsch, 2023; Jesson et al., 2021), estimating its parameters from the $S$ posterior draws. For each draw $s$, we compute the pair $(f^{(s)}, \tau^{(s)})$, where $f^{(s)}$ is the expected outcome. The Gaussian's mean vector is the sample mean of these pairs. Its covariance matrix is the sample covariance of the pairs.

**BCF Posterior Distribution Analysis.** While this approximation is unlikely to hold perfectly in practice, it is crucial to assess its plausibility and understand the nature of any potential violations. Therefore, we investigate the degree to which this assumption holds across our five experimental data-generating processes: CausalBALD, Hahn (linear and nonlinear), IHDP, and ACTG. In the first step, we visualize the posterior of different quantities for all these datasets we used in the paper and the results are shown in Fig. 6, Fig. 7, Fig. 8, Fig. 9, and Fig. 10. Then, as might be expected for a simplifying approximation, the formal statistical tests presented in Tab. 4 reject the null hypothesis of perfect normality for all five datasets at the $\alpha = 0.05$ significance level. However, these tests are more useful in helping us quantify the nature and severity of the deviation. The results show a clear pattern: the semi-synthetic datasets, *ACTG* and *IHDP*, exhibit more modest deviations. They have the lowest Henze-Zirkler statistics and Mardia's kurtosis values (8.407 and 8.868, respectively) that are closest to the theoretical value of 8 for a bivariate normal distribution. In contrast, the synthetic datasets show more pronounced violations, primarily due to heavy tails (leptokurtosis), with *CausalBALD* showing the most significant departure (Mardia's kurtosis of 11.430). In summary, this analysis confirms that while the Gaussian posterior is indeed an approximation, the violations are not uniform across data types. For the more realistic semi-synthetic datasets, the deviations from normality are relatively contained. This suggests that using a multivariate normal approximation is a justifiable and reasonable trade-off for the significant computational tractability it provides, rather than an overly strong assumption that would undermine the method's validity.

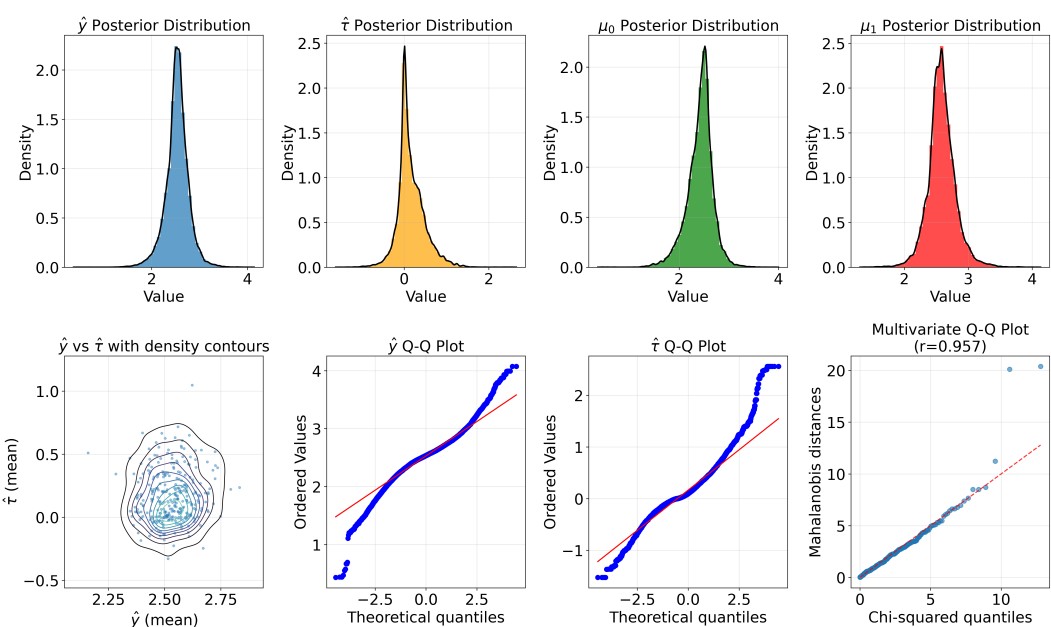

Figure 6: Posterior distribution analysis for the BCF model on the **CausalBALD Dataset**.

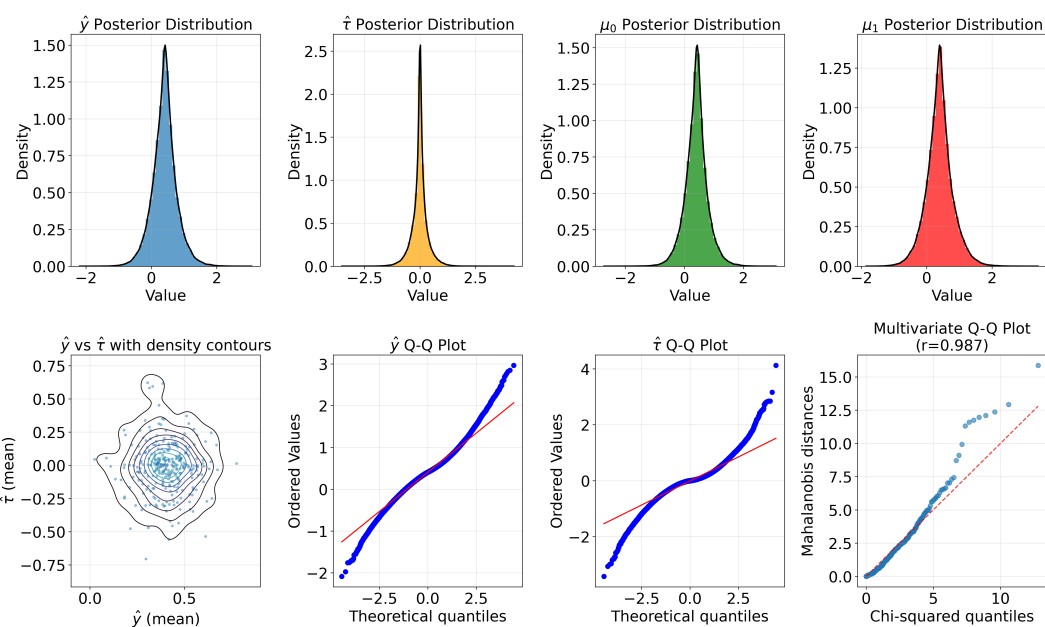

Figure 7: Posterior distribution analysis for the BCF model on the **Hahn linear Dataset**.

Table 4: Multivariate normality tests for the joint posterior of $(\hat{y}, \hat{\tau})$

| Dataset | Q-Q corr | $\chi^2$ GoF p-value | HZ stat | Mardia kurtosis |
|---------|----------|---------------------|---------|-----------------|
| CausalBALD | 0.974 | <1e-10 | 0.210 | 11.430 |
| Hahn (linear) | 0.981 | <1e-10 | 0.189 | 9.655 |
| Hahn (nonlinear) | 0.993 | <1e-10 | 0.194 | 9.878 |
| IHDP | 0.989 | 1.45e-3 | 0.173 | 8.868 |
| ACTG | 0.989 | 6.94e-3 | 0.167 | 8.407 |

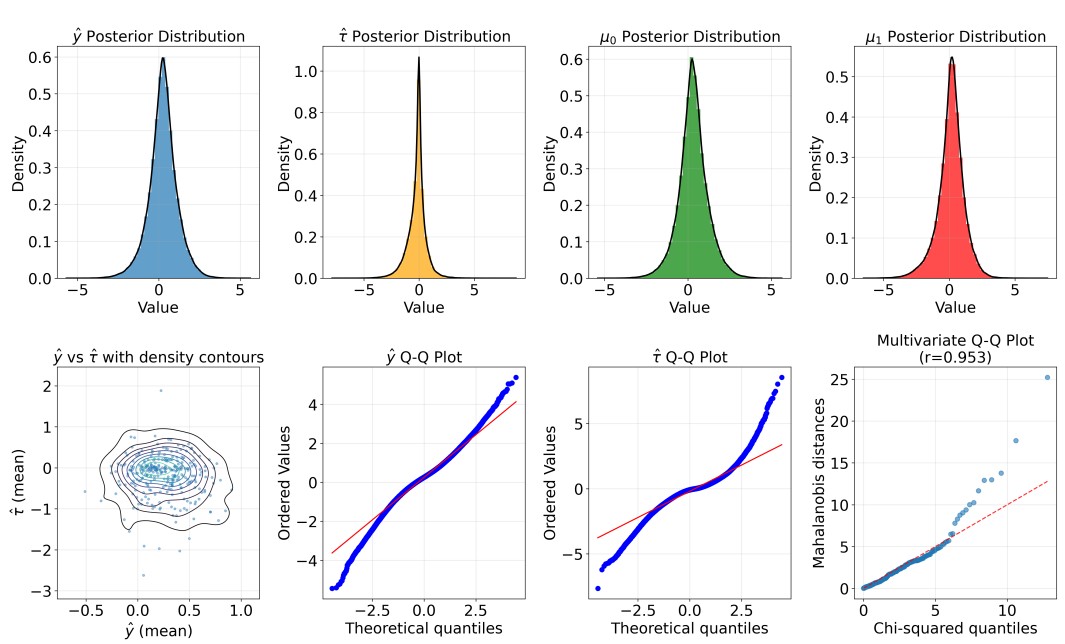

Figure 8: Posterior distribution analysis for the BCF model on the **Hahn non-linear Dataset**.

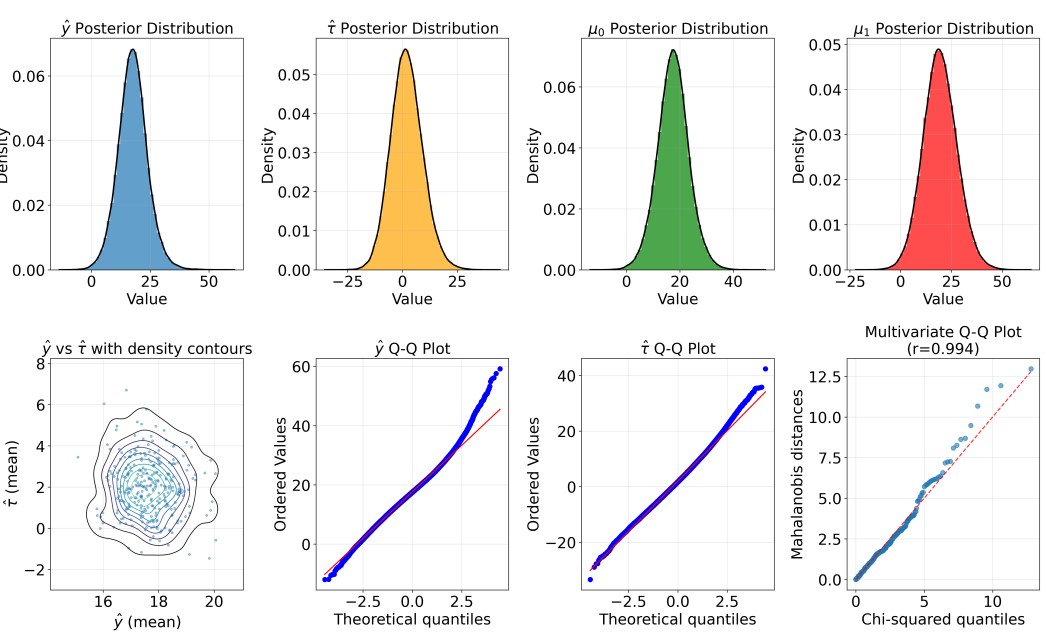

Figure 9: Posterior distribution analysis for the BCF model on the **IHDP Dataset**.

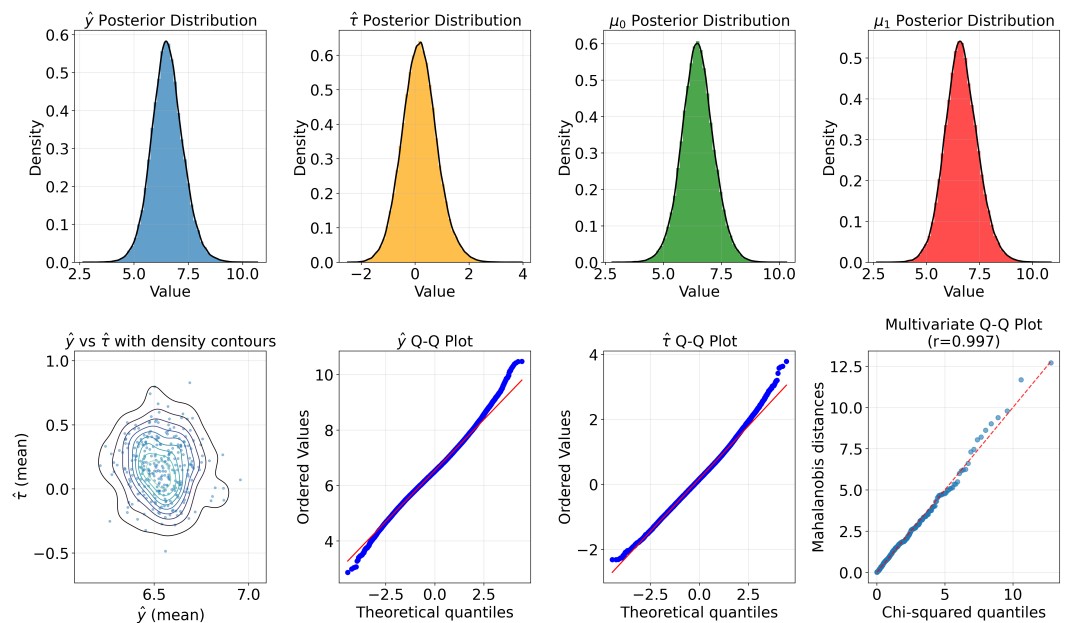

Figure 10: Posterior distribution analysis for the BCF model on the **ACTG Dataset**.

### E.4    GAUSSIAN PROCESS MODELS

Our other two primary estimators are based on Gaussian Processes. GP models are a natural fit for Causal-EPIG because they provide a closed-form, analytic posterior for the CATE, which in turn allows for the highly efficient computation of the acquisition function. We consider two distinct GP formulations.

#### E.4.1    CAUSAL MULTITASK GAUSSIAN PROCESSES (CMGP)

CMGP treats potential outcome estimation as a multitask learning problem, enabling the model to borrow statistical strength across treatment arms (Alaa & Van Der Schaar, 2017). It places a joint GP prior over the vector $[f_0(\boldsymbol{x}), f_1(\boldsymbol{x})]^\top$ using a $2 \times 2$ matrix-valued kernel $\boldsymbol{K}_\eta$, typically constructed via a Linear Model of Coregionalization (LMC). The observed outcomes $y_i$ are noisy realizations of the latent function $f_{t_i}(\boldsymbol{x}_i)$, i.e., $y_i \mid \boldsymbol{x}_i, t_i \sim \mathcal{N}(f_{t_i}(\boldsymbol{x}_i), \sigma_n^2)$.

Given the GP prior and Gaussian likelihood, the posterior over $[f_0(\boldsymbol{x}), f_1(\boldsymbol{x})]$ is also a GP. The posterior for the CATE, $\tau(\boldsymbol{x}_*) = f_1(\boldsymbol{x}_*) - f_0(\boldsymbol{x}_*)$, is therefore also Gaussian, with mean and variance derived analytically from the posterior of the potential outcomes:

$$\hat{\tau}(\boldsymbol{x}_*) \sim \mathcal{N}\big(\boldsymbol{e}^\top \boldsymbol{\mu}_{\text{post}}(\boldsymbol{x}_*),\ \boldsymbol{e}^\top \boldsymbol{\Sigma}_{\text{post}}(\boldsymbol{x}_*, \boldsymbol{x}_*)\boldsymbol{e}\big), \tag{31}$$

where $\boldsymbol{e} = [-1, 1]^\top$, and $\boldsymbol{\mu}_{\text{post}}$ and $\boldsymbol{\Sigma}_{\text{post}}$ are the posterior mean and covariance from standard GP regression conditioned on the training data $D_T$.

#### E.4.2    NON-STATIONARY GAUSSIAN PROCESS (NSGP)

Our third estimator is the NSGP, which models potential outcomes by defining a single GP over an augmented input space $\mathcal{X} \times \{0, 1\}$ (Alaa & Schaar, 2018). This is achieved by placing a GP prior over a function $f(\boldsymbol{x}, t)$, where the treatment indicator $t$ is an input. The model's key feature is its non-stationary kernel:

$$\boldsymbol{K}_\beta\big((\boldsymbol{x}, t), (\boldsymbol{x}', t')\big) = \begin{cases} k_{\beta_0}(\boldsymbol{x}, \boldsymbol{x}') & \text{if } t = t' = 0 \\ k_{\beta_1}(\boldsymbol{x}, \boldsymbol{x}') & \text{if } t = t' = 1 \\ k_{\beta_0}(\boldsymbol{x}, \boldsymbol{x}') + k_{\beta_1}(\boldsymbol{x}, \boldsymbol{x}') & \text{if } t \neq t' \end{cases} \tag{32}$$

where $k_{\beta_0}$ and $k_{\beta_1}$ are standard Matérn kernels with their own hyperparameters. This allows the response surfaces for the control and treatment arms, $f_0$ and $f_1$, to exhibit different properties

(e.g., smoothness), capturing complex heterogeneity. Posterior inference for the CATE, $\tau(\boldsymbol{x}_*) = f(\boldsymbol{x}_*, 1) - f(\boldsymbol{x}_*, 0)$, follows the same logic as in CMGP, yielding a closed-form Gaussian posterior derived from the joint posterior of the potential outcomes.

### E.5 Deep Kernel Learning for Ablation

For a targeted ablation study, we also include the DUE (Deep Uncertainty Estimation) estimator used in Causal-BALD (Jesson et al., 2021). DUE represents a significant architectural departure from our primary models. It is a deep learning model that uses deep kernel learning to define a sparse variational GP over high-dimensional features learned by a neural network. This end-to-end approach is highly flexible but lacks the strong inductive biases for causal modeling present in BCF and the other GP methods. Its distinct architecture makes it a valuable case for testing the robustness of our acquisition function.

## F Interpretations and Derivations

This section provides the detailed derivations for the information-theoretic acquisition functions discussed in this paper. As all these methods are instantiations of entropy gain—which is equivalently represented by the mutual information principle (see App. C), their mathematical derivations share a common structure. We focus our detailed step-by-step derivation on **Causal-EPIG-$\tau$**, as it represents the most direct application of our framework's principle. The derivation for **Causal-EPIG-$\mu$** follows the same fundamental steps, differing only in the dimensionality of the target variable (a 2D vector vs. a 1D scalar).

### F.1 Detailed Derivation and Estimation of Causal-EPIG

**Step-by-Step Derivation.** We begin with the definition of information gain and show its equivalence to the mutual information and KL divergence forms. The information gain in the CATE at a target point $\boldsymbol{x}^*$, denoted $\tau(\boldsymbol{x}^*)$, that results from observing a new outcome $y$ for a candidate point $(\boldsymbol{x}, t)$ in the pool dataset is the reduction in the entropy of the CATE posterior:

$$\text{IG}((\boldsymbol{x}, t), y, \boldsymbol{x}^*) = \text{H}(\tau(\boldsymbol{x}^*) \mid D_T) - \text{H}(\tau(\boldsymbol{x}^*) \mid D_T \cup \{(\boldsymbol{x}, t, y)\}). \tag{33}$$

The Causal-EPIG is then the expectation of this information gain over both the unknown outcome $y$ and the unknown target point $\boldsymbol{x}^*$. The derivation proceeds as follows:

$$\text{Causal-EPIG}(\boldsymbol{x}, t) \coloneqq \mathbb{E}_{p_{\text{tar}}(\boldsymbol{x}^*)} \mathbb{E}_{p(y|\boldsymbol{x}, t, D_T)} \Big[ \text{IG}_\tau((\boldsymbol{x}, t), y, \boldsymbol{x}^*) \Big] \tag{34}$$

$$= \mathbb{E}_{p_{\text{tar}}(\boldsymbol{x}^*)} \mathbb{E}_{p(y|\boldsymbol{x}, t, D_T)} \Big[ \text{H}(\tau(\boldsymbol{x}^*) \mid D_T) - \text{H}(\tau(\boldsymbol{x}^*) \mid D_T \cup \{(\boldsymbol{x}, t, y)\}) \Big]$$
$$\text{(Expand IG definition)}$$

$$= \mathbb{E}_{p_{\text{tar}}(\boldsymbol{x}^*)} \mathbb{E}_{p(y,\tau(\boldsymbol{x}^*)|\boldsymbol{x}, t, D_T)} \left[ \log \frac{p(\tau(\boldsymbol{x}^*) \mid D_T, y, \boldsymbol{x}, t)}{p(\tau(\boldsymbol{x}^*) \mid D_T)} \right]$$
$$\text{(Combine expectations and logs)}$$

$$= \mathbb{E}_{p_{\text{tar}}(\boldsymbol{x}^*)} \mathbb{E}_{p(y,\tau(\boldsymbol{x}^*)|\boldsymbol{x}, t, D_T)} \left[ \log \frac{p(y,\tau(\boldsymbol{x}^*) \mid \boldsymbol{x}, t, D_T)/p(y \mid \boldsymbol{x}, t, D_T)}{p(\tau(\boldsymbol{x}^*) \mid D_T)} \right]$$
$$\text{(Use def. of conditional prob.)}$$

$$= \mathbb{E}_{p_{\text{tar}}(\boldsymbol{x}^*)} \mathbb{E}_{p(y,\tau(\boldsymbol{x}^*)|\boldsymbol{x}, t, D_T)} \left[ \log \frac{p(y,\tau(\boldsymbol{x}^*) \mid \boldsymbol{x}, t, D_T)}{p(y \mid \boldsymbol{x}, t, D_T)p(\tau(\boldsymbol{x}^*) \mid D_T)} \right]$$
$$\text{(Rearrange terms)}$$

$$= \mathbb{E}_{p_{\text{tar}}(\boldsymbol{x}^*)} \Big[ \text{I}(y; \tau(\boldsymbol{x}^*) \mid (\boldsymbol{x}, t), D_T) \Big]. \quad \text{(Equivalent to Mutual Information)}$$

The final line above is the definition presented in Eq. 5. It is also equivalent to the expected KL Divergence form presented in Eq. 6. The final expressions reveal the core of our method. The mutual information form, $\mathbb{E}_{p_{\text{tar}}(\boldsymbol{x}^*)} \big[ \text{I}(y; \tau(\boldsymbol{x}^*) \mid (\boldsymbol{x}, t), D_T) \big]$, frames the utility as the answer to the question: "On average, across all target points $\boldsymbol{x}^*$, how much will observing a new outcome $y$ reduce our uncertainty about the CATE $\tau(\boldsymbol{x}^*)$?"

**Realization with Specific CATE Models.** Without a closed-form solution, estimating this mutual information would require expensive nested Monte Carlo simulations. However, this general procedure can be made highly efficient for certain model classes.

- **GP Models (CMGP, NSGP):** For GP-based models, the joint predictive posterior $p(y, \tau(\boldsymbol{x}^*) \mid (\boldsymbol{x}, t), D_T)$ is a multivariate Gaussian. In this case, the mutual information has a closed-form analytical solution based on the posterior predictive variances and covariance.

- **BCF:** The BCF posterior is represented by MCMC samples. To make Causal-EPIG computationally feasible, we adopt the approximation strategy from Sec. 4.2, fitting a multivariate Gaussian to the joint posterior samples of $(y, \tau(\boldsymbol{x}^*))$. This allows us to again use the closed-form solution, bypassing the need for nested sampling.

### F.1.1 ANALYTICAL FORM OF CAUSAL-EPIG FOR GAUSSIAN MODELS

A key advantage of our framework is that when the underlying CATE estimator has a Gaussian posterior predictive distribution (such as GP models), the mutual information term in the Causal-EPIG objective has a closed-form analytical solution. Here, we provide a step-by-step derivation.

**Assumption: Gaussian Predictive Distribution.** We assume that for a candidate point $(\boldsymbol{x}, t)$ and a target point $\boldsymbol{x}^*$, the joint posterior predictive distribution of the potential outcome $y$ and the CATE $\tau(\boldsymbol{x}^*)$ is a bivariate Gaussian. All distributions are implicitly conditioned on the existing data $D_T$.

$$p(y, \tau(\boldsymbol{x}^*) \mid \boldsymbol{x}, t, \boldsymbol{x}^*, D_T) = \mathcal{N}(\boldsymbol{\mu}, \boldsymbol{\Sigma}) \tag{35}$$

where the covariance matrix $\boldsymbol{\Sigma}$ is given by:

$$\boldsymbol{\Sigma} = \begin{pmatrix} \mathrm{Var}[y] & \mathrm{Cov}[y, \tau(\boldsymbol{x}^*)] \\ \mathrm{Cov}[\tau(\boldsymbol{x}^*), y] & \mathrm{Var}[\tau(\boldsymbol{x}^*)] \end{pmatrix} \tag{36}$$

The marginal distributions for $y$ and $\tau(\boldsymbol{x}^*)$ are also Gaussian, with variances corresponding to the diagonal elements of $\boldsymbol{\Sigma}$.

**Derivation.** We begin with the definition of mutual information in terms of differential entropies:

$$I(y; \tau(\boldsymbol{x}^*)) = H(y) + H(\tau(\boldsymbol{x}^*)) - H(y, \tau(\boldsymbol{x}^*)) \qquad \text{(by definition)} \tag{37}$$

For a univariate Gaussian variable $z$ with variance $\sigma^2$, the differential entropy is $H(z) = \frac{1}{2}\log(2\pi e \sigma^2)$. For a $k$-dimensional multivariate Gaussian with covariance matrix $\boldsymbol{\Sigma}$, the joint entropy is $H(\mathbf{z}) = \frac{1}{2}\log\big((2\pi e)^k \det(\boldsymbol{\Sigma})\big)$. Applying these formulas to our bivariate case ($k = 2$):

$$I(y; \tau(\boldsymbol{x}^*)) = \left(\frac{1}{2}\log(2\pi e \mathrm{Var}[y])\right) + \left(\frac{1}{2}\log(2\pi e \mathrm{Var}[\tau(\boldsymbol{x}^*)])\right) - \left(\frac{1}{2}\log\big((2\pi e)^2 \det(\boldsymbol{\Sigma})\big)\right)$$

$$\text{(substitute Gaussian entropies)}$$

$$= \frac{1}{2}\left[\log(2\pi e \mathrm{Var}[y]) + \log(2\pi e \mathrm{Var}[\tau(\boldsymbol{x}^*)]) - \log((2\pi e)^2 \det(\boldsymbol{\Sigma}))\right] \tag{38}$$

$$= \frac{1}{2}\left[\log\big((2\pi e)^2 \mathrm{Var}[y]\mathrm{Var}[\tau(\boldsymbol{x}^*)]\big) - \log\big((2\pi e)^2 \det(\boldsymbol{\Sigma})\big)\right] \quad \text{(combine log terms)}$$

$$= \frac{1}{2}\log\left(\frac{\mathrm{Var}[y]\mathrm{Var}[\tau(\boldsymbol{x}^*)]}{\det(\boldsymbol{\Sigma})}\right) \qquad \text{(cancel terms)}$$

$$\tag{39}$$

Now, we substitute the determinant of the 2x2 covariance matrix, $\det(\boldsymbol{\Sigma}) = \mathrm{Var}[y]\mathrm{Var}[\tau(\boldsymbol{x}^*)] - \mathrm{Cov}[y, \tau(\boldsymbol{x}^*)]^2$:

$$I(y; \tau(\boldsymbol{x}^*)) = \frac{1}{2}\log\left(\frac{\mathrm{Var}[y]\mathrm{Var}[\tau(\boldsymbol{x}^*)]}{\mathrm{Var}[y]\mathrm{Var}[\tau(\boldsymbol{x}^*)] - \mathrm{Cov}[y, \tau(\boldsymbol{x}^*)]^2}\right). \qquad \text{(substitute determinant)}$$

This is the closed-form solution for the mutual information under the Gaussian assumption.

**Final Causal-EPIG-$\tau$ Formulation.** The final Causal-EPIG utility is the expectation of this analytical term over the target distribution $p_{\text{tar}}(\boldsymbol{x}^*)$. In practice, this expectation is approximated by the empirical average over the finite target set $\boldsymbol{X}_{\text{tar}}$:

$$\text{Causal-EPIG} - \tau(\boldsymbol{x}, t) = \mathbb{E}_{p_{\text{tar}}(\boldsymbol{x}^*)} \left[ \text{I}(y; \tau(\boldsymbol{x}^*)) \right]$$

$$\approx \frac{1}{|\boldsymbol{X}_{\text{tar}}|} \sum_{\boldsymbol{x}^* \in \boldsymbol{X}_{\text{tar}}} \frac{1}{2} \log \left( \frac{\text{Var}[y]\text{Var}[\tau(\boldsymbol{x}^*)]}{\text{Var}[y]\text{Var}[\tau(\boldsymbol{x}^*)] - \text{Cov}[y, \tau(\boldsymbol{x}^*)]^2} \right), \tag{40}$$

where the variances and covariance are computed for each candidate-target pair $(\boldsymbol{x}, \boldsymbol{x}^*)$.

## F.2 A TAXONOMY OF INFORMATION-THEORETIC ACQUISITION FUNCTIONS

Our proposed Causal-EPIG framework is part of a broader family of information-theoretic acquisition functions. To clarify its specific contributions and design choices, it is useful to deconstruct the landscape of these methods along four key axes. Tab. 5 provides a detailed taxonomy that informs the following discussion.

Table 5: A Taxonomy of Information-Theoretic Acquisition Functions for active CATE Estimation. The table distinguishes methods along several key axes, including their core target (parameters vs. predictions) and their formulation (mean-marginal vs. global).

| Family | Target of Information Gain | Mean-Marginal Formulation | Global / Full Formulation |
|---|---|---|---|
| **EIG** | Model Parameters ($\theta$) | $\text{I}(y; \theta_\tau \mid (\boldsymbol{x}, t), D_T)$  or  $\text{I}(y; \theta \mid (\boldsymbol{x}, t), D_T)$ | |
| **EPIG / ITL** | Factual Outcome ($y^*$) | $\mathbb{E}_{p_{\text{pool}}(\boldsymbol{x}^*, t^*)}[\text{I}(y; y^* \mid (\boldsymbol{x}, t), (\boldsymbol{x}^*, t^*), D_T)]$ | $\text{I}(y; \boldsymbol{y}^* \mid (\boldsymbol{x}, t), D_T)$ |
| | Potential Outcomes ($y^*(t^*)$) | $\mathbb{E}_{p_{\text{tar}}(\boldsymbol{x}^*)} \left[ \sum_{t^*} \text{I}(y; y^*(t^*)) \right]$ (Additive Approx.) $\mathbb{E}_{p_{\text{tar}}(\boldsymbol{x}^*)} \left[ \text{I}(y; (y^*(0), y^*(1))) \right]$ **(Joint PO)** | $\text{I}(y; \boldsymbol{y}_{PO}^* \mid (\boldsymbol{x}, t), D_T)$ |
| | **CATE** ($\tau(\boldsymbol{x}^*)$) | $\mathbb{E}_{p_{\text{tar}}(\boldsymbol{x}^*)} \left[ \text{I}(y; \tau(\boldsymbol{x}^*)) \right]$ | $\text{I}(y; \boldsymbol{\tau} \mid (\boldsymbol{x}, t), D_T)$ |

**Axis 1: Parameters vs. Predictions.** The most fundamental distinction is the target of the information gain. The **EIG** family (which includes Causal-EIG and Causal-BALD) is **parameter-focused**. These methods aim to reduce uncertainty over the model's internal representation, such as the CATE-specific parameters $\theta_\tau$ or the full parameter set $\theta$. In contrast, the entire **EPIG/ITL** family, including our work, is **prediction-focused**, directly targeting uncertainty in the model's outputs. This is generally preferred for function estimation tasks, as it concentrates effort on the final quantity of interest.

**Axis 2: The Hierarchy of Predictive Targets.** Within the prediction-focused family, a clear hierarchy emerges based on the causal relevance of the target:

- **Factual EPIG:** Targets a future factual outcome $y^*$, which is insufficient as it does not actively pursue counterfactual knowledge.
- **Potential Outcomes (PO-based):** Targets the foundational components of the causal effect, $y^*(0)$ and $y^*(1)$. This is a robust causal objective.
- **CATE-based:** Targets the final causal estimand, $\tau(\boldsymbol{x}^*)$, itself. This is the most direct causal objective.

**Axis 3: Formulating the PO-based Objective.** Once potential outcomes are chosen as the target, there are two primary ways to formulate the mutual information objective:

- **Additive Formulation:** The simpler approach approximates the information gain by summing the MI for each potential outcome separately: $\text{I}(y; y^*(0)) + \text{I}(y; y^*(1))$. This is computationally efficient but ignores the dependency structure between the potential outcomes. This is the "simpler, additive variant" we refer to in the main text. In the App. H, we mark this method as Causal-EPIG-$\mu$-S, which means Separation.

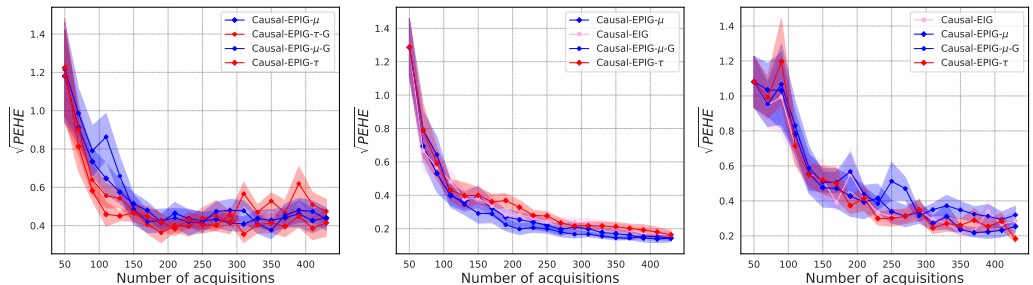

Figure 11: Performance comparison on the Hahn (linear) dataset with distribution shift. Panels show results for different underlying CATE estimators. The summation-based methods consistently perform on par with or better than their global counterparts.

- **Joint Formulation:** A more theoretically robust approach is to target the joint distribution of the potential outcomes, as in $I(y; (y^*(0), y^*(1)))$. This correctly accounts for the correlation between the two outcomes. **This is the formulation we adopt for our primary Causal-EPIG-$\mu$ method.**

**Axis 4: Aggregation Across the Target Population.** The final distinction lies in how information gain is aggregated across the entire target population.

- The **Mean-Marginal** formulation (left column in Tab. 5) is computationally efficient. It approximates the total information gain by averaging the gains over each target point independently, ignoring correlations between target predictions (e.g., between $\tau(x_1^*)$ and $\tau(x_2^*)$). **Our work focuses on this formulation for its scalability.**

- The **Global / Full** formulation (right column) is more theoretically complete. It calculates the information gain with respect to the entire set of target predictions jointly (e.g., $I(y; \boldsymbol{\tau})$), capturing all interdependencies but at a significantly higher computational cost. We denote this method with a $-G$ suffix, where $G$ indicates Global.

Our choice of the mean-marginal formulation for Causal-EPIG represents a pragmatic trade-off between computational scalability and theoretical completeness.

### F.3 COMPUTATIONAL COMPLEXITY AND RUNTIME ANALYSIS

The primary computational cost of the Causal-EPIG acquisition functions is driven by the size of the candidate pool ($n_P$), the target set ($n_{\text{tar}} = |X_{\text{tar}}|$), and the number of posterior samples ($S$).

**Theoretical Complexity.** A key design choice in our framework is between the **summation** and **global** formulations. The summation approach, which we adopt, is designed for efficiency. The total complexity to score all $n_P$ candidates is $\mathcal{O}(n_P \cdot n_{\text{tar}} \cdot S)$, scaling linearly with the pool and target set sizes. In contrast, the global formulation requires inverting an $n_{\text{tar}} \times n_{\text{tar}}$ covariance matrix for each candidate, leading to a total complexity of $\mathcal{O}(n_P \cdot n_{\text{tar}}^3)$. This cubic scaling makes the global approach computationally prohibitive for even moderately large target populations.

**Empirical Validation and Comparison.** This theoretical trade-off is strongly validated by our empirical results, presented in Tab. 6. The data confirms that our chosen summation-based methods are one to two orders of magnitude faster than their global counterparts, justifying our design choice. For instance, Causal-EPIG-$\mu$ is approximately **20 times faster** than its global version (0.45s vs. 9.27s).

Having justified our formulation, we next compare its runtime to established baselines in Tab. 7. While Causal-EPIG-$\tau$ (0.44s in Tab. 6) is slower than the fast BALD variants, this represents a deliberate trade-off. The effectiveness of our approach is most pronounced in settings where the cost of labeling is the dominant factor, such as in clinical trials. In these scenarios, the marginal computational overhead is typically negligible compared to the cost of acquiring each new label, making the superior sample efficiency of Causal-EPIG a highly practical choice.

Table 6: Average acquisition time (seconds) per batch, comparing summation-based (our choice) vs. global (-G) formulations. Results are mean ± std across trials.

| Estimator | Causal-EPIG-$\mu$ | Causal-EPIG-$\mu$-G | Causal-EPIG-$\tau$ | Causal-EPIG-$\tau$-G |
|---|---|---|---|---|
| BCF | $0.7731 \pm 0.0508$ | $16.1392 \pm 0.5763$ | $0.5392 \pm 0.0342$ | $4.1811 \pm 0.1036$ |
| CMGP | $0.1813 \pm 0.0164$ | $4.1814 \pm 0.1598$ | $0.4016 \pm 0.0289$ | $1.9840 \pm 0.0883$ |
| NSGP | $0.3931 \pm 0.0331$ | $7.5014 \pm 1.0600$ | $0.3812 \pm 0.0334$ | $2.9213 \pm 0.4922$ |
| Overall | $0.4492 \pm 0.2999$ | $9.2740 \pm 6.1728$ | $0.4406 \pm 0.0860$ | $3.0288 \pm 1.1025$ |

Table 7: Average running times (in seconds) of Causal-EPIG-$\tau$ compared to other baselines.

| Methods | Random | $\mu$-BALD | $\mu\rho$-BALD | $\mu\pi$-BALD | Causal-EPIG-$\tau$ |
|---|---|---|---|---|---|
| Time (s) | $(6.5 \pm 0.7)$ $\times 10^{-5}$ | $(3.6 \pm 0.1)$ $\times 10^{-3}$ | $(9.6 \pm 0.4)$ $\times 10^{-3}$ | $(9.6 \pm 0.4)$ $\times 10^{-3}$ | $(4.4 \pm 0.1)$ $\times 10^{-1}$ |

## G THEORETICAL ANALYSIS DETAILS

### G.1 FRAMEWORK MAPPING TO TAL

Our theoretical analysis leverages the framework of TAL (Hübotter et al., 2024), which requires mapping our CATE estimation problem to their single-task GP setting. The full mapping is as follows:

- Augmented Space $\tilde{\mathcal{X}}$ and Kernel $\tilde{k}$: We model the potential outcome surfaces $f_t(\boldsymbol{x})$ as a single GP $f(\tilde{\boldsymbol{x}})$ over an augmented space $\tilde{\mathcal{X}} = \mathcal{X} \times \{0,1\}$, where $\tilde{\boldsymbol{x}} = (\boldsymbol{x}, t)$. The augmented kernel $\tilde{k}$ is a sum of separable kernels (a standard Linear Mode coregionlization (LMC) construction) (Alaa & Van Der Schaar, 2017):

$$\tilde{k}((\boldsymbol{x},t),(\boldsymbol{x}',t')) := \sum_{q=1}^{Q}(B_q)_{t,t'} \cdot k_q(\boldsymbol{x},\boldsymbol{x}'), \tag{41}$$

where $B_q$ are $2 \times 2$ coregionalization matrices and $k_q$ are base kernels. We also set $Q = 2$ in our paper as that in CMGP.

- Augmented Target Space $\tilde{\boldsymbol{X}}_{\text{tar}}$: The set of paired potential outcomes for the target population $\boldsymbol{X}_{\text{tar}}$, defined as $\tilde{\boldsymbol{X}}_{\text{tar}} = \{(\boldsymbol{x}^*, t) \mid \boldsymbol{x}^* \in \boldsymbol{X}_{\text{tar}}, t \in \{0,1\}\}$.

- Augmented Pool Space $\tilde{\mathcal{D}}_P$: This corresponds to the factual observations available in our pool $\mathcal{D}_P$, defined as $\tilde{\mathcal{D}}_P = \{(\boldsymbol{x}, t) \mid (\boldsymbol{x}, t) \in \mathcal{D}_P\}$. We use this augmented notation for consistency with the target-space construction.

Our convergence analysis focuses on the global, joint PO-based strategy, denoted Causal-EPIG-$\mu$-G (discussed in App. F.2), as it is a direct instantiation of the Global Information Theoretic Learning (ITL) strategy, defined in Hübotter et al. (2024, Eq. 2). We adapt the proof structure from Hübotter et al. (2024, Thm. 3.3), which demonstrates convergence for this Global ITL strategy. This proof relies on the utility function being submodular, which we state in our main paper as Ass. 2. We now provide the justification for this assumption.

**Justification 1** *The validity of Ass. 2 depends on the relationship between the augmented target space $\tilde{\boldsymbol{X}}_{\text{tar}}$ and the augmented pool space $\tilde{\mathcal{D}}_P$:*

- *Case 1: "Regular Setup" (No Distribution Shift over Covariates). In this standard setup, the target population $\boldsymbol{X}_{\text{tar}}$ and the pool $\mathcal{D}_P$ are defined over the same set of underlying covariates. As every factual observation $(\boldsymbol{x}_i, t_i)$ in $\mathcal{S} = \tilde{\mathcal{D}}_P$ corresponds to a target individual $\boldsymbol{x}_i$ in $\boldsymbol{X}_{\text{tar}}$ (for which both outcomes $(\boldsymbol{x}_i, 0)$ and $(\boldsymbol{x}_i, 1)$ are in $\mathcal{A} = \tilde{\boldsymbol{X}}_{\text{tar}}$), we have the relationship $\mathcal{S} \subset \mathcal{A}$. Under this condition ($\mathcal{S} \subseteq \mathcal{A}$), Hübotter et al. (2024, Lemma C.9) prove that the Global ITL objective is submodular. Thus, Ass. 2 is guaranteed to hold.*

- *Case 2: "Distribution Shift." In this setting, the target population $\boldsymbol{X}_{\text{tar}}$ and the pool $\mathcal{D}_P$ are defined over different sets of covariates. Therefore, an observation $(\boldsymbol{x}, t)$ in $\mathcal{S} = \tilde{\mathcal{D}}_P$ does not necessarily correspond to a target $\boldsymbol{x}^*$ in $\boldsymbol{X}_{\text{tar}}$. In this general case, $\mathcal{S} \notsubseteq \mathcal{A}$. This is the general transductive setting where submodularity is not guaranteed, as "synergistic" effects can occur (Hübotter et al., 2024, Example C.8). In this case, our Thm. 1 relies on an unproven assumption, mirroring the theoretical gap in the TAL framework itself.*

The theoretical implications of using the structured LMC kernel are discussed in Sec. G.4.

## G.2 Approximate Markov Boundary

**Intuition 1** *In our active CATE estimation problem, even if we acquire all the factual outcomes in the pool $\tilde{\mathcal{D}}_P$, the uncertainty over a target point $\tilde{\boldsymbol{x}}^* \in \tilde{\boldsymbol{X}}_{\text{tar}}$ may not be zero. This remaining uncertainty is the irreducible uncertainty $\eta^2_{\tilde{\mathcal{D}}_P}(\tilde{\boldsymbol{x}}^*)$. The AMB is the smallest subset of the pool $\tilde{\mathcal{D}}_P$ that is "good enough" to achieve this minimal uncertainty, up to an error $\epsilon$.*

**Definition 2 (Approximate Markov Boundary (AMB))** *For any $\epsilon > 0$, $n_T \geq 0$, and target point $\tilde{\boldsymbol{x}}^* \in \tilde{\boldsymbol{X}}_{\text{tar}}$, we define $\boldsymbol{B}_{n_T,\epsilon}(\tilde{\boldsymbol{x}}^*)$ as the smallest (multi-)set of $\tilde{\mathcal{D}}_P$, satisfying:*

$$\mathbb{V}\text{ar}\left[f(\tilde{\boldsymbol{x}}^*) \mid \mathcal{D}_T, \mathbf{y}_{\boldsymbol{B}_{n_T,\epsilon}(\tilde{\boldsymbol{x}}^*)}\right] \leq \eta^2_{\tilde{\mathcal{D}}_P}(\tilde{\boldsymbol{x}}^*) + \epsilon. \tag{42}$$

*where $n_T = |\mathcal{D}_T|$ is the number of observations in $\mathcal{D}_T$. We refer to $\boldsymbol{B}_{n_T,\epsilon}(\tilde{\boldsymbol{x}}^*)$ as the $\epsilon$-approximate Markov boundary of $\tilde{\boldsymbol{x}}^*$ in $\tilde{\mathcal{D}}_P$.*

The existence and finite size of this set are guaranteed, as shown in Lemma C.16 of (Hübotter et al., 2024). We restate the consequence here:

**Lemma 2 (AMB Existence, adapted from (Hübotter et al., 2024))** *Let $\epsilon > 0$ and define $r$ as the smallest integer satisfying*

$$\frac{\gamma_r}{r} \leq \frac{\epsilon \lambda_{\min}(\boldsymbol{K}_{\tilde{\mathcal{D}}_P \tilde{\mathcal{D}}_P})}{2|\tilde{\mathcal{D}}_P|\sigma_I^2 \tilde{\sigma}_I^2}, \tag{43}$$

*where $\gamma_r \stackrel{\text{def}}{=} \max_{\tilde{\boldsymbol{X}} \subseteq \tilde{\mathcal{D}}_P, |\tilde{\boldsymbol{X}}| \leq r} I(f_{\tilde{\mathcal{D}}_P}; \mathbf{y}_{\tilde{\boldsymbol{X}}})$, and $\sigma_I^2, \tilde{\sigma}_I^2$ are variance constants. For any $n_T \geq 0$ and $\tilde{\boldsymbol{x}}^* \in \tilde{\boldsymbol{X}}_{\text{tar}}$, there exists an $\epsilon$-approximate Markov boundary $\boldsymbol{B}_{n_T,\epsilon}(\tilde{\boldsymbol{x}}^*)$ for $\tilde{\boldsymbol{x}}^*$ within $\tilde{\mathcal{D}}_P$, with a size bounded by $r$.*

## G.3 Proof of Thm. 1

To prove Thm. 1, we adapt the analytical framework of Hübotter et al. (2024, Thm. 3.3). This proof is simpler than that of ActiveCQ (Gao & Sejdinovic, 2025) as we are bounding the variance of a point prediction $f(\tilde{\boldsymbol{x}}^*)$ rather than an integral $\upsilon_{\bar{\boldsymbol{x}}}$.

The proof proceeds in three main steps:

1. We leverage the AMB (Def. 2) to relate the current variance $\mathbb{V}\text{ar}[f(\tilde{\boldsymbol{x}}^*)|\mathcal{D}_T]$ to the information gain of the AMB set $\boldsymbol{B}$.

2. We bound this AMB information gain by the Global ITL maximal marginal gain $\Gamma_{n_T}$ (defined in Step 2).

3. We select a decaying approximation error $\epsilon$ and use the convergence rate of $\Gamma_{n_T}$ (which relies on Ass. 2) to derive the final rate.

**Step 1: Bound Variance by AMB Information Gain.** From Lemma C.17 in (Hübotter et al., 2024), we can bound the current variance of the estimator for any $\tilde{\boldsymbol{x}}^* \in \tilde{\boldsymbol{X}}_{\text{tar}}$ as:

$$\mathbb{V}\text{ar}[f(\tilde{\boldsymbol{x}}^*)|\mathcal{D}_T] \leq C_0 \cdot I(f(\tilde{\boldsymbol{x}}^*); \mathbf{y}_{\boldsymbol{B}_{n_T,\epsilon}(\tilde{\boldsymbol{x}}^*)}|\mathcal{D}_T) + \eta^2_{\tilde{\mathcal{D}}_P}(\tilde{\boldsymbol{x}}^*) + \epsilon. \tag{44}$$

where $C_0$ is a constant related to the max variance (e.g., $2\sigma_I^2$).

**Step 2: Bound Point Gain by Global ITL Marginal Gain.** Next, we bound the mutual information term from Step 1, $\mathrm{I}(f(\tilde{\boldsymbol{x}}^*); \mathrm{y}_{\boldsymbol{B}} | \mathcal{D}_T)$, where $\boldsymbol{B} \triangleq \boldsymbol{B}_{n_T, \epsilon}(\tilde{\boldsymbol{x}}^*)$ and $b_\epsilon \triangleq |\boldsymbol{B}|$. The derivation proceeds by first relating the information gain about a single target point $f(\tilde{\boldsymbol{x}}^*)$ to the joint information gain about all target points $f_{\tilde{\boldsymbol{X}}_{\mathrm{tar}}}$, which is the utility function $\psi_{\tilde{\boldsymbol{X}}_{\mathrm{tar}}}(\cdot)$ for our Causal-EPIG-$\mu$-G strategy.

$$\mathrm{I}(f(\tilde{\boldsymbol{x}}^*); \mathrm{y}_{\boldsymbol{B}} | \mathcal{D}_T) \le \mathrm{I}(f_{\tilde{\boldsymbol{X}}_{\mathrm{tar}}}; \mathrm{y}_{\boldsymbol{B}} | \mathcal{D}_T) \tag{45}$$

$$\stackrel{\text{def}}{=} \psi_{\tilde{\boldsymbol{X}}_{\mathrm{tar}}}(\boldsymbol{B} \mid \mathcal{D}_T) \tag{46}$$

where Eq. 45 holds because the information about a single component $f(\tilde{\boldsymbol{x}}^*)$ cannot exceed the information about the entire vector $f_{\tilde{\boldsymbol{X}}_{\mathrm{tar}}}$ it belongs to. Next, we bound the utility of the set $\boldsymbol{B}$ using Ass. 2. Submodularity implies that the gain from a set is no more than the sum of the marginal gains of its individual elements (evaluated without conditioning on each other):

$$\psi_{\tilde{\boldsymbol{X}}_{\mathrm{tar}}}(\boldsymbol{B} \mid \mathcal{D}_T) \le \sum_{\tilde{\boldsymbol{x}}_i \in \boldsymbol{B}} \psi_{\tilde{\boldsymbol{X}}_{\mathrm{tar}}}(\{\tilde{\boldsymbol{x}}_i\} \mid \mathcal{D}_T) \tag{47}$$

$$= \sum_{\tilde{\boldsymbol{x}}_i \in \boldsymbol{B}} \mathrm{I}(f_{\tilde{\boldsymbol{X}}_{\mathrm{tar}}}; \mathrm{y}_{\tilde{\boldsymbol{x}}_i} \mid \mathcal{D}_T) \tag{48}$$

We now define the maximal marginal gain for the Global ITL strategy at step $n_T + 1$ (i.e., given $\mathcal{D}_T$ which has $n_T$ points) as:

$$\Gamma_{n_T+1} \stackrel{\text{def}}{=} \max_{\tilde{\boldsymbol{x}} \in \tilde{\mathcal{D}}_P} \mathrm{I}(f_{\tilde{\boldsymbol{X}}_{\mathrm{tar}}}; \mathrm{y}_{\tilde{\boldsymbol{x}}} \mid \mathcal{D}_T). \tag{49}$$

By definition, the gain of any individual point $\tilde{\boldsymbol{x}}_i$ in the sum Eq. 48 is bounded by this maximum:

$$\sum_{\tilde{\boldsymbol{x}}_i \in \boldsymbol{B}} \mathrm{I}(f_{\tilde{\boldsymbol{X}}_{\mathrm{tar}}}; \mathrm{y}_{\tilde{\boldsymbol{x}}_i} \mid \mathcal{D}_T) \le \sum_{i=1}^{b_\epsilon} \Gamma_{n_T+1} \tag{50}$$

$$\le b_\epsilon \cdot \Gamma_{n_T} \tag{51}$$

where Eq. 51 follows if we assume the maximal marginal gain $\Gamma_k$ is non-increasing, which is a direct consequence of Assumption 2 (Hübotter et al., 2024, Thm. D.1).

Combining Step 1 and Step 2 (Eq. Eq. 45 through Eq. 51), we have the intermediate bound:

$$\mathbb{V}\mathrm{ar}[f(\tilde{\boldsymbol{x}}^*)|\mathcal{D}_T] \le C_0 b_\epsilon \Gamma_{n_T} + \eta^2_{\mathcal{D}_P}(\tilde{\boldsymbol{x}}^*) + \epsilon. \tag{52}$$

**Step 3: Substitute Decaying Bounds.** Now, we select a specific value for $\epsilon$ that decays with $n_T$. Let $\epsilon = c \frac{\gamma_{\sqrt{n_T}}}{\sqrt{n_T}}$, for a constant $c$. Here, $\gamma_k \triangleq \max_{|\tilde{\boldsymbol{X}}| \le k} \mathrm{I}(f_{\tilde{\mathcal{D}}_P}; \mathrm{y}_{\tilde{\boldsymbol{X}}})$ is the *pool's* global information capacity, as defined in Hübotter et al. (2024, Lem. C.16) (which corresponds to $\gamma_r$ in our Lem. 2). From Lemma 2 (which is based on Hübotter et al. (2024, Eq. 18)), this choice of $\epsilon$ ensures an AMB exists with a size $b_\epsilon$ bounded by $r$. By setting $k = r = \sqrt{n_T}$, the condition $\gamma_k/k \le \epsilon \cdot K$ (where $K$ is a constant) is satisfied, thus $b_\epsilon \le \sqrt{n_T}$. Substituting $b_\epsilon \le \sqrt{n_T}$ and the expression for $\epsilon$ into our intermediate bound, we get:

$$\mathbb{V}\mathrm{ar}[f(\tilde{\boldsymbol{x}}^*)|\mathcal{D}_T] \le (C_0)\sqrt{n_T}\Gamma_{n_T} + \eta^2_{\mathcal{D}_P}(\tilde{\boldsymbol{x}}^*) + c\frac{\gamma_{\sqrt{n_T}}}{\sqrt{n_T}}. \tag{53}$$

The term $\Gamma_{n_T}$ itself must decay. We use the bound for the Global ITL strategy from Hübotter et al. (2024, Thm. C.12). Under Ass. 2 (which implies $\alpha_n \le 1$), this theorem provides the bound:

$$\Gamma_{n_T} \le \frac{\gamma_{n_T}(\tilde{\boldsymbol{X}}_{\mathrm{tar}}; \tilde{\mathcal{D}}_P)}{n_T} \tag{54}$$

where $\gamma_{n_T}(\tilde{\boldsymbol{X}}_{\mathrm{tar}}; \tilde{\mathcal{D}}_P)$ is precisely the global information capacity $\gamma_{n_T}$ as defined in our Definition 1. Substituting this bound for $\Gamma_{n_T}$ into Eq. 53 resolves the term:

$$(C_0)\sqrt{n_T}\Gamma_{n_T} \le (C_0)\sqrt{n_T}\left(\frac{\gamma_{n_T}}{n_T}\right) = C_0 \frac{\gamma_{n_T}}{\sqrt{n_T}}. \tag{55}$$

Combining all the pieces, the total variance is bounded by:

$$\mathbb{V}\mathrm{ar}[f(\tilde{\boldsymbol{x}}^*)|\mathcal{D}_T] \le C_0 \frac{\gamma_{n_T}}{\sqrt{n_T}} + \eta^2_{\mathcal{D}_P}(\tilde{\boldsymbol{x}}^*) + c\frac{\gamma_{\sqrt{n_T}}}{\sqrt{n_T}}$$

$$\le \eta^2_{\mathcal{D}_P}(\tilde{\boldsymbol{x}}^*) + C\frac{\gamma_{n_T}}{\sqrt{n_T}}, \tag{56}$$

where the final step combines all constants into a single $C$ and leverages the structural properties of the two capacity terms. Specifically, the term $C_0 \frac{\gamma_{n_T}}{\sqrt{n_T}}$ captures the uncertainty reduction related to the transductive objective ($\mathcal{A}$-to-$\mathcal{S}$ capacity), while $c \frac{\gamma_{\sqrt{n_T}}}{\sqrt{n_T}}$ bounds the error introduced by the AMB approximation (related to the pool capacity). Since both terms decay at the same asymptotic rate $\frac{1}{\sqrt{n_T}}$, the final bound is simplified to the form determined by the transductive term $\gamma_{n_T}$. In the main paper, we use $n_B$ for the total number of acquired samples, which corresponds to $n_T$ here. This completes the proof sketch for Thm. 1.

## G.4 Implications of the LMC Kernel Structure

The convergence rate in Thm. 1 is determined by the global information capacity $\gamma_{n_B}$. This quantity is defined by the joint mutual information $I(f_{\tilde{\boldsymbol{X}}_{\text{tar}}}; y_{\tilde{\boldsymbol{X}}})$, which is governed by our augmented kernel $\tilde{k}$. Our augmented kernel, $\tilde{k}((\boldsymbol{x}, t), (\boldsymbol{x}', t')) = \sum_{q=1}^{Q} (B_q)_{t,t'} \cdot k_q(\boldsymbol{x}, \boldsymbol{x}')$, explicitly models the correlation between the potential outcome surfaces $f_0$ and $f_1$ via the off-diagonal elements of the task-correlation matrices $B_q$. This has a direct and crucial consequence on the information capacity $\gamma_{n_B}$. The information capacity $\gamma_{n_B}$ is known to be sublinear, and its magnitude depends on the effective dimensionality of the function space. Let $\boldsymbol{K}_{\tilde{\mathcal{D}}_P}$ be the kernel matrix for the entire augmented pool $\tilde{\mathcal{D}}_P$. This matrix inherits a sum-of-Kronecker-products structure from $\tilde{k}$:

$$\boldsymbol{K}_{\tilde{\mathcal{D}}_P} = \sum_{q=1}^{Q} \boldsymbol{K}_{\mathcal{X},q} \otimes B_q \tag{57}$$

where $\boldsymbol{K}_{\mathcal{X},q}$ is the kernel matrix on the covariate space $\mathcal{X}$ for the $q$-th base kernel. The eigenvalues $\lambda(\boldsymbol{K}_{\tilde{\mathcal{D}}_P})$ are thus a combination of the eigenvalues of the base kernels and the coregionalization matrices. This structure has a direct, quantifiable impact on the convergence rate:

- Independent Model (Non-Causal-Aware): If we model $f_0$ and $f_1$ independently, this is equivalent to setting all $A_q$ to be diagonal matrices. The information capacity $\gamma_{n_B}^{\text{indep}}$ reflects the complexity of learning two independent functions.

- LMC Model (Causally-Aware): If $f_0$ and $f_1$ are correlated (i.e., $A_q$ are not diagonal), the off-diagonal elements $(A_q)_{0,1}$ are non-zero. This correlation "compresses" the spectrum of $\boldsymbol{K}_{\tilde{\mathcal{D}}_P}$ and reduces the effective dimensionality of the problem. For instance, a strong positive correlation implies that $f_0$ and $f_1$ share a large component, reducing the "new" information needed to learn both. In Consequence: This reduction in effective dimensionality leads to a smaller information capacity, $\gamma_{n_B}^{\text{LMC}} < \gamma_{n_B}^{\text{indep}}$. Therefore, our causally-aware LMC model achieves a provably faster convergence rate (a smaller $C(\gamma_{n_B}/\sqrt{n_B})$ term) than a non-causal-aware approach that models the potential outcomes independently. This provides a formal theoretical justification for our causally-aligned, multitask approach.

## G.5 Connections to Other Causal Strategies

**1. Connection to Mean-Marginal (MM-ITL) Strategies.** The analysis in Sec. G.1-G.3 provides a convergence guarantee for the global, joint PO-based strategy (Causal-EPIG-$\mu$-G). This strategy is theoretically robust as it aligns with the joint Global ITL objective. However, as discussed in App. F.2, this global strategy can be computationally expensive. A common, more scalable alternative is the mean-marginal (MM-ITL) strategy, which Causal-EPIG-$\mu$-S instantiates. This strategy approximates the joint gain as a sum of marginal gains. While we do not provide a convergence proof for this approximate strategy, Hübotter et al. (2024, Thm. D.1) analyze it. We note that their analysis also relies on a non-trivial assumption of non-increasing marginal gains ($\Gamma_k \geq \Gamma_{k+1}$), which, much like our Ass. 2, is not guaranteed to hold in the general $\mathcal{A} \neq \mathcal{S}$ transductive setting.

**2. The Non-Intuitive ($\mu$ vs. $\tau$) Relationship.** The relationship between the PO-based objective (Causal-EPIG-$\mu$) and the CATE-based objective (Causal-EPIG-$\tau$) is not straightforward. This relationship can be formally understood using the Data Processing Inequality. Let $(f_0, f_1)$ be the joint potential outcomes for a target $x^*$, and $\tau = f_1 - f_0$ be the CATE. The CATE is a function (a simple subtraction) of the joint potential outcomes. This creates an information-processing Markov

chain $y_{\tilde{x}} \to (f_0, f_1) \to \tau$. The Data Processing Inequality states that information cannot be created by post-processing. Therefore, for any query $y_{\tilde{x}}$:

$$I(y_{\tilde{x}}; (f_0, f_1)) \geq I(y_{\tilde{x}}; \tau) \tag{58}$$

This inequality provides the formal basis for the "comprehensiveness-focus" trade-off:

- The $\mu$-strategy (LHS) always optimizes an information quantity that is greater than or equal to the $\tau$-strategy (RHS).
- This is "non-intuitive" because a query $y_{\tilde{x}}$ might be highly informative about the prognostic part of the joint distribution (e.g., $f_0$) but provide little information about the difference ($\tau$).
- The $\mu$-strategy would (correctly) assign high value to this query, as it reduces uncertainty about the full causal mechanism. The $\tau$-strategy would (also correctly for its objective) assign low value, focusing only on the estimand of interest.

This confirms that neither strategy is universally superior; the choice is context-dependent, as concluded in the main paper.

### G.6 PROOF OF PROP. 1

**Proof 1** *Let $\mathcal{F}_s$ denote the information available after $s$ acquisition steps. Our objective, as defined in Prop. 1, is to choose $(\boldsymbol{x}, t) \in D_P$ to minimize the expected model-based PEHE, $\mathbb{E}_{s+1}\left[\epsilon_{\text{PEHE}}^{\mathcal{M}}(\hat{\tau}_{s+1})\right]$. We use $\mathbb{E}_{s+1}[\cdot]$ to denote the pre-posterior expectation $\mathbb{E}_{y \sim p(y|\boldsymbol{x}, t, \mathcal{F}_s)}[\cdot]$.*

$$\arg\min_{(\boldsymbol{x}, t) \in D_P} \mathbb{E}_{s+1}\left[\epsilon_{\text{PEHE}}^{\mathcal{M}}(\hat{\tau}_{s+1})\right] \tag{59}$$

*The model-based error $\epsilon_{\text{PEHE}}^{\mathcal{M}}$ measures the squared error against the oracle posterior mean $\hat{\tau}_{\Omega}(\boldsymbol{x})$ (i.e., the posterior mean given all data, $\mathbb{E}[\tau(\boldsymbol{x}) \mid D_P]$).*

$$\mathbb{E}_{s+1}\left[\epsilon_{\text{PEHE}}^{\mathcal{M}}(\hat{\tau}_{s+1})\right] = \mathbb{E}_{s+1}\left[\mathbb{E}_{p_{\text{tar}}(\boldsymbol{x})}\left[\left(\hat{\tau}_{s+1}(\boldsymbol{x}) - \hat{\tau}_{\Omega}(\boldsymbol{x})\right)^2\right]\right] \tag{60}$$

$$= \mathbb{E}_{p_{\text{tar}}(\boldsymbol{x})}\left[\mathbb{E}_{s+1}\left[\left(\hat{\tau}_{s+1}(\boldsymbol{x}) - \hat{\tau}_{\Omega}(\boldsymbol{x})\right)^2\right]\right] \qquad \text{[via Fubini's theorem]}$$

*The inner term $\mathbb{E}_{s+1}\left[(\hat{\tau}_{s+1} - \hat{\tau}_{\Omega})^2\right]$ is the expected squared error of the (Bayesian) oracle estimator. This can be decomposed using the law of total variance. This objective simplifies to minimizing the posterior variance of the oracle estimator $\hat{\tau}_{\Omega}(\boldsymbol{x})$:*

$$\mathbb{E}_{s+1}\left[(\hat{\tau}_{s+1}(\boldsymbol{x}) - \hat{\tau}_{\Omega}(\boldsymbol{x}))^2\right] = \mathbb{E}_{s+1}\left[\mathbb{V}\text{ar}_{s+1}[\hat{\tau}_{\Omega}(\boldsymbol{x})]\right] \tag{61}$$

*We then apply variance decomposition to the term $\mathbb{V}\text{ar}_{s+1}[\hat{\tau}_{\Omega}(\boldsymbol{x})]$:*

$$\mathbb{V}\text{ar}_{s+1}[\hat{\tau}_{\Omega}(\boldsymbol{x})] = \mathbb{V}\text{ar}_{s+1}[\tau(\boldsymbol{x})] - \mathbb{V}\text{ar}_{\Omega}[\tau(\boldsymbol{x})] \tag{62}$$

*Here, $\mathbb{V}\text{ar}_{\Omega}[\tau(\boldsymbol{x})]$ is the (constant) oracle variance after seeing all data. Substituting this back, our objective (Eq. 59) becomes:*

$$\arg\min_{(\boldsymbol{x}, t) \in D_P} \mathbb{E}_{p_{\text{tar}}(\boldsymbol{x})}\left[\mathbb{E}_{s+1}\left[\mathbb{V}\text{ar}_{s+1}[\tau(\boldsymbol{x})] - \mathbb{V}\text{ar}_{\Omega}[\tau(\boldsymbol{x})]\right]\right] \tag{63}$$

*Since $\mathbb{V}\text{ar}_{\Omega}[\tau(\boldsymbol{x})]$ is a constant term that does not depend on the choice of $(\boldsymbol{x}, t)$, it can be dropped from the minimization:*

$$\arg\min_{(\boldsymbol{x}, t) \in D_P} \mathbb{E}_{p_{\text{tar}}(\boldsymbol{x})}\left[\mathbb{E}_{s+1}\left[\mathbb{V}\text{ar}_{s+1}[\tau(\boldsymbol{x})]\right]\right] \tag{64}$$

*For a GP, the posterior variance $\mathbb{V}\text{ar}_{s+1}[\cdot]$ is a deterministic function of the query point $(\boldsymbol{x}, t)$ and the existing data $D_T$. It does not depend on the (random) future outcome $y$. Therefore, $\mathbb{V}\text{ar}_{s+1}[\tau(\boldsymbol{x})]$ is a constant with respect to the expectation $\mathbb{E}_{s+1}$ (which is $\mathbb{E}_y$). Applying this to Eq. 64, the inner expectation $\mathbb{E}_{s+1}\left[\mathbb{V}\text{ar}_{s+1}[\tau(\boldsymbol{x})]\right]$ simplifies to just $\mathbb{V}\text{ar}_{s+1}[\tau(\boldsymbol{x})]$. The objective thus simplifies to:*

$$\arg\min_{(\boldsymbol{x}, t) \in D_P} \mathbb{E}_{p_{\text{tar}}(\boldsymbol{x})}\left[\mathbb{V}\text{ar}_{s+1}[\tau(\boldsymbol{x})]\right] \tag{65}$$

*Finally, we expand the variance term using its definition $\tau(\boldsymbol{x}) = f(\boldsymbol{x}, 1) - f(\boldsymbol{x}, 0)$.*

### G.6.1 CONNECTION TO THE CAUSAL-EPIG FRAMEWORK

Prop. 1 shows that under the GP and model-based error assumptions, the optimization target simplifies to minimizing the integrated posterior CATE variance. The Causal-EPIG framework aligns with the objective from Prop. 1 by providing two principled strategies that correctly account for the joint posterior structure. Both strategies maximize a proxy objective based on mutual information, which is equivalent to maximizing the expected reduction in posterior entropy.

**Causal-EPIG-$\mu$ (Comprehensive Strategy)**    This strategy maximizes the joint information gain:

$$\arg\max \mathbb{E}_{p_{\text{tar}}(\boldsymbol{x}^*)}\Big[\text{I}\big(y; (f(\boldsymbol{x}^*, 0), f(\boldsymbol{x}^*, 1))\big)\Big]. \tag{66}$$

This is equivalent to maximizing the expected reduction in the joint entropy $\text{H}(f(\boldsymbol{x}^*, 0), f(\boldsymbol{x}^*, 1))$. For Gaussian processes, $\text{H} \propto \log(\det(\Sigma))$. Since $\det(\Sigma)$ depends on the covariance term, this objective directly targets the full joint uncertainty. It aligns with Prop. 1 by addressing the underlying components from which $\mathbb{V}\text{ar}[\tau]$ is constructed.

**Causal-EPIG-$\tau$ (Focused Strategy)**    This strategy maximizes the information gain of the estimand itself:

$$\arg\max \mathbb{E}_{p_{\text{tar}}(\boldsymbol{x}^*)}\Big[\text{I}\big(y; \tau(\boldsymbol{x}^*)\big)\Big]. \tag{67}$$

This is equivalent to minimizing the expected posterior CATE entropy, $\arg\min \mathbb{E}_{p_{\text{tar}}(\boldsymbol{x}^*)}\big[\mathbb{E}_y[\text{H}_{s+1}(\tau(\boldsymbol{x}^*))]\big]$. For Gaussian posteriors, $\text{H}(\tau) \propto \log(\mathbb{V}\text{ar}[\tau])$. Under the GP assumption, $\mathbb{V}\text{ar}_{s+1}$ is deterministic w.r.t. $y$, so $\mathbb{E}_y[\log(\mathbb{V}\text{ar}_{s+1})] = \log(\mathbb{V}\text{ar}_{s+1})$. The objective for Causal-EPIG-$\tau$ therefore simplifies to:

$$\arg\min_{(\boldsymbol{x}, t) \in \mathcal{D}_P} \mathbb{E}_{p_{\text{tar}}(\boldsymbol{x})}\big[\log(\mathbb{V}\text{ar}_{s+1}[\tau(\boldsymbol{x})])\big]. \tag{68}$$

This is a principled proxy for the objective from Prop. 1, which targets the variance $\mathbb{V}\text{ar}_{s+1}$ itself, not its logarithm. While not mathematically equivalent (by Jensen's inequality), it is a closely related information-theoretic criterion that minimizes a measure of posterior uncertainty. This strategy aligns with Prop. 1 by targeting the final quantity of interest. Both Causal-EPIG strategies are principled and aligned with the true CATE optimization objective derived from Prop. 1. They provide a trade-off between targeting the full causal mechanism (Causal-EPIG-$\mu$) and directly targeting the final estimand (Causal-EPIG-$\tau$).

### G.7 EFFICIENCY COMPARISONS BETWEEN CAUSAL-EPIG AND CAUSAL-BALD

**Proposition 2** *Assume that, for any target covariate $\boldsymbol{x}^*$, both the potential outcomes $(y^*(0), y^*(1))$ and the CATE $\tau(\boldsymbol{x}^*)$ are deterministic functions of the model parameters (or corresponding parameter subsets). Then the parameter-oriented information utilities upper-bound their prediction-oriented counterparts:*

$$(a) \quad \text{Causal-BALD} - \mu(\boldsymbol{x}, t) \geq \text{Causal-EPIG} - \mu(\boldsymbol{x}, t), \tag{69}$$

$$(b) \quad \text{Causal-EIG}(\boldsymbol{x}, t) \geq \text{Causal-EPIG} - \tau(\boldsymbol{x}, t). \tag{70}$$

Both inequalities follow from the Data Processing Inequality (DPI). Equality holds iff the prediction quantity $(y^*(0), y^*(1))$ or $\tau(\boldsymbol{x}^*)$ is a sufficient statistic for the corresponding parameter; otherwise the inequalities are strict.

**Proof 2** *Let the updated training dataset be $D'_T = D_T \cup \{(\boldsymbol{x}, t)\}$. The four utility function can be written as*

$$\text{Causal-BALD} - \mu(\boldsymbol{x}, t) \coloneqq \text{I}\big(y; \theta \mid D'_T\big),$$

$$\text{Causal-EPIG} - \mu(\boldsymbol{x}, t) \coloneqq \mathbb{E}_{p_{\text{tar}}(\boldsymbol{x}^*)}\Big[\text{I}\big(y; (y^*(0), y^*(1)) \mid \boldsymbol{x}^*, D'_T\big)\Big],$$

$$\text{Causal-EIG}(\boldsymbol{x}, t) \coloneqq \text{I}\big(y; \theta_\tau \mid D'_T\big),$$

$$\text{Causal-EPIG} - \tau(\boldsymbol{x}, t) \coloneqq \mathbb{E}_{p_{\text{tar}}(\boldsymbol{x}^*)}\Big[\text{I}\big(y; \tau(\boldsymbol{x}^*) \mid \boldsymbol{x}^*, D'_T\big)\Big].$$

*(a) **Proof of** Causal-BALD $-\mu \geq$ Causal-EPIG $-\mu$. Since $(y^*(0), y^*(1))$ is a deterministic function of $\theta$, conditioning on $\{\boldsymbol{x}^*, D_T'\}$ yields the Markov chain $y \to \theta \to (y^*(0), y^*(1))$. By the data processing inequality, we have*

$$\mathrm{I}\big(y; \theta \mid \boldsymbol{x}^*, D_T'\big) \geq \mathrm{I}\big(y; (y^*(0), y^*(1)) \mid \boldsymbol{x}^*, D_T'\big). \tag{71}$$

*The left-hand side is independent of $\boldsymbol{x}^*$:*

$$\mathrm{I}\big(y; \theta \mid \boldsymbol{x}^*, D_T'\big) = \mathrm{I}\big(y; \theta \mid D_T'\big) = \text{Causal-BALD} - \mu(\boldsymbol{x}, t). \tag{72}$$

*Taking expectation over $\boldsymbol{x}^* \sim p_{\text{tar}}$ gives*

$$\text{Causal-BALD} - \mu(\boldsymbol{x}, t) \geq \text{Causal-EPIG} - \mu(\boldsymbol{x}, t). \tag{73}$$

*(b) **Proof of** Causal-EIG $\geq$ Causal-EPIG $-\tau$.*

*Since $\tau(\boldsymbol{x}^*)$ is a deterministic function of $\theta_\tau$, conditioning on $\{\boldsymbol{x}^*, D_T'\}$ gives the Markov chain $y \to \theta_\tau \to \tau(\boldsymbol{x}^*)$. By the data processing inequality, we have*

$$\mathrm{I}\big(y; \theta_\tau \mid \boldsymbol{x}^*, D_T'\big) \geq \mathrm{I}\big(y; \tau(\boldsymbol{x}^*) \mid \boldsymbol{x}^*, D_T'\big). \tag{74}$$

*Again, the LHS does not depend on $\boldsymbol{x}^*$:*

$$\mathrm{I}\big(y; \theta_\tau \mid \boldsymbol{x}^*, D_T'\big) = \mathrm{I}\big(y; \theta_\tau \mid D_T'\big) = \text{Causal-EIG}(\boldsymbol{x}, t). \tag{75}$$

*Taking expectation over $\boldsymbol{x}^* \sim p_{\text{tar}}$ yields*

$$\text{Causal-EIG}(\boldsymbol{x}, t) \geq \text{Causal-EPIG} - \tau(\boldsymbol{x}, t). \tag{76}$$

# H FURTHER EXPERIMENTAL RESULTS

In this section, we provide a comprehensive set of supplementary experimental results to complement our main findings. First, we present additional performance curves and detailed metrics for our primary experiments on the synthetic (Hahn, Causal-BALD) and semi-synthetic (IHDP, ACTG-175) benchmarks. Second, we conduct a series of ablation studies to analyze the robustness of our Causal-EPIG framework. These studies, conducted primarily on the Hahn simulation dataset, evaluate the impact of varying initial random starts, acquisition batch sizes, and the size of the unlabeled pool.

**Analysis of CausalBALD Dataset.** The results on the CausalBALD synthetic dataset, presented in Fig. 12 (regular setup) and Fig. 13 (shift setup), demonstrate the effectiveness of the Causal-EPIG framework, which shows the top-tier performance compard with all baseline methods.

## H.1 CAUSALBLAD DATASET

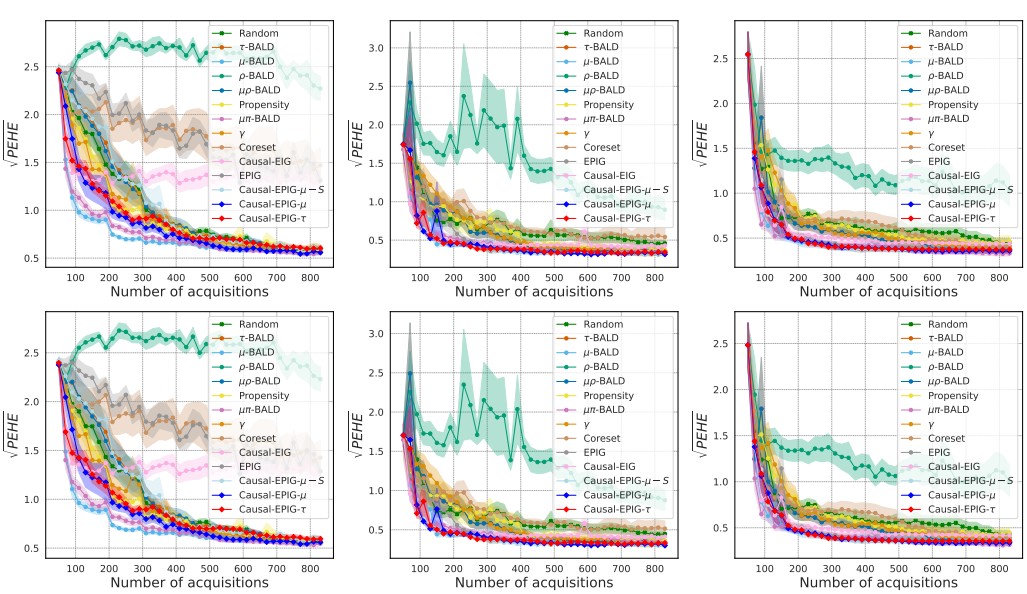

Figure 12: Performance comparison on the CausalBALD synthetic dataset with the regular setup. Each plot shows the $\sqrt{\text{PEHE}}$ (lower is better) as a function of the number of acquired samples. Rows distinguish between the training performance (top) and the testing performance (bottom). Columns correspond to the three different underlying CATE estimators: BCF, CMGP, and NSGP.

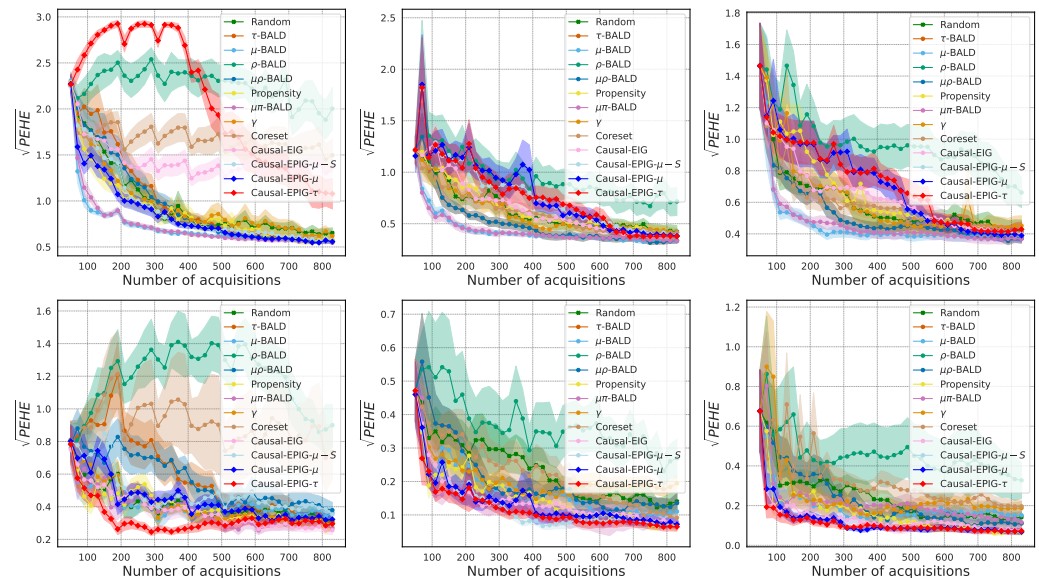

Figure 13: Performance comparison on the CausalBALD synthetic dataset with the target distribution shift setup. Each plot shows the $\sqrt{\text{PEHE}}$ (lower is better) as a function of the number of acquired samples. Rows distinguish between the training performance (top) and the testing performance (bottom). Columns correspond to the three different underlying CATE estimators: BCF, CMGP, and NSGP.

## H.2 HAHN DATASET

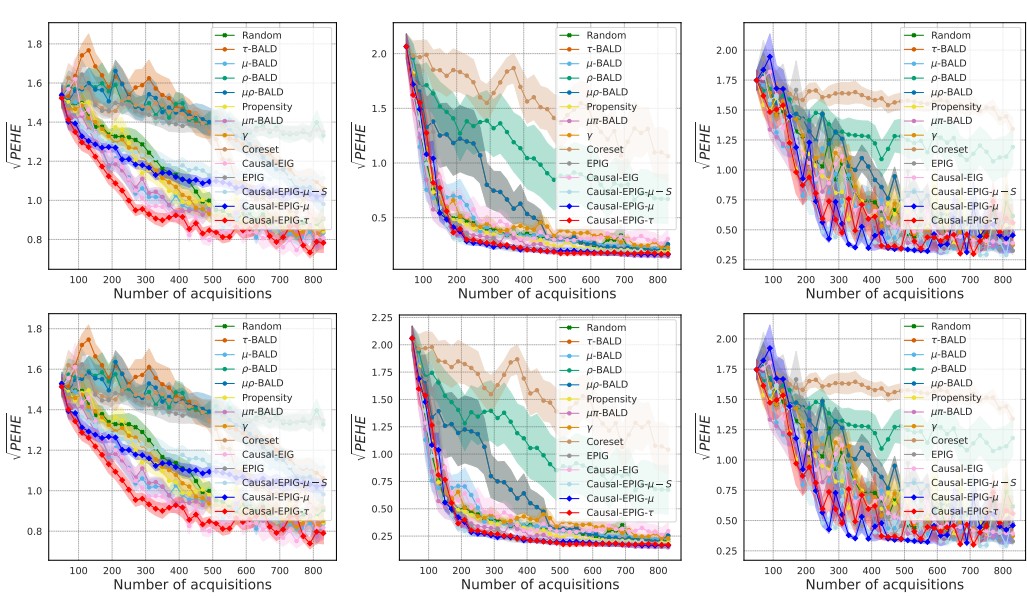

Figure 14: Performance comparison on the Hahn (linear function) synthetic dataset with the regular setup. Each plot shows the $\sqrt{\text{PEHE}}$ (lower is better) as a function of the number of acquired samples. Rows distinguish between the training performance (top) and the testing performance (bottom). Columns correspond to the three different underlying CATE estimators: BCF, CMGP, and NSGP.

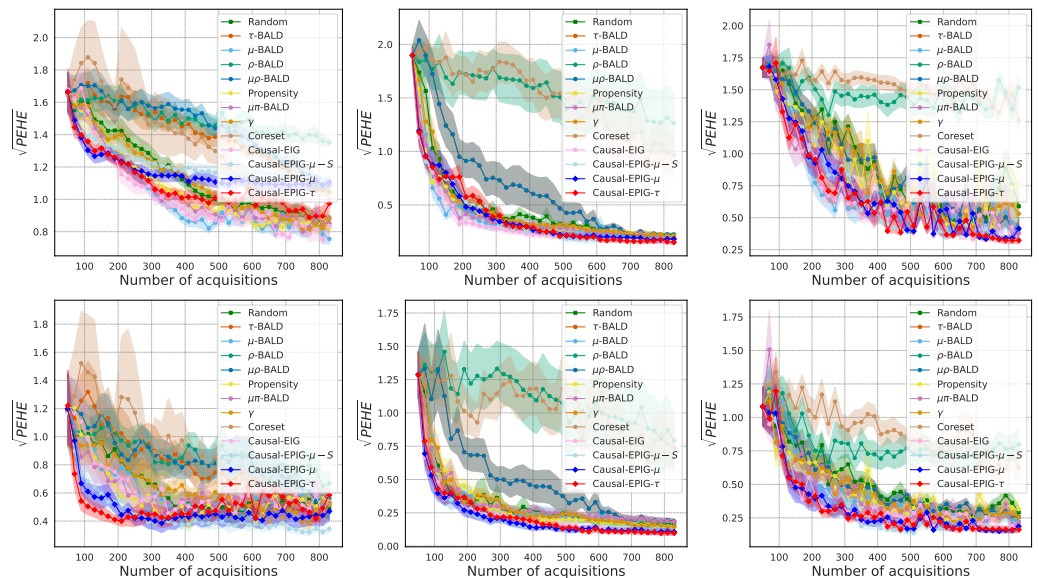

Figure 15: Performance comparison on the Hahn (linear function) synthetic dataset with the target distribution shift setup. Each plot shows the $\sqrt{\text{PEHE}}$ (lower is better) as a function of the number of acquired samples. Rows distinguish between the training performance (top) and the testing performance (bottom). Columns correspond to the three different underlying CATE estimators: BCF, CMGP, and NSGP.

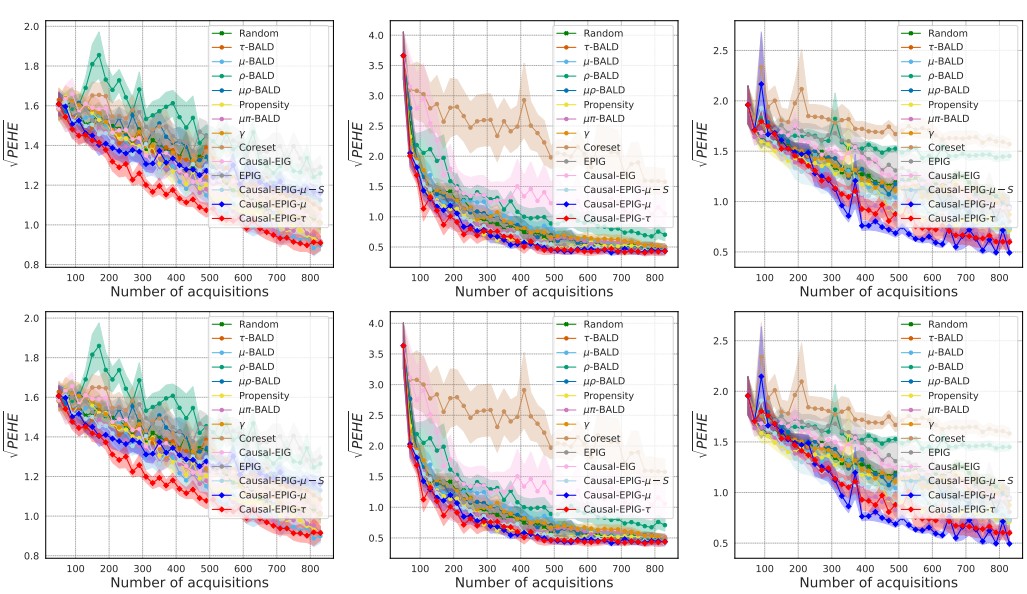

Figure 16: Performance comparison on the Hahn (nonlinear function) synthetic dataset with regular. Each plot shows the $\sqrt{\text{PEHE}}$ (lower is better) as a function of the number of acquired samples. Rows distinguish between the training performance (top) and the testing performance (bottom). Columns correspond to the three different underlying CATE estimators: BCF, CMGP, and NSGP.

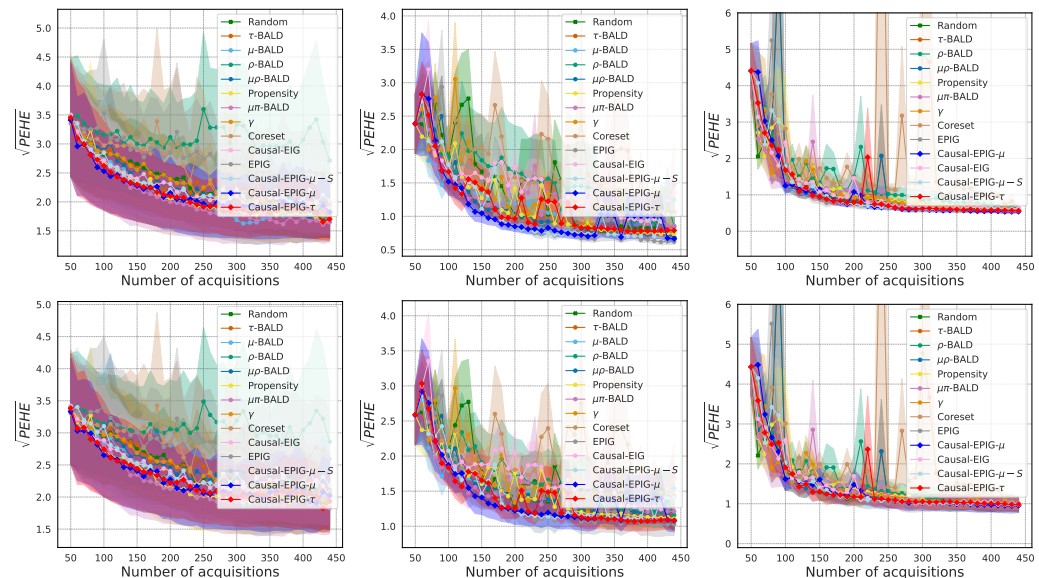

Figure 18: Performance comparison on the IHDP semi-synthetic dataset with the regular setup. Each plot shows the $\sqrt{\text{PEHE}}$ (lower is better) as a function of the number of acquired samples. Rows distinguish between the training performance (top) and the testing performance (bottom). Columns correspond to the three different underlying CATE estimators: BCF, CMGP, and NSGP.

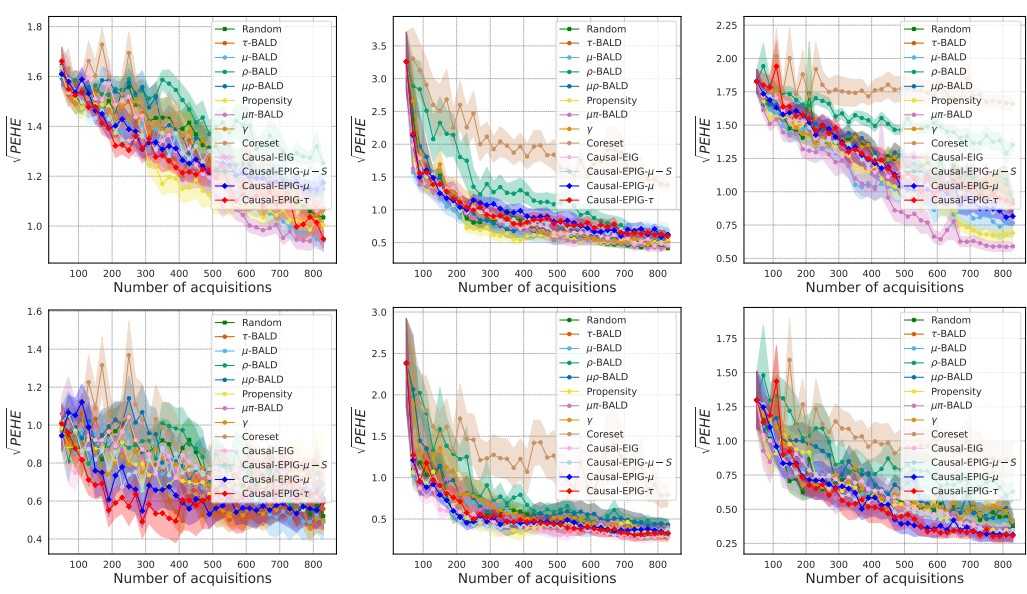

Figure 17: Performance comparison on the Hahn (nonlinear function) synthetic dataset with the target distribution shift setup. Each plot shows the $\sqrt{\text{PEHE}}$ (lower is better) as a function of the number of acquired samples. Rows distinguish between the training performance (top) and the testing performance (bottom). Columns correspond to the three different underlying CATE estimators: BCF, CMGP, and NSGP.

## H.3   IHDP DATASET

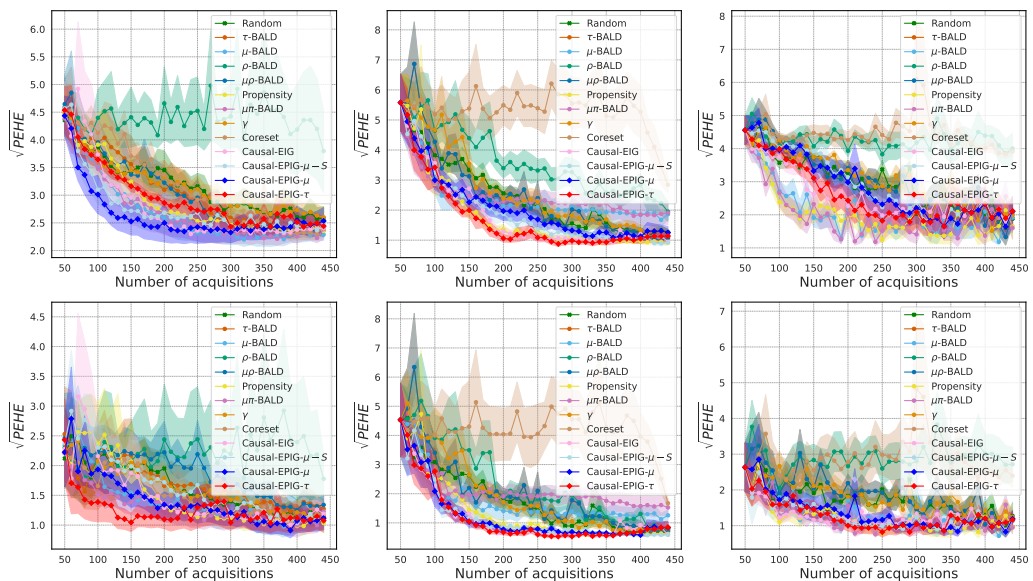

Figure 19: Performance comparison on the IHDP semi-synthetic dataset with the target distribution shift setup. Each plot shows the $\sqrt{\text{PEHE}}$ (lower is better) as a function of the number of acquired samples. Rows distinguish between the training performance (top) and the testing performance (bottom). Columns correspond to the three different underlying CATE estimators: BCF, CMGP, and NSGP.

## H.4   ACTG-175 DATASET

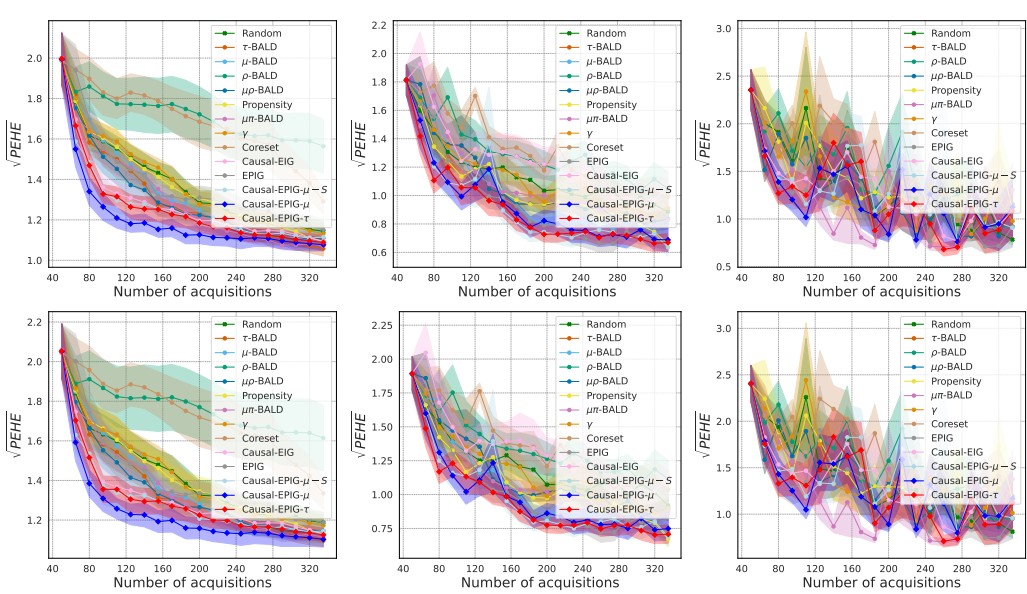

Figure 20: Performance comparison on the ACTG-175 semi-synthetic dataset with the standard setup. Each plot shows the $\sqrt{\text{PEHE}}$ (lower is better) as a function of the number of acquired samples. Rows distinguish between the training performance (top) and the testing performance (bottom). Columns correspond to the three different underlying CATE estimators: BCF, CMGP, and NSGP.

## H.5 ABLATION STUDIES

### H.5.1 DIFFERENT STARTING POINTS

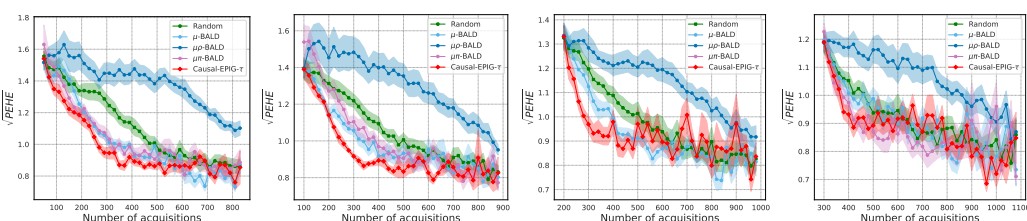

Figure 21: Ablation study on the impact of the **warm-start size**. Performance ($\sqrt{\text{PEHE}}$) is evaluated on the Hahn (linear) dataset using the BCF base estimator. Each panel shows the result for a different number of initial random samples used for the warm-start. From left to right: 50, 100, 200, and 300 initial samples.

**Ablation Study: Sensitivity to Warm-Start Size.**   To assess the robustness of our method to the size of the initial random batch, we conduct an ablation study on the warm-start phase. We vary the number of initial samples from 50 to 300 on the Hahn (linear) dataset with the BCF estimator, with results shown in Fig. 21. The key finding is that the superior performance of **Causal-EPIG** is robust to the choice of the warm-start size. Across all four settings, our method consistently outperforms the included baselines, establishing a clear performance advantage early in the acquisition process and maintaining it. While a larger warm-start set leads to a better initial model and lower starting PEHE for all methods, the relative performance ranking remains unchanged. This demonstrates that the effectiveness of our acquisition strategy is not highly sensitive to this hyperparameter, highlighting its practical stability.

### H.5.2 DIFFERENT POOL SIZES

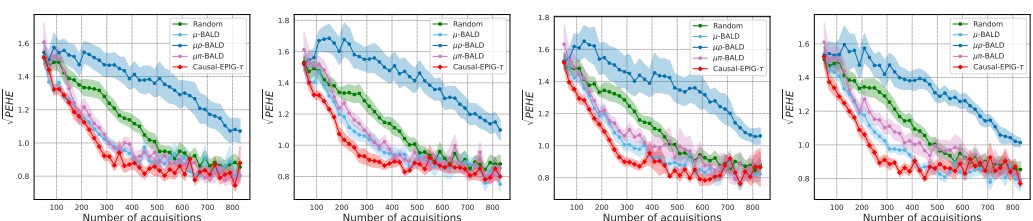

Figure 22: Ablation study on the impact of the **unlabeled pool size**. Performance ($\sqrt{\text{PEHE}}$) is evaluated on the Hahn (linear) dataset using the BCF base estimator. Each panel shows the result for a different initial size of the unlabeled pool $D_P$. From left to right: $|D_P| = 1000, 1500, 2000$, and 2500.

**Ablation Study: Sensitivity to Pool Size.**   We investigate the sensitivity of our method to the size of the unlabeled pool from which candidates are selected. In Fig. 22, we vary the initial pool size $|D_P|$ from 1000 to 2500, while keeping the dataset and base model fixed. The results clearly show that the performance advantage of **Causal-EPIG** is robust across different pool sizes. In all four configurations, our method consistently and significantly outperforms the baselines. We observe that a larger pool provides a modest performance benefit to all active methods, including our own, as it increases the diversity of candidates available for selection. Crucially, however, the relative performance ordering remains stable, and the superiority of **Causal-EPIG** holds regardless of the pool size. This study demonstrates that our target-aware selection strategy is a fundamental advantage, not an artifact of a specific data environment.

### H.5.3 DIFFERENT STEP SIZES

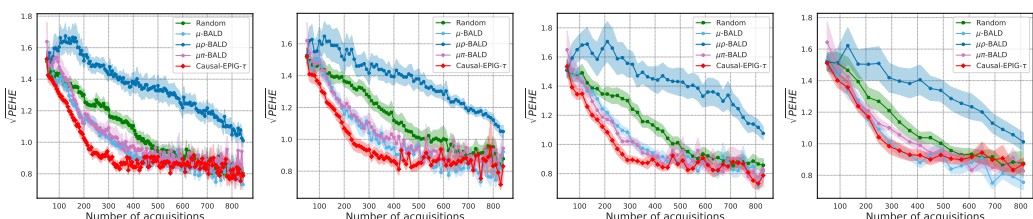

Figure 23: Ablation study on the impact of the **acquisition batch size** ($n_b$). Performance ($\sqrt{\text{PEHE}}$) is evaluated on the Hahn (linear) dataset using the BCF base estimator. Each panel shows the result for a different number of samples acquired per round. From left to right: $n_b = 5, 10, 20$, and $40$.

**Ablation Study: Sensitivity to Batch Size.** Finally, we analyze the effect of the acquisition batch size, $n_b$, a key hyperparameter in the active learning loop. Fig. 23 shows the performance as we vary the number of samples acquired per round from 5 to 40. The primary finding is that **Causal-EPIG** consistently outperforms all baselines across every batch size tested, demonstrating its robust superiority regardless of this hyperparameter choice. We also observe a trend common in active learning: smaller, more frequent acquisition batches (e.g., $n_b = 5$) tend to yield slightly better final performance for all active methods. This is because more frequent model updates allow for more responsive and adaptive sample selection. Nevertheless, the relative performance advantage of **Causal-EPIG** is maintained across all settings, confirming the robustness of our approach.

### H.5.4 DUE ESTIMATOR

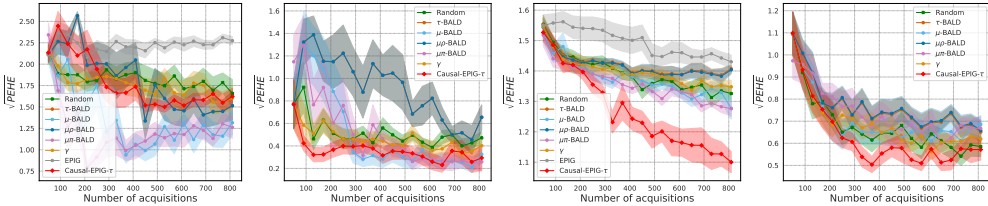

Figure 24: Performance comparison using the **Deep Variational GP estimator** from the original CausalBALD study (Jesson et al., 2021). Each panel shows the $\sqrt{\text{PEHE}}$ on a different dataset. From left to right: CausalBALD, CuaslBALD with distribution shift, Hahn (linear), Hahn (linear) with distribution shift.

**Analysis with the DeepGP Base Estimator.** To ensure a direct and fair comparison with the original CausalBALD study, we conduct a final experiment using the specific Deep Variational GP estimator proposed in their work. The results across our four benchmark datasets (in the standard setting) are shown in Fig. 24. The findings are remarkably consistent with our main results. **Causal-EPIG** demonstrates robustly superior performance across the diverse set of datasets. It achieves the fastest error reduction on the CausalBALD and IHDP benchmarks and shows a clear advantage on the Hahn (linear) dataset. While the Hahn (non-linear) setting proves challenging for all methods when paired with this estimator, **Causal-EPIG** remains a top-tier performer. This provides compelling evidence that the effectiveness of **Causal-EPIG** is not tied to a specific model architecture (such as the standard GPs or BCF used in our main experiments). Its principled, target-aware design provides significant performance gains across a variety of Bayesian CATE estimators, confirming its flexibility and general applicability.

