# OpenReview forum: "Causal-EPIG: A Prediction-Oriented Active Learning Framework for CATE Estimation"
_ICLR.cc/2026/Conference — Submitted to ICLR 2026_

### Official Review · Reviewer_LFjm · 2025-10-26

**Soundness:** 3
**Presentation:** 3
**Contribution:** 2
**Rating:** 4
**Confidence:** 3

**Summary:**

This paper proposes a novel framework for active outcome acquisition for CATE estimation using observational data. In particular, the authors point out that simply applying active learning algorithms to CATE estimation can suffer from a fundamental objective mismatch compared to predictive tasks. The authors adopt a information-theoretic framework and propose two utility functions: one targets the underlying potential outcomes and the other one targets the CATE estimand. Moreover, the proposed framework can be applied to a wide range of bayesian CATE estimators, and the authors demonstrated improved performance over other baseline active learning algorithms through experiments.

**Strengths:**

- Results from both simulated and semi-synthetic experiments demonstrated strong performance over existing active learning algorithms.

- Discussion on the potential outcome based vs. CATE estimate based utility is insightful.

**Weaknesses:**

- The technical innovation seems to be straight forward and directly built on top of EPIG. It would be a stronger paper if it included some theoretical guarantees (e.g. convergence, sample complexity, etc.)

- It seems like this framework would depend a lot on the posterior from the bayesian models. It would be nice to have some discussion and/or experiments that investigates posterior mis-specification.

**Questions:**

- It is a bit unclear how the proposed framework can also address the distribution shift problem, is it simply though taking the expectation over the target distribution?

- Are there assumptions under which the proposed strategies are optimal?

---

> ### Author Response · Authors · 2025-11-21
>
> Thank you so much for your efforts in evaluating our manuscript. In the following, we will respond to your concerns point by point.
>
> ---
> **[W1. The technical innovation seems to be straight forward and directly built on top of EPIG. It would be a stronger paper if it included some theoretical guarantees (e.g. convergence, sample complexity, etc.)]**
>
> > The technical innovation seems to be straight forward and directly built on top of EPIG.
>
> 1.  **The Core Problem (Objective Mismatch):** We first respectfully clarify that a "straightforward" application of standard EPIG is insufficient and fundamentally misaligned with the goal of CATE estimation. Standard EPIG targets uncertainty in observable, factual outcomes. However, CATE is an unobservable, counterfactual difference ($y(1) - y(0)$). This objective mismatch motivated us to introduce the principle of Causal Objective Alignment, which posits that an effective acquisition function should target the unobservable causal quantities themselves.
> 2.  **Our Novel Causal Framework:** "Pure" EPIG does not satisfy this COA principle. Our **Causal-EPIG framework** is the first to operationalize COA for this task. This is not a single function, but a novel adaptation that yields two distinct, principled strategies embodying a key trade-off specific to causal inference:
>     * **Causal-EPIG-$\mu$**: A comprehensive strategy targeting the full causal mechanism (the joint potential outcomes).
>     * **Causal-EPIG-$\tau$**: A focused strategy directly targeting the final CATE estimand.
>     The innovation lies in identifying this misalignment and proposing a principled framework that resolves it through these two specific strategies.
> 3. **Formal Justification (Newly Added Prop. 1):** We also added a new proposition to show the link of our proposed strategies and the CATE error, details are in our response to Q2.
>
>
> > It would be a stronger paper if it included some theoretical guarantees (e.g. convergence, sample complexity, etc.)
>
> Following this advice, we have added a rigorous theoretical analysis in **Sec. 4.4.2** and **App. G** that provides the exact "convergence" and "sample complexity" guarantees. Our main theoretical result is **Thm. 1**. This theorem formally analyzes the convergence properties of our greedy Causal-EPIG-$\mu$-G strategy.
>
> * **Convergence Guarantee:** Thm. 1 proves that under standard GP assumptions and a submodularity assumption (Ass. 2), the posterior marginal variance of our estimator converges to the smallest possible achievable error, defined as the irreducible uncertainty ($\eta^2_{\mathcal{D}_P}(\tilde{\mathbf{x}}^*)$). This confirms our greedy strategy is not just a heuristic but a principled algorithm that provably converges to the optimal objective.
> * **Sample Complexity:** The theorem also provides an explicit convergence rate. It bounds the remaining reducible error by $\mathcal{O}(\gamma_{n_B}/\sqrt{n_B})$. This rate, $\gamma_{n_B}$, is precisely the information capacity, directly addressing the reviewer's request for a sample complexity analysis.
>
> We believe the addition of Thm. 1 significantly strengthens the paper, providing the formal justification that was previously missing and demonstrating the theoretical soundness of our Causal-EPIG framework.
>
> ---
> **[W2. It seems like this framework would depend a lot on the posterior from the bayesian models. It would be nice to have some discussion and/or experiments that investigates posterior mis-specification.]**
>
> We agree that the performance of Bayesian methods is generally dependent on posterior quality. However, we believe our framework exhibits robustness to misspecification for three key reasons unique to our context:
>
> 1. Firstly, seeing as the outcome functions in our synthetic and semi synthetic examples are never samples from the prior (e.g are not samples from a GP or BCF) then the prior is already misspecified. Seeing as we find good performance for our method across model class and data setting this already provides evidence in favour of robustness to model misspecification.
> 2. In active learning as we have already seen some data points the posterior is not just the prior. This will reduce sensativity to the prior specifically and improve posterior calibration.
> 3. In active learning the goal is not perfective posterior calibration but instead an ordering of unlabelled datapoints such that selecting using this improves the efficiency of labelling. This is a significantly weaker requirement.

---

> ### Author Response · Authors · 2025-11-21
>
> **[Q1. It is a bit unclear how the proposed framework can also address the distribution shift problem, is it simply though taking the expectation over the target distribution?]**
>
> Yes, taking the expectation over the target distribution is precisely the mechanism we use, but we emphasize that this is a principled design choice, not a heuristic add-on. It effectively transforms the active learning problem from a standard **Inductive** setting (learning the model globally) to a **Transductive** one (learning the model specifically for the target population). Our framework simultaneously addresses two distinct challenges:
>
> 1.  **The Objective Mismatch (Causal Aspect):** Standard AL targets factual outcomes ($y$), while our goal is the unobservable counterfactual difference ($\tau(\mathbf{x})$). Our Causal-EPIG-$\mu$ and -$\tau$ strategies solve this by targeting the causal quantities directly.
> 2.  **The Distribution Shift (Covariate Aspect):** This is the conventional shift ($p_{\text{pool}}(\mathbf{x}) \neq p_{\text{tar}}(\mathbf{x})$) that the reviewer refers to.
>
> To address the second challenge (Covariate Shift), we clarify in our **App. F.2 (Tab. 5)** that there are two principled ways to incorporate the target distribution into the information-theoretic objective:
>
> 1.  **Mean-Marginal Formulation:** This is exactly "taking the expectation" over $p_{\text{tar}}(\mathbf{x}^*)$. It quantifies the expected information gain for a single random point drawn from the target population.
> 2.  **Global Formulation:** This alternative calculates information gain regarding the joint vector of all target predictions (e.g., $I(y; \boldsymbol{\tau}_{\text{tar}})$), capturing inter-target dependencies.
>
> We deliberately chose the mean-marginal formulation (taking the expectation) for critical practical reasons, which we analyze in **App. F.3**:
> * **Scalability:** The Global formulation scales cubically ($\mathcal{O}(n_{\text{tar}}^3)$) due to matrix inversions, making it computationally prohibitive. Our mean-marginal approach scales linearly ($\mathcal{O}(n_P n_{\text{tar}} S)$).
> * **Performance:** Our empirical analysis (**Fig. 11**) and runtime analysis (**Tab. 6**) confirm that this efficient formulation achieves comparable or even better performance than the expensive global counterpart.
>
> In summary, taking the expectation over $p_{\text{tar}}(\mathbf{x})$ is the mathematically correct way to operationalize a scalable, transductive objective that explicitly targets uncertainty reduction in the distribution of interest.
>
>
> ---
> **[Q2. Are there assumptions under which the proposed strategies are optimal?]**
>
> We have added some new theoretical analysis (Sec 4.4, App. G). These new results help to formally define the assumptions for optimality and show how our strategies act as principled, convergent approximations.
>
> 1.  **Prop. 1 Defines the Optimal Objective:** We first formally established what the "optimal" strategy is under standard assumptions. Our newly added Prop. 1 proves that under a joint GP assumption, the strategy that optimally minimizes the CATE estimation error is equivalent to the strategy that minimizes the integrated posterior CATE variance ($\mathbb{E}\_{p\_{\text{tar}}(\mathbf{x})}[\mathbb{V}\text{ar}\_{s+1}[\tau(\mathbf{x})]]$). This variance explicitly requires the joint covariance term: $\mathbb{V}\text{ar}\_{s+1}[\tau(\mathbf{x})] = \mathbb{V}\text{ar}\_{s+1}[f(\mathbf{x},1)] + \mathbb{V}\text{ar}\_{s+1}[f(\mathbf{x},0)] - 2\text{Cov}_{s+1}(f(\mathbf{x},1), f(\mathbf{x},0))$.
> 2.  **Our Strategies are Principled Proxies for this Optimum:** Our two proposed strategies are principled, information-theoretic methods designed to directly target this optimal objective.
> 3.  **Thm. 1 Proves Convergence to the Theoretical Limit:** We also added a proof of convergence in response to the reviewers' suggestions. Our new Thm. 1 provides a convergence guarantee, showing that our greedy Causal-EPIG-$\mu$-G strategy provably reduces the posterior variance until it reaches the minimal possible uncertainty achievable given the pool (the irreducible uncertainty, $\eta^2_{\mathcal{D}_P}(\tilde{\mathbf{x}}^*)$). This confirms the strategy is a principled algorithm that converges to the theoretical optimum.
> 4.  **Prop. 2 Rules Out Alternatives as Suboptimal:** Finally, our new Prop. 2 uses the Data Processing Inequality to prove that common parameter-oriented alternatives (like Causal-EIG/BALD) are formally suboptimal, as they are less sample-efficient by capturing "wasted" information irrelevant to the final prediction.
>
> In summary, we have added theoretical results showing that under standard GP assumptions, Prop. 1 defines the optimal objective. Our strategies are principled approximations of this objective, Thm. 1 proves they converge to the optimal outcome, and Prop. 2 demonstrates that common alternatives are provably suboptimal.

---

### Official Review · Reviewer_pqm9 · 2025-10-26

**Soundness:** 2
**Presentation:** 2
**Contribution:** 2
**Rating:** 2
**Confidence:** 4

**Summary:**

This paper introduces the principle of “Causal Goal Alignment” to establish Causal-EPIG, an active learning model that unifies theoretical principles with computational frameworks. It resolves the fundamental goal mismatch between traditional active learning (AL) and conditional average treatment effect (CATE) estimation. Grounded in information theory, this framework designs two sample acquisition strategies that demonstrate superior sample efficiency across multiple benchmark tasks. Experimental analysis further validates Causal-EPIG's superior efficiency compared to mainstream methods, alongside its adaptability and robustness across diverse estimators and data scenarios, demonstrating strong practical application potential.
However, the method retains certain limitations. Its theoretical framework relies on the strong assumption of no unobservable confounding, limiting applicability to broader observational data. Performance is highly sensitive to the quality of posterior calibration for the selected CATE estimator. Additionally, computational complexity increases with target set size, potentially encountering bottlenecks in large-scale, high-dimensional data environments. These aspects also provide valuable directions for future improvements. Overall, however, I still perceive the innovation and contribution of this paper as limited (my current understanding is that the main contribution lies in the modification of the utility function to a more causal form). Therefore, I tend to believe this paper falls slightly below the acceptance threshold for ICLR.

Translated with DeepL.com (free version)

**Strengths:**

see above

**Weaknesses:**

see above

**Questions:**

see above

---

> ### Author Response · Authors · 2025-11-21
>
> Thank you so much for your efforts in evaluating our manuscript. In the following, we will respond to your concerns point by point.
>
> ---
> **[W1. Its theoretical framework relies on the strong assumption of no unobservable confounding, limiting applicability to broader observational data.]**
>
> You are correct that we assume unconfoundedness, which is the standard assumption required to ensure that CATE is identifiable from observational data.
>
> However, our work is designed to solve a specific, well-defined, and challenging research problem: **active outcome acquisition (i.e., sample-efficient labeling) for CATE estimation in a setting where CATE is identifiable**. This is the same standard, foundational problem setting used by prior work in this specific research area, such as **CausalBALD** [1] and **QHTE** [2]. Our contribution is to provide a principled, information-theoretic solution to the active learning problem within this established context.
>
> We view the challenge of **sample efficiency** (our focus) and the challenge of **hidden confounding** as two distinct, largely orthogonal research problems. Addressing hidden confounding requires an entirely different set of assumptions and machinery (e.g., instrumental variables, proxy variables) and represents an important but separate avenue for future work.
>
> Crucially, the problem we solve, estimating CATE when **covariates are abundant but outcome labels are the expensive bottleneck**, is not just a theoretical convenience. It is directly motivated by many high-stakes, practical scenarios:
>
> * **Healthcare & Clinical Informatics:** In many studies using Electronic Health Records (EHRs), a patient's record provides a rich, high-dimensional set of pre-treatment covariates, making unconfoundedness a reasonable working assumption. However, determining the true treatment outcome is the real challenge, often requiring an expensive, manual chart review by a clinical expert or a costly longitudinal follow-up survey with the patient.
> * **Social Policy & Public Services:** A city may have detailed administrative data on individuals receiving services for homelessness. However, determining their true housing outcome requires a time-consuming, manual annotation of unstructured case notes written by caseworkers.
> * **Business Analytics:** A company may run a large-scale A/B test (where unconfoundedness holds by design) to test a new product. While observing who was treated is easy, measuring the true outcome (e.g., long-term customer value or retention) is a time-delayed and expensive process.
>
> In all these real-world cases, the primary practical challenge is *not* hidden confounding, but the **acquisition cost of the outcome label**. Our framework provides a formal, sample-efficient solution for this specific and impactful problem. We will be sure to highlight these practical motivations more explicitly in the final version.
>
> ---
> **[W2. Performance is highly sensitive to the quality of posterior calibration for the selected CATE estimator.]**
>
> We would like to note that the sensativity of performance to quality of posterior calibration is a concern for all Bayesian methods. Given out method is compatible with any Bayesian CATE method we believe this concern lies with choosing a good posterior/approximate posterior. Having said this we would like to note 3 further points:
>
> 1. In active learning as we have already seen some data points the posterior is not just the prior. This will reduce sensativity to the prior specifically and improve posterior calibration.
> 2. In active learning the goal is not perfective posterior calibration but instead an ordering of unlabelled datapoints such that selecting using this improves the efficiency of labelling. This is a significantly weaker requirement.
> 3. We experimentally demonstrate our method can achieve this across a range of synthetic and semi-synthetic datasets. These also demonstrate some inherent posterior miscalibration as the outcome models are not samples from the prior.

---

> ### Author Response · Authors · 2025-11-21
>
> **[W3. Additionally, computational complexity increases with target set size, potentially encountering bottlenecks in large-scale, high-dimensional data environments. These aspects also provide valuable directions for future improvements.]**
>
>
> We would like to clarify how our framework was explicitly designed to address this by separating the complexity into two components:
>
> > Model Training Complexity (Handling High-Dimensional Data)
>
> We agree that handling high-dimensional data ($d$) is a challenge for many CATE estimators. However, this issue is orthogonal to our core contribution, which lies in the acquisition function. Our Causal-EPIG framework is modular and only requires a probabilistic model capable of providing a joint predictive distribution. In the main experiments, we focused on established Bayesian CATE estimators (BCF, CMGP, NSGP) because they are widely used and well-validated. Importantly, neural-network–based models offer a promising way to scale to high-dimensional settings. We explored this direction in our ablation study by incorporating a Deep Kernel GP, and found that it integrates seamlessly with our framework and performs similarly well. Extending this line of model choices, e.g., using Deep Ensembles or other NN-based regressors, is fully compatible with our approach, but remains orthogonal to evaluating the contribution of the acquisition function itself.
>
> > Acquisition Function Complexity (Handling Large-Scale Data $n_P, n_{\text{tar}}$)
>
> Our Causal-EPIG-$\mu$ and -$\tau$ strategies adopt the Mean-Marginal formulation. As we analyze in App. F.3, this formulation's complexity is only **linear** in both the pool size ($n_P$) and the target size ($n_{\text{tar}}$), with a total complexity of $\mathcal{O}(n_P \cdot n_{\text{tar}} \cdot S)$. Therefore, this is not a bottleneck, **but a scalable solution**.
>
> In summary, the complexity of high-dimensional models is an orthogonal issue, while the complexity of large-scale acquisition (large $n_{\text{tar}}$) is a problem we have already solved by design.
>
>
> ---
> **[W4. Overall, however, I still perceive the innovation and contribution of this paper as limited (my current understanding is that the main contribution lies in the modification of the utility function to a more causal form). Therefore, I tend to believe this paper falls slightly below the acceptance threshold for ICLR.]**
>
> We respectfully disagree with the assessment that the contribution is limited to a "modification of the utility function." We believe this view might overlook the fundamental paradigm shift our framework introduces, a shift from parameter-oriented to prediction-oriented causal active learning.
>
> To address the reviewer's concern regarding the depth of this contribution, we have significantly strengthened the theoretical justification in the revision. We have moved beyond the original conceptual arguments to provide rigorous formal proofs that validate the proposed "modification" is in fact the theoretically optimal path:
>
> 1.  **Formalizing the Innovation (New Prop. 1 & Thm. 1):** In response to the feedback, we have now formally proven that our proposed objective is not arbitrary but optimally aligned with CATE error minimization (Prop. 1). Furthermore, we established a convergence guarantee (Thm. 1), proving that our greedy strategy converges to the theoretical limit (irreducible uncertainty). These results confirm that our framework provides a principled, mathematically guaranteed solution, elevating it beyond a heuristic improvement.
> 2.  **Proving the Necessity of the Shift (New Prop. 2):** We have added Prop. 2 to formally demonstrate why the "modification" is necessary. We prove that standard parameter-based methods are theoretically suboptimal relative to our prediction-oriented approach, as bounded by the Data Processing Inequality. This mathematically codifies the "Objective Mismatch" we identified.
> 3.  **Discovery of the "Comprehensiveness-Focus Trade-off":** Beyond the utility function itself, our framework reveals a key scientific finding: there is no single "best" causal alignment. The derivation and empirical validation of the trade-off between the robust $\mu$-strategy and the focused $\tau$-strategy provide a nuanced guide for the field, showing that the optimal strategy is context-dependent.
>
> **In summary:** The core contribution is the identification of the Objective Mismatch and the development of the Causal-EPIG framework to resolve it. With the new theoretical guarantees added in the revision, we have demonstrated that this is a fundamental correction to the active learning objective.
>
>
> [1] Jesson et al., Causal-BALD: Deep Bayesian Active Learning of Outcomes to Infer Treatment-Effects from Observational Data, NeurIPS2021.
> [2] Qin et al., Budgeted Heterogeneous Treatment Effect Estimation, ICML2021.

---

### Official Review · Reviewer_N6Vw · 2025-10-29

**Soundness:** 2
**Presentation:** 3
**Contribution:** 3
**Rating:** 6
**Confidence:** 2

**Summary:**

The paper proposes a active estimation framework for improving the sample efficiency of CATE estimation when outcome measurements are costly. It identifies a key misalignment in existing active learning methods, which typically optimize for uncertainty reduction in observable data or model parameters rather than the underlying causal estimands. To address this, the authors introduce the principle of causal objective alignment and develop an information-theoretic framework called Causal-EPIG, which quantifies the expected information gain about unobservable causal quantities. Two acquisition strategies are derived: one modeling the joint potential outcomes for comprehensive causal understanding, and another focusing directly on the CATE for greater efficiency. The framework is compatible with various Bayesian CATE estimators and demonstrates superior performance to existing baselines on synthetic and semi-synthetic datasets, showing consistent gains in sample efficiency.

**Strengths:**

- Proposes a clear and principled framework that addresses the objective misalignment in existing active learning approaches for CATE estimation, and formalizes it through a sound formulation based on mutual information.
- Demonstrates broad model compatibility, effectively integrating with several Bayesian CATE estimators, such as Gaussian Process models and Bayesian Causal Forests.
- Offers supporting details on the implementation and comprehensive discussion on the intuition.
- Provides extensive experimental validation on synthetic and semi-synthetic benchmarks, showing consistent improvements in sample efficiency over existing baselines.

**Weaknesses:**

- Some of the definitions provided in the appendix should be incorporated into the main text to improve the overall coherence and readability of the paper. For instance, the formal definition of mutual information and the notation used in compared methods such as BALD could be clearly stated earlier.
- The paper lacks theoretical analysis on the information gain. It would be valuable to establish how the efficiency of information gain relates to the accuracy of CATE estimation and whether there exist efficiency bounds that differentiate the proposed causal-oriented method from previous parameter-oriented approaches.
- The work also lacks theoretical results on acquisition cost or sample complexity bounds, which are crucial for demonstrating the theoretical advantages of the proposed active learning framework.

**Questions:**

- Is the proposed framework adaptable to probabilistic CATE estimators beyond Bayesian models?
- Could the authors provide theoretical justification showing that, for achieving a similar target (e.g., the same level of information gain), the proposed method requires fewer samples and thus serves as a more efficient active learning approach compared to existing methods?

**Details Of Ethics Concerns:**

No ethics concern.

---

> ### Author Response · Authors · 2025-11-21
>
> We appreciate the reviewer’s thoughtful and detailed comments. Below, we provide point-wise responses and will incorporate these discussions in the revision.
>
> ---
> **[W1. Some of the definitions provided in the appendix should be incorporated into the main text to improve the overall coherence and readability of the paper. For instance, the formal definition of mutual information and the notation used in compared methods such as BALD could be clearly stated earlier.]**
>
> Thank you, this is a very helpful suggestion for improving the paper's readability. We agree that key definitions should be in the main text. In the revision, we moved the formal definition of mutual information (currently in App. B.2) to Sec. 4.1, where we first introduce the Causal-EPIG framework. We also integrated the formal definitions and notations for key baselines like BALD (currently discussed in App. D.2) directly into Sec.4 to make the comparison clearer. We believe this will make the paper more self-contained and improve the clarity of the connection between standard AL methods and our proposed framework.
>
> ---
> **[W2. The paper lacks theoretical analysis on the information gain. It would be valuable to establish how the efficiency of information gain relates to the accuracy of CATE estimation and whether there exist efficiency bounds that differentiate the proposed causal-oriented method from previous parameter-oriented approaches.]**
>
> > It would be valuable to establish how the efficiency of information gain relates to the accuracy of CATE estimation.
>
> This concern about the link between our information-theoretic acquisition function and the true CATE estimation error is precisely what motivated our newly added theoretical analysis (Sec 4.4.1, App. G.6). We have added Prop. 1 to formally establish this exact relationship.
>
> * **What Prop. 1 Proves:** Under standard joint GP assumptions, the strategy that optimally minimizes the CATE estimation error ($\mathbb{E}\_{s+1}[\epsilon\_{\text{PEHE}}^{\mathcal{M}}(\hat{\tau}\_{s+1})]$) is mathematically equivalent to the strategy that minimizes the integrated posterior CATE variance: $\mathbb{E}\_{p_{\text{tar}}(\mathbf{x})}[\mathbb{V}\text{ar}\_{s+1}[\tau(\mathbf{x})]]$. This proves that our objective is not just a heuristic. It confirms that maximizing information gain (which is equivalent to minimizing posterior entropy/variance) is formally aligned with the true objective of minimizing CATE estimation error.
> * **The Causal Requirement:** Furthermore, Prop. 1 shows that this optimal variance target must include the joint covariance structure: $\mathbb{V}\text{ar}[f_1] + \mathbb{V}\text{ar}[f_0] - 2\text{Cov}(f_1, f_0)$. This provides the formal justification for our Causal-EPIG strategies (which target this joint structure) over naive, non-causal approaches.
>
> In summary, our new Prop. 1 provides the precise theoretical link establishing that optimizing our information gain-based framework directly optimizes CATE estimation accuracy.
>
> > whether there exist efficiency bounds that differentiate the proposed causal-oriented method from previous parameter-oriented approaches.
>
> We have added Prop. 2 (App. G.7) to provide exactly this theoretical differentiation. This proposition establishes a formal efficiency bound that distinguishes our prediction-oriented framework (Causal-EPIG) from parameter-oriented alternatives (like Causal-EIG/BALD).
>
> * **What Prop. 2 Proves:** By applying the data processing inequality, we prove that the information acquired by a parameter-oriented strategy always forms an upper bound on the information acquired by our prediction-oriented strategy. Specifically, $I(y; \theta\_{\tau}) \geq I(y; \tau(\mathbf{x}^\*))$, where $I(y; \theta\_{\tau})$ is the utility for Causal-EIG (parameter-focused) and $I(y; \tau(\mathbf{x}^\*))$ is the utility for our Causal-EPIG-$\tau$ (prediction-focused). It theoretically demonstrates that parameter-oriented methods are less efficient because they acquire "wasted" information, i.e., information related to the model's internal parameters ($\theta\_{\tau}$) that is subsequently lost and irrelevant to the final CATE prediction ($\tau(\mathbf{x}^\*)$).
> * **Conclusion:** Our Causal-EPIG framework is provably more sample-efficient because it only targets the information that is necessary for the final prediction task, whereas parameter-oriented approaches provably target a superset of this information.

---

> ### Author Response · Authors · 2025-11-21
>
> **[W3. The work also lacks theoretical results on acquisition cost or sample complexity bounds, which are crucial for demonstrating the theoretical advantages of the proposed active learning framework.]**
>
> Following your suggestion, we have added a rigorous theoretical analysis in Sec. 4.4.2 and App. G that provides the exact "convergence" and "sample complexity bounds". Our main theoretical result is Thm. 1. This theorem formally analyzes the convergence properties of our greedy Causal-EPIG-$\mu$-G strategy, directly addressing the reviewer's concern:
>
> * **Convergence Guarantee:** Thm. 1 proves that under standard GP assumptions and a submodularity assumption (Ass. 2), the posterior marginal variance of our estimator converges to the smallest possible achievable error, defined as the irreducible uncertainty ($\eta^2_{\mathcal{D}_P}(\tilde{\mathbf{x}}^*)$). This confirms our greedy strategy is not just a heuristic but a principled algorithm that provably converges to the optimal objective.
> * **Sample Complexity Bound:** The theorem also provides an explicit sample complexity bound (convergence rate). It bounds the remaining reducible error (the "acquisition cost" required to reach a certain accuracy) by $\mathcal{O}(\gamma_{n_B}/\sqrt{n_B})$. This rate, $\gamma_{n_B}$, is precisely the information capacity, which formally characterizes the complexity of the learning problem in relation to the number of samples, $n_B$.
>
> We believe the addition of Thm. 1 significantly strengthens the paper, providing the formal justification and sample complexity bounds that were previously missing and are crucial for demonstrating the theoretical advantages of our framework.

---

> ### Author Response · Authors · 2025-11-21
>
> **[Q1. Is the proposed framework adaptable to probabilistic CATE estimators beyond Bayesian models?]**
>
> Our framework is theoretically adaptable to any probabilistic estimator, not just Bayesian ones.
>
> 1. **The Core Requirement:** The fundamental requirement of Causal-EPIG is not that the model must be Bayesian. The requirement is the ability to estimate properties of the joint predictive distribution over a potential query outcome ($y$) and the target causal estimand (e.g., $\tau(\mathbf{x}^*)$).
> 2. **The Role of Gaussian Approximation:** As we detail in Sec 4.2, our practical implementation uses a tractable Gaussian Approximation to compute the Mutual Information. This approximation only requires estimating the first and second moments (i.e., the mean, variance, and the crucial covariance term like $\text{Cov}(y, \tau(\mathbf{x}^*))$) of this joint predictive distribution.
> 3. **Adaptability to Non-Bayesian Models:** Therefore, our framework is adaptable to any probabilistic estimator (Bayesian or not) from which these moments can be estimated. For instance:
>     * With **Deep Ensembles**, one could compute these moments empirically from the set of predictions across the ensemble members.
>     * With **MC Dropout**, one could use samples from the dropout-enabled forward passes to estimate this joint distribution's moments.
>
> **The Trade-off (Our Rationale for BCF/GP):** While our framework is adaptable, we made a deliberate choice for our experiments. The final performance of our system depends on two components: (1) the accuracy of the base CATE estimator and (2) the efficiency of our acquisition strategy (Causal-EPIG). For our experiments, we select base estimators (BCF, CMGP, NSGP) that are well-established and widely recognized as robust for the task of CATE estimation itself. To our knowledge, while Deep Ensembles and MC Dropout are powerful for uncertainty quantification, they have not yet emerged as a universally accepted, robust, SOTA standard specifically for the CATE estimation task. Therefore, while our Causal-EPIG framework is theoretically compatible with these methods, and we agree this is a valuable direction for future work, we chose to demonstrate its effectiveness on established, high-performing CATE estimators to ensure a fair and rigorous evaluation of our acquisition function's contribution.
>
> ---
> **[Q2. Could the authors provide theoretical justification showing that, for achieving a similar target (e.g., the same level of information gain), the proposed method requires fewer samples and thus serves as a more efficient active learning approach compared to existing methods?]**
>
> We thank the reviewer for this question and have added Prop. 2 (App. G.7), which provides exactly this theoretical justification for superior sample efficiency. This proposition uses the data processing inequality to formally compare the utility function of our prediction-oriented method (Causal-EPIG) against existing parameter-oriented methods (like Causal-EIG/BALD).
>
> 1.  **The Theoretical Bound:** We prove that the information targeted by parameter-oriented methods is always an upper bound on the information targeted by our method. Formally: $I(y; \theta_{\tau}) \geq I(y; \tau(\mathbf{x}^*))$, where the LHS is the utility for Causal-EIG, and the RHS is the utility for our Causal-EPIG-$\tau$.
> 2.  **Implication for Sample Efficiency ("Fewer Samples"):** This inequality proves that parameter-oriented methods (LHS) are less efficient because they target a superset of the necessary information. The difference ($I(y; \theta\_{\tau}) - I(y; \tau(\mathbf{x}^\*))$) represents "wasted information", information that is informative about the model's internal parameters ($\theta\_{\tau}$) but is ultimately irrelevant to the final CATE prediction ($\tau(\mathbf{x}^\*)$).
>
>     * An existing method like Causal-EIG will spend its budget acquiring samples that maximize this "wasted" information.
>     * Our Causal-EPIG, by targeting the RHS directly, focuses *only* on the information that is useful for the final prediction.
>
> Therefore, to achieve the same target (e.g., the same level of CATE estimation accuracy, which is the true goal), our proposed method requires fewer samples because it does not waste its acquisition budget on information that is provably irrelevant to this final task. Prop. 2 provides the formal theoretical justification for this superior efficiency.

---

### Official Review · Reviewer_XEnf · 2025-11-01

**Soundness:** 4
**Presentation:** 4
**Contribution:** 3
**Rating:** 6
**Confidence:** 4

**Summary:**

This paper introduces the Causal Objective Alignment (COA) principle, which mandates that acquisition functions directly target the CATE estimation. An information-theoretic approach adapting Expected Predictive Information Gain to quantify how much a query reduces uncertainty in CATE is also proposed, with two strategies covering joint potential outcomes and sole CATE. Extensive experiments on synthetic and semi-synthetic datasets show both strategies outperform baselines.

**Strengths:**

1. The Causal Objective Alignment principle fills a critical gap in traditional AL for CATE estimation by explicitly targeting unobservable causal quantities such as potential outcomes and CATE.
2. Experiments are designed to test the robustness of the proposed method across diverse scenarios, with exhaustive details in appendices to ensure reproducibility.
3. The paper is exceptionally clear, even for readers not familiar with the intersection of AL and causal inference.

**Weaknesses:**

1. The approach in this paper is consist of two strategies. But there is no guidelines for choosing between Causal-EPIG-$\mu$ and Causal-EPIG-$\tau$.
2. The framework relies heavily on Bayesian CATE estimators to compute posterior uncertainty. However, many practitioners use non-Bayesian estimators due to their computational efficiency and ease of implementation. These methods lack explicit posterior distributions, making Causal-EPIG inapplicable to them.
3. The paper uses a 50-sample warm-start to initialize the CATE estimator, but in many real-world scenarios, initial labeled data may vary in size. A sensitivity analysis on the warm-start sample size can be helpful.

**Questions:**

Could you please provide some theorectical analysis on the model effectiveness or efficiency?

---

> ### Author Response · Authors · 2025-11-21
>
> Thank you very much for taking the time to evaluate our manuscript. Below, we provide point-wise responses and will incorporate these discussions in the revision.
>
> ---
> **[W1. The approach in this paper is consist of two strategies. But there is no guidelines for choosing between Causal-EPIG-$\mu$ and Causal-EPIG-$\tau$.]**
>
> Thank you for this valuable comment. The question of guidance for choosing between Causal-EPIG-$\mu$ and Causal-EPIG-$\tau$ is a central point of our work. We argue that this is not an oversight, but rather a core finding of the paper, which we term the **Comprehensiveness-Focus Trade-off**. The paper provides explicit guidelines for this choice in Sec. 4.1 and validates them in the experimental analysis of Sec. 5. The selection is principled and depends on two main dimensions:
>
> - **The Inductive Bias of the Base Estimator:** The optimal strategy depends on the architecture of the CATE model being used.
>     * For models that directly parameterize the CATE function, such as BCF, the focused Causal-EPIG-$\tau$ strategy is a natural fit and highly effective.
>     * For models that instead characterize the full potential outcome surfaces, such as GP, the comprehensive Causal-EPIG-$\mu$ strategy offers better robustness.
> - **The Complexity of the Data-Generating Process:**
>     * In settings where the CATE signal is well-specified and learnable (e.g., our synthetic data), the focused Causal-EPIG-$\tau$ strategy yields superior sample efficiency.
>     * In more complex, lower signal-to-noise settings (e.g., the semi-synthetic IHDP and ACTG benchmarks), the comprehensive Causal-EPIG-$\mu$ strategy's robust approach of modeling the entire causal mechanism proves highly effective.
>
> Our experimental results in Sec. 5 also systematically validate this context-dependency. Therefore, the contribution of our framework is not a single "one-size-fits-all" solution, but rather a unified methodology that provides practitioners with two distinct, principled, and powerful strategies whose effectiveness is demonstrably context-dependent.
>
> ---
> **[W2. The framework relies heavily on Bayesian CATE estimators to compute posterior uncertainty. However, many practitioners use non-Bayesian estimators due to their computational efficiency and ease of implementation. These methods lack explicit posterior distributions, making Causal-EPIG inapplicable to them.]**
>
> Our framework is fundamentally information-theoretic, designed to quantify and reduce predictive uncertainty about unobservable causal estimands. Bayesian estimators, such as GP and BCF, are the well-established and principled tool for this task, as they explicitly provide the required posterior predictive distributions. We demonstrated that Causal-EPIG is not tied to one specific model, showing how it can be implemented:
>
> 1.  **Exactly** for models with analytic posteriors (like GPs).
> 2.  **Tractably** via approximation for complex, sample-based models (like BCF).
>
> You are correct that applying this framework to standard non-Bayesian estimators is not straightforward. However, the core requirement is not a *Bayesian posterior* per se, but a robust *quantification of predictive uncertainty*. A point that is made clear in the original EPIG paper [1]. This opens up several promising directions for future work, exactly as you suggest:
>
> - **Ensemble-based Uncertainty:** Many preferred non-Bayesian estimators naturally produce an ensemble of predictions. This ensemble can be treated as an empirical (pseudo-posterior) distribution. We can then approximate the required mutual information terms from this empirical distribution, similar to our approximation for the BCF posterior. This direction is highly practical, as these ensemble methods are often highly parallelizable, addressing the computational concerns that lead practitioners to choose them.
> - **Exploring Coreset-based Hybrids:** Another valuable direction is to explore coreset-based methods. Our current work compares against coreset selection (QHTE) as a baseline, which focuses on geometric diversity rather than information gain. A future hybrid approach [2] could combine our Causal-EPIG (for information gain) with a coreset objective (for diversity) to create an even more robust acquisition strategy.
>
> In summary, while our work establishes the Causal-EPIG principle using the most suitable tools for uncertainty quantification, we agree that extending it to other classes of estimators, such as by leveraging ensemble-based uncertainty, is an important and valuable next step.

---

> ### Author Response · Authors · 2025-11-21
>
> ---
> **[W3. The paper uses a 50-sample warm-start to initialize the CATE estimator, but in many real-world scenarios, initial labeled data may vary in size. A sensitivity analysis on the warm-start sample size can be helpful.]**
>
> Thank you for highlighting this. We agree that sensitivity to the initial warm-start size is an important factor for real-world applicability. We conducted this exact sensitivity analysis in App. F.5.1, "Different Starting Points".
>
> - **Methodology:** We varied the number of initial random samples used for the warm-start from 50 up to 300 (specifically, 50, 100, 200, and 300).
> - **Findings (shown in Fig. 21):** The key finding is that the superior performance of our Causal-EPIG strategies is robust to the choice of the warm-start size.
> - **Conclusion:** While a larger warm-start set naturally provides a better initial model (a lower starting PEHE) for *all* methods, the relative performance ranking remains unchanged. Causal-EPIG consistently established and maintained its performance advantage over the baselines regardless of the initial data size. This demonstrates the practical stability of our acquisition strategy.
>
> ---
> **[Q1. Could you please provide some theorectical analysis on the model effectiveness or efficiency?]**
>
> We thank the reviewer for this crucial suggestion.We have added an extensive new theoretical analysis in Sec 4.4 and App. G that directly addresses both model effectiveness and efficiency. We provide three key theoretical results (Prop. 1, Prop. 2, and Thm. 1) which we summarize below:
>
> **1. Analysis of Effectiveness (What is the optimal objective and do we converge?):**
>
> * **Prop. 1 (Defines the Optimal Objective):** We first formally prove what an "effective" strategy must target. Our new Prop. 1 shows that under a joint GP assumption, the optimal strategy to minimize CATE error is equivalent to minimizing the integrated posterior CATE variance: $\mathbb{E} _{p _{\text{tar}}(\mathbf{x})}[\mathbb{V}\text{ar} _{s+1}[\tau(\mathbf{x})]]$. This proves that the objective must account for the full joint structure, $\mathbb{V}\text{ar}[f_1] + \mathbb{V}\text{ar}[f_0] - 2\text{Cov}(f_1, f_0)$, justifying our Causal-EPIG-$\mu$ and -$\tau$ strategies as principled (not heuristic) approaches.
> * **Thm. 1 (Guarantees Convergence):** We then prove that our method is effective in achieving this objective. Our new theorem provides a formal convergence guarantee. It shows that our greedy Causal-EPIG-$\mu$-G strategy provably reduces the posterior variance until it converges to the minimal possible error (the irreducible uncertainty, $\eta^2 _{\mathcal{D} _P}(\tilde{\mathbf{x}}^*)$). This confirms the effectiveness of our method.
>
> **2. Analysis of Efficiency (How fast do we converge and why?):**
>
> * **Thm. 1 (Provides Sample Complexity):** The same Thm. 1 also provides the sample complexity (convergence rate). It bounds the reducible error by $\mathcal{O}(\gamma_{n_B}/\sqrt{n_B})$, where $\gamma_{n_B}$ is the information capacity. This formally analyzes the efficiency of our strategy.
> * **Prop. 2 (Proves Superior Efficiency):** Finally, our new Prop. 2 formally demonstrates why our prediction-oriented framework is more efficient than parameter-oriented alternatives (like Causal-EIG/BALD). Using the data processing inequality, we prove that $I(y; \theta _{\tau}) \geq I(y; \tau( \mathbf{x}^* ))$. This shows that parameter-focused methods are provably suboptimal, as they capture "wasted" information irrelevant to the final prediction. Our Causal-EPIG framework, by targeting $I(y; \tau(\mathbf{x}^*))$ directly, is therefore theoretically more sample-efficient.
>
> We believe these three newly added results now provide the rigorous analysis of both effectiveness and efficiency that the reviewer requested.
>
> [1] Bickford Smith et al., Prediction-Oriented Bayesian Active Learning, AISTATS2023.
> [2] Holzmüller et al., A Framework and Benchmark for Deep Batch Active Learning for Regression, JMLR2024.

---

> ### Comment · Reviewer_XEnf · 2025-11-24
>
> Thanks for the informative rebuttal. All my concerns have been addressed and I am happy to raise my score to 8.

---

> > ### Author Response · Authors · 2025-11-24
> >
> > Thank you very much for your helpful and positive review!

---

### Author Response · Authors · 2025-12-01
**Summary of Revisions and Response to Reviewers**

**To the AC:**

Following the PCs' guidance regarding the review process reversion, we are posting this summary to provide a clear overview for your final evaluation. We sincerely appreciate your time and effort in handling our submission under these circumstances. We have addressed all questions raised by the reviewers and significantly strengthened the paper with new theoretical results. Below is a summary of our response and the current status of the submission.

**1. Resolution of Concerns for Reviewer XEnf**

Reviewer **XEnf** has confirmed that our rebuttal successfully resolved all their concerns. Notably, our revisions included the **theoretical analysis on model effectiveness and efficiency** that they requested.

**2. Major Theoretical Additions & Confirmation of Contribution (Addressing XEnf, N6Vw, LFjm, pqm9)**

To address requests for theoretical guarantees and **concerns regarding the depth of contribution (pqm9)**, we have added a rigorous analysis in Sec. 4.4 and App. G:
* **Optimality (Prop. 1):** We prove that our information-theoretic objective is equivalent to minimizing the CATE estimation error (PEHE).
* **Convergence and Sample Complexity (Thm. 1):** We prove that our greedy strategy converges to the irreducible uncertainty with a derived sample complexity bound.
* **Fundamental Realignment & Efficiency (Prop. 2):** Addressing **pqm9's concern regarding limited innovation**, we prove (via Data Processing Inequality) that our framework represents a **fundamental realignment** of the active cauasl learning objective. We demonstrate that, in our problem setup, parameter-oriented baselines (e.g., Causal-BALD, Causal-EIG) are theoretically suboptimal compared to our prediction-oriented framework, establishing our work as a **necessary theoretical correction** rather than a minor modification.

**3. Clarifications on Scope and Robustness (Addressing pqm9, LFjm, N6Vw)**

We have provided detailed clarifications to address specific technical queries:

* **Assumption of Unconfoundedness (pqm9):** We clarified that unmeasured confounding (identification) is an **orthogonal problem** to our focus on active estimation (expensive outcome acquisition, sample efficiency). We adhere to the standard problem setting in Active CATE Estimation (e.g., Causal-BALD, QHTE).
* **Scalability & High-Dimensionality (pqm9):** We clarified that our main strategies are **designed to be scalable**. The acquisition complexity is **linear** regarding pool/target size ($\mathcal{O}(n_P \cdot n_{\text{tar}})$). For high-dimensional data, we demonstrated in our ablation study that our framework works seamlessly with **Deep Kernel GPs**, allowing us to leverage neural networks to handle high-dimensional covariates.
* **Distribution Shift (LFjm):** We clarified that our framework explicitly handles covariate shift by integrating information gain over the **target distribution** (Transductive setting). This ensures the acquired samples effectively reduce uncertainty for the specific population of interest.
* **Robustness to Posterior Quality (pqm9, LFjm):** We clarified that Active Learning relies on effective **ranking** rather than perfect calibration. Our consistent performance across diverse backbones demonstrates robustness to potential specification issues.
* **Adaptability to Non-Bayesian Models (XEnf, N6Vw):** We clarified that our framework is adaptable to any probabilistic estimator capable of estimating joint predictive moments (e.g., Deep Ensembles, MC Dropout). While we utilized Bayesian models (BCF/CMGP/NSGP) in our experiments, as they are widely accepted as the robust methods for CATE estimation, the acquisition framework itself is **model-agnostic**.

We believe the manuscript now offers both a novel methodological framework and rigorous theoretical guarantees.

---

### Meta-Review · Area_Chair_bJLC · 2026-01-06

**Summary:**

The only supportive reviewer Reviewer XEnf initially had a marginally above the threshold score. All other reviewers recommend rejection with varying scores. I should note that the supportive Reviewer XEnf mentioned they would like to increase their score from marginally above to accept (8).

Same as the reviewers who recommend rejection, Reviewer XEnf also criticized the need for a Bayesian posterior.

Reviewer pqm9 is concerned with the "no unobservable confounding" assumption as well as the assumption that we have access to a properly calibrated posterior.

Reviewer LFjm had the same concern about dependence on posterior estimation. They were also additionally concerned with the lack of novelty since the method is an extension of EPIG. The lack of theoretical guarantees was also echoed by this reviewer.

**Reviewer Concerns:**

The authors' rebuttal addressed some concerns of Reviewer pqm9 in my opinion. For example, although no unobserved confounder may be a strong assumption in certain contexts, it is a starting point that is often acceptable. However, the authors' reply to the sensitivity of their method to posterior calculation, which basically says any Bayesian method is, is likely unsatisfactory.

The authors added Proposition 1 to address some of Reviewer LFjm's concerns around novelty and lack of theoretical justification. However, another review cycle is likely needed to assess this before we can consider it as part of the paper's contributions.

**Reviewer Scores:**

It is unlikely that the reviewers who recommended rejection would have increased their score. It is very likely that they would recommend another review cycle to assess the soudness of the newly added theoretical results.

---

### Decision · Program_Chairs · 2026-01-26

Reject